# Cell-type-specific regulation of neuronal intrinsic excitability by macroautophagy

Ori J Lieberman[1]*, Micah D Frier[1], Avery F McGuirt[1], Christopher J Griffey[2], Elizabeth Rafikian[3], Mu Yang[3], Ai Yamamoto[2], Anders Borgkvist[4], Emanuela Santini[4], David Sulzer[1,2,5,6]*

[1]Department of Psychiatry, Columbia University Vagelos College of Physicians and Surgeons, New York, United States; [2]Department of Neurology, Columbia University Vagelos College of Physicians and Surgeons, New York, United States; [3]Mouse NeuroBehavior Core, Institute for Genomic Medicine, Columbia University Vagelos College of Physicians and Surgeons, New York, United States; [4]Department of Neuroscience, Karolinska Institute, Stockholm, Sweden; [5]Department of Pharmacology, Columbia University Vagelos College of Physicians and Surgeons, New York, United States; [6]Division of Molecular Therapeutics, New York State Psychiatric Institute, New York, United States

**Abstract** The basal ganglia are a group of subcortical nuclei that contribute to action selection and reinforcement learning. The principal neurons of the striatum, spiny projection neurons of the direct (dSPN) and indirect (iSPN) pathways, maintain low intrinsic excitability, requiring convergent excitatory inputs to fire. Here, we examined the role of autophagy in mouse SPN physiology and animal behavior by generating conditional knockouts of Atg7 in either dSPNs or iSPNs. Loss of autophagy in either SPN population led to changes in motor learning but distinct effects on cellular physiology. dSPNs, but not iSPNs, required autophagy for normal dendritic structure and synaptic input. In contrast, iSPNs, but not dSPNs, were intrinsically hyperexcitable due to reduced function of the inwardly rectifying potassium channel, Kir2. These findings define a novel mechanism by which autophagy regulates neuronal activity: control of intrinsic excitability via the regulation of potassium channel function.

***\*For correspondence:***
ojl2106@columbia.edu (OJL);
ds43@cumc.columbia.edu (DS)

**Competing interests:** The authors declare that no competing interests exist.

## Introduction

Although classically studied in the context of neurodegeneration, macroautophagy, hereafter referred to as autophagy, has recently been identified as a critical regulator of neuronal activity and function (*Friedman et al., 2012*; *Hernandez et al., 2012*; *Nikoletopoulou and Tavernarakis, 2018*; *Nixon, 2013*; *Tang et al., 2014*; *Yamamoto and Yue, 2014*). Autophagy is a process in which proteins or organelles are enclosed in a double membrane organelle, known as an autophagosome, which then traffics to the lysosome for degradation of its cargo (*Klionsky, 2007*). Autophagosomes also fuse with recycling endosomes, to form amphisomes that act as key intermediates for the degradation of endosomal cargo including plasma membrane proteins (*Eskelinen, 2005*; *Yamamoto and Yue, 2014*).

Autophagy plays evolutionarily conserved roles in the regulation of synapse formation, structure and function. In early development, autophagic function is required presynaptically for axon pathfinding and circuit formation (*Dragich et al., 2016*; *Stavoe et al., 2016*) and, in mature neural circuits, the absence of presynaptic autophagy leads to deficits in neurotransmitter release and synaptic vesicle recycling due to the accumulation of damaged synaptic vesicles or mitochondria (*Hernandez et al., 2012*; *Hoffmann et al., 2019*; *Vijayan and Verstreken, 2017*). In excitatory

projection neurons of the hippocampus and cortex, autophagy controls neurotransmission postsynaptically via the degradation of ionotropic glutamate and GABA receptors and synaptic scaffolding proteins such as PSD95 and Arc (*Nikoletopoulou et al., 2017*; *Rowland, 2006*; *Shehata et al., 2012*; *Sumitomo et al., 2018*; *Tang et al., 2014*; *Yan et al., 2018*). Developmental synaptic pruning in the cortex is dependent on both neuronal and microglial autophagy (*Kim et al., 2017*; *Lieberman et al., 2019b*; *Tang et al., 2014*) and autophagic degradation of unidentified neuronal proteins contributes to hippocampal synaptic plasticity (*Glatigny et al., 2019*; *Nikoletopoulou et al., 2017*). Thus, autophagy is a ubiquitous cellular process critical to proper synaptic function in the central nervous system.

The brain is, however, composed of heterogenous populations of neurons with distinctive patterns of activity determined by the interplay of a complement of ion channels and synaptic inputs. Whether autophagy regulates neurotransmission via distinct processes in specific classes of neurons remains unknown.

The striatum is the primary input nucleus of the basal ganglia, a collection of subcortical brain structures required for learning and executing goal-directed behaviors. In contrast to hippocampal and cortical pyramidal neurons where the contribution of autophagy to neurotransmission has been most studied, the striatum is composed of two intermixed and morphologically indistinguishable classes of GABAergic projection neurons: direct pathway (dSPN) and indirect pathway (iSPN) spiny projection neurons (*Gerfen and Surmeier, 2011*). SPNs receive convergent excitatory inputs from cortical and thalamic projection neurons, inhibitory input from local interneurons, and a variety of modulatory synaptic inputs (*Doig et al., 2010*; *Gerfen and Surmeier, 2011*). SPN activity in response to these inputs is governed by their low intrinsic excitability and high latency to fire, necessitating coordinated excitatory synaptic inputs on dendritic spines and shafts to elicit an action potential (*Wilson and Kawaguchi, 1996*). The disruption of normal SPN activity produces deficits in motor activity (*Durieux et al., 2012*; *Kravitz et al., 2010*); motivation (*Carvalho Poyraz et al., 2016*; *Kellendonk et al., 2006*); and responses to drugs of abuse (*Dong et al., 2006*). Striatal dysfunction has also been linked to autism spectrum disorders (*Chang et al., 2015*; *Fuccillo, 2016*), a disease in which autophagic dysfunction has been reported in mouse models and postmortem human brain tissue (*Poultney et al., 2013*; *Tang et al., 2014*; *Yan et al., 2018*). How autophagy contributes to striatal function and SPN physiology, however, remains unknown.

Here, we generated conditional knockouts of Atg7, a protein required for autophagy (*Komatsu et al., 2005*), in dSPNs and iSPNs and find that loss of Atg7 in either SPN subtype led to behavioral abnormalities in the absence of neurodegeneration. Loss of Atg7 in dSPNs led to alterations in their dendritic arbor, dendritic spines and synaptic inputs, similar to that reported in pyramidal neurons. In contrast, dendritic arborization, spine density and synaptic inputs were not changed in iSPNs lacking Atg7. Instead, Atg7 in iSPNs was required for the function of the inwardly rectifying Kir2 channel, which maintains the hyperpolarized resting membrane potential and low input resistance that are characteristic of SPNs. We found that ongoing Atg5- and Atg7-dependent autophagy contributes to the lysosomal delivery of endocytosed Kir2 channels and, in its absence, Kir2 channel abundance was *elevated* on the plasma membrane but exhibited *reduced* activity. We further found that the excess channels are inactivated by acetylation, explaining the intrinsic hyperexcitability of Atg7-deficient iSPNs. These results introduce the regulation of neuronal intrinsic excitability by autophagy, indicate how this occurs at a specific molecular target, and demonstrate a role for this pathway in normal behaviors.

## Results

### Generation of conditional knockout mice lacking autophagy in dSPNs or iSPNs

To address the role of autophagy in SPN physiology and striatal function, we generated separate lines of mice lacking Atg7 in dSPNs or iSPNs using Cre driver lines (CreDrd1aey262 for dSPNs and CreAdora2aKG139 for iSPNs) (*Figure 1A–B*) (*Gong et al., 2007*; *Komatsu et al., 2006*; *Komatsu et al., 2005*). Mice lacking Atg7 in dSPNS or iSPNs (referred to as dSPN$^{Atg7cKO}$ and iSPN$^{Atg7cKO}$, respectively) were born at Mendelian ratios and survived into adulthood (data not shown). Littermate control mice harbored the floxed Atg7 allele without the Cre driver. We conducted a

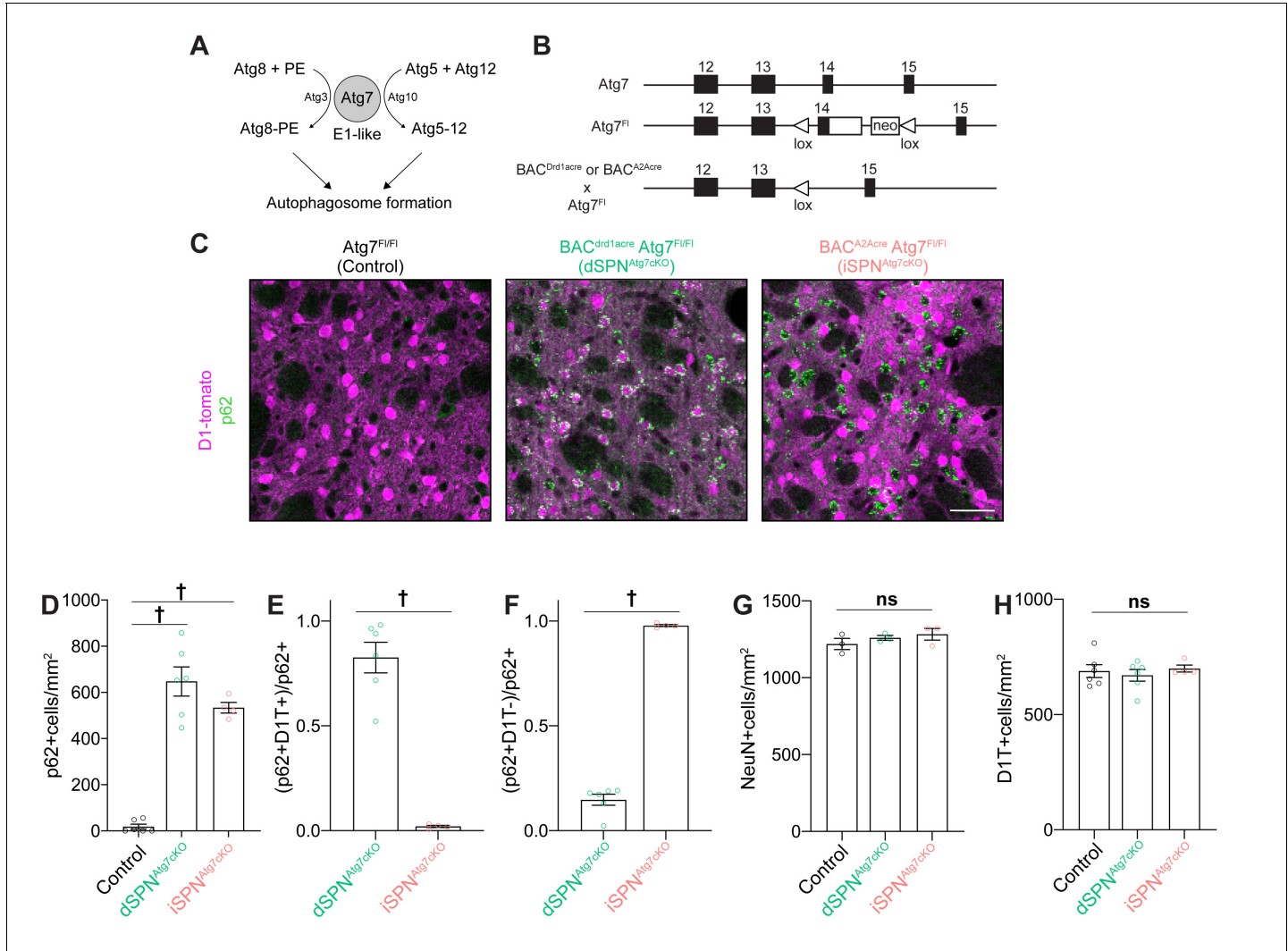

**Figure 1.** Specific loss of Atg7-mediated autophagy in dSPNs or iSPNs in the absence of neurodegeneration in dSPN$^{Atg7cKO}$ mice and iSPN$^{Atg7cKO}$, respectively. (A) Schematic representation of the role of Atg3, 5, 7, 8, 10, and Atg12 in a cascade leading to autophagosome formation. (B) Schematic of Atg7 locus in Atg7$^{Fl/Fl}$ mice and following Cre-mediated recombination. (Adapted from *Komatsu et al., 2006*). (C) Immunofluorescent images of striatal sections from Atg7$^{Fl/Fl}$ (Control), D1-cre Atg7$^{Fl/Fl}$ (dSPN$^{Atg7cKO}$) or A2Acre-Atg7$^{Fl/Fl}$ (iSPN$^{Atg7cKO}$) mice. (D) p62$^+$ cells per field in control, dSPN$^{Atg7cKO}$ or iSPN$^{Atg7cKO}$ mice. Control: N=6 mice, dSPN$^{Atg7cKO}$: N = 6 mice, iSPN$^{Atg7cKO}$: N=4 mice. Data analyzed by one-way ANOVA, $F_{(2,13)}$=65.73, p = 0.001. (E-F) Number of (E) D1-tomato+ or (F) D1-Tomato-, p62$^+$ cells in dSPNAtg7cKO and iSPN$^{Atg7cKO}$ mice. N = 6 dSPN$^{Atg7cKO}$ mice and N=4 iSPN$^{Atg7cKO}$ mice. Data in (E-F) were analyzed by two-tailed unpaired t test. (E) $t_8$=8.816, p = 0.0001. (F) $t_8$=24.94, p = 0.0001. (G-H) There were no differences in NeuN$^+$ [Control: N = 3 mice, dSPN$^{Atg7cKO}$: N = 3 mice, iSPN$^{Atg7cKO}$: N = 3 mice; analyzed by one-way ANOVA $F_{(2,6)}$ = 1.019, p = 0.4160] or D1-tomato+ cells per field [Control: N=6 mice, dSPNAtg7cKO: N = 6 mice, iSPNAtg7cKO: N=4 mice; analyzed by one-way ANOVA F (2,13) = 0.3144, p = 0.7356]. ns p>0.05, * p<0.05, ** p<0.01, *** p<0.001, † p<0.0001.

subset of electrophysiological and biochemical experiments in Cre$^+$ Atg7$^{wt}$ mice but found no effect of Cre expression compared to Cre$^-$ Atg7$^{Fl/Fl}$ controls and have thus combined these data.

To identify SPN subtype, we utilized BAC transgenic mice expressing the tdTomato fluorescent protein under the dopamine D1 receptor promoter (D1-tomato), which specifically labels dSPNs (*Ade et al., 2011*) and crossed this line into control, iSPN$^{Atg7cKO}$ and dSPN$^{Atg7cKO}$ mice. A Cre-dependent reporter at the Rosa26 locus could have been used to mark recombined cells but the Atg7 locus is close to the Rosa26 locus on chromosome six and we were therefore unable to generate recombinants (data not shown).

As SPNs represent >95% of neurons in the striatum, D1-tomato negative cells are predominantly iSPNs (*Shuen et al., 2008*). In the absence of autophagy, cytosolic inclusions containing the

autophagy cargo receptor, p62, form. p62 puncta were selectively observed in ~80% of D1-tomato-positive neurons (presumed dSPNs) in dSPNAtg7cKO mice and ~95% D1-tomato-negative (presumed iSPNs) in iSPN[Atg7cKO] mice (*Figure 1C–F*). We note that in dSPN[Atg7cKO] mice ~ 15% of p62$^+$ cells were D1-tomato-negative, suggesting that some non-specific Cre-mediated recombination of the 'floxed' Atg7 allele occurs as previously described for this Cre driver line. Finally, we confirmed that similar numbers of p62+ cells are present in the striatum of dSPN[Atg7cKO] and iSPN[Atg7cKO] mice at P28, when electrophysiological recordings in Figures 3 and 4 were collected (data not shown).

Loss of neuronal autophagy can lead to neurodegeneration (*Hara et al., 2006*; *Komatsu et al., 2006*). We found no difference in the number of NeuN$^+$ or D1-Tomato$^+$ cells at 5 months of age in dSPN[Atg7cKO] or iSPN[Atg7cKO] mice compared to controls (*Figure 1G–H*). Together, these results demonstrate that these conditional knockouts permit the analysis of the consequences of the specific loss of dSPN or iSPN autophagy in the absence of neurodegeneration.

## Loss of autophagy in SPNs leads to behavioral deficits

We generated multiple cohorts of male and female dSPN[Atg7cKO] and iSPN[Atg7cKO] mice along with respective littermate controls. We combined data from the littermate control groups of each cross as there was no difference between littermate controls from these two crosses in any behavioral task. We included approximately equal numbers of male and female mice from each genotype in these cohorts and did not find an interaction between sex and genotype. *Table 1* includes behavioral results split by sex.

We identified several behavioral and phenotypic consequences of loss of autophagy in SPNs. Both male and female dSPN[Atg7cKO] mice weighed less than control mice from weaning until 3 months of age while iSPN[Atg7cKO] mice were indistinguishable from littermate controls (*Figure 2A*).

We assessed motor learning, a task dependent on the striatum, as well as other brain regions, using the accelerating rotarod (*Durieux et al., 2012*). Both dSPN[Atg7cKO] and iSPN[Atg7cKO] mice had a significantly lower latency to fall off the rotarod than controls on all three trials, indicating that Atg7 in dSPNs and iSPNs was required for motor learning and performance (*Figure 2B*). We note that differences in motor learning may exist between dSPN[Atg7cKO] and iSPN[Atg7cKO] mice as the rotarod test is highly sensitive to body weight (*McFadyen et al., 2003*) and dSPN[Atg7cKO] mice weight significantly less than iSPN[Atg7cKO] mice (*Figure 2A*).

Changes in striatal activity can also lead to alterations in locomotor behavior in a novel environment (*Durieux et al., 2012*). iSPN[Atg7cKO] mice displayed hyperactivity in the open-field arena, which arose during the later habituation phase, than controls while dSPN[Atg7cKO] were not different (*Figure 2C–D*). Automated scoring of iSPN[Atg7cKO] mutants during the open field test revealed increased stereotypies (*Figure 2E*), which was confirmed by manually scoring self-directed grooming bouts in a separate session (*Figure 2F*). These results demonstrate that loss of Atg7 in dSPNs or iSPNs leads to behavioral deficits and disruptions in striatal function.

## Autophagy is required for proper excitatory synapse function on dSPNs but not iSPNs

Autophagy regulates dendritic spine density and both excitatory and inhibitory synaptic transmission in the cortex and hippocampus (*Nikoletopoulou and Tavernarakis, 2018*; *Sumitomo et al., 2018*; *Tang et al., 2014*; *Yan et al., 2018*). Due to the deficit in motor learning in dSPN[Atg7cKO] and iSPN[Atg7cKO] mice, we examined if synaptic deficits were present on dSPNs or iSPNs in the dorsal striatum, a striatal subregion required for motor learning (*Durieux et al., 2012*; *Yin et al., 2009*). Note that we have combined data from both sexes in this analysis. We did not design these experiments to detect sex differences as there was no interaction between sex and genotype in the behavioral analysis.

Reconstruction of the dendritic tree from neurobiotin-filled dSPNs (D1-tomato-positive cells) revealed reductions in the total dendritic length and the complexity of the dendritic tree in dSPNs from dSPN[Atg7cKO] mice compared to controls (*Figure 3A–D*). In contrast, loss of autophagy in iSPNs from iSPN[Atg7cKO] mice did not affect total dendritic length or dendritic complexity (*Figure 3A–D*). We also found a reduction in the dendritic spine density on dSPNs from dSPN[Atg7cKO] mice while the dendritic spine density on iSPNs from iSPN[Atg7cKO] mice was not different than controls (*Figure 3E*).

**Table 1.** Statistics split by sex for behavioral experiments in *Figure 2*.

| Sex | Genotype (N) | Mean (SEM) | Statistics |
|---|---|---|---|
| **Rotarod – learning rate (rpm/trial; trial 3- trial 1)** | | | |
| Male | Control (28) | 57.29 (10.17) | p = 0.0088 |
| | dSPN[Atg7cKO] (20) | 15.85 (8.333) | |
| | iSPN[Atg7cKO] (11) | 22.45 (13.17) | |
| Female | Control (22) | 57.18 (12.89) | p = 0.0029 |
| | dSPN[Atg7cKO] (14) | 38.43 (12.69) | |
| | iSPN[Atg7cKO] (12) | −11.92 (12.33) | |
| Combined | Control (50) | 57.24 (7.954) | p = 0.0001 |
| | dSPN[Atg7cKO] (34) | 25.15 (7.306) | |
| | iSPN[Atg7cKO] (23) | 4.522 (9.526) | |
| Sex x Genotype $F_{(2,101)}$ = 2.218 p = 0.1141 | | | |
| **Weight (g)** | | | |
| Male | Control (31) | 26.07 (0.4074) | p = 0.0003 |
| | dSPN[Atg7cKO] (19) | 22.66 (0.8216) | |
| | iSPN[Atg7cKO] (12) | 24.67 (0.6438) | |
| Female | Control (31) | 21.15 (0.4341) | p = 0.0003 |
| | dSPN[Atg7cKO] (11) | 17.85 (0.5850) | |
| | iSPN[Atg7cKO] (13) | 20.29 (0.4439) | |
| Sex x Genotype $F_{(2,111)}$ = 0.1073 p = 0.8983 | | | |
| **Open Field – Distance traveled (cm)** | | | |
| Male | Control (17) | 6777 (397.9) | p = 0.0159 |
| | dSPN[Atg7cKO] (7) | 8073 (883.5) | |
| | iSPN[Atg7cKO] (6) | 9318 (573.1) | |
| Female | Control (15) | 9307 (369.4) | p = 0.0257 |
| | dSPN[Atg7cKO] (7) | 7967 (773.3) | |
| | iSPN[Atg7cKO] (8) | 10433 (584.5) | |
| Combined | Control (32) | 7963 (351.7) | p = 0.0048 |
| | dSPN[Atg7cKO] (15) | 7762 (585.4) | |
| | iSPN[Atg7cKO] (15) | 9955 (427.4) | |
| Sex x Genotype $F_{(2,54)}$ = 3.045 p = 0.0558 | | | |
| **Open Field – Stereotypies (data normalized)** | | | |
| Male | iSPN[Ctrl] (7) | 1.00 (0.030) | p = 0.0293 |
| | iSPN[cKO] (6) | 1.136 (0.047) | |
| Female | iSPN[Ctrl] (8) | 1.00 (0.038) | p = 0.0819 |
| | iSPN[cKO] (8) | 1.094 (0.032) | |
| Combined | iSPN[Ctrl] (15) | 1.00 (0.024) | p = 0.0043 |
| | iSPN[cKO] (14) | 1.112 (0.027) | |
| Sex x genotype $F_{(1,25)}$ = 0.3226 p = 0.5751 | | | |
| **Grooming bouts** | | | |
| Male | iSPN[Ctrl] (11) | 3.091 (0.8252) | p = 0.0542 |
| | iSPN[cKO] (13) | 5.846 (1.031) | |
| Female | iSPN[Ctrl] (12) | 3.500 (0.4174) | p = 0.0048 |
| | iSPN[cKO] (13) | 7.385 (1.130) | |
| Combined | iSPN[Ctrl] (23) | 3.304 (0.4420) | p = 0.0007 |
| | iSPN[cKO] (26) | 6.615 (0.7648) | |

Sex x genotype $F_{(1,45)}$ = 0.3781 p = 0.5417.

Changes in dendritic spine density and dendritic architecture may lead to functional differences in excitatory or inhibitory inputs onto SPNs. We recorded miniature excitatory postsynaptic currents

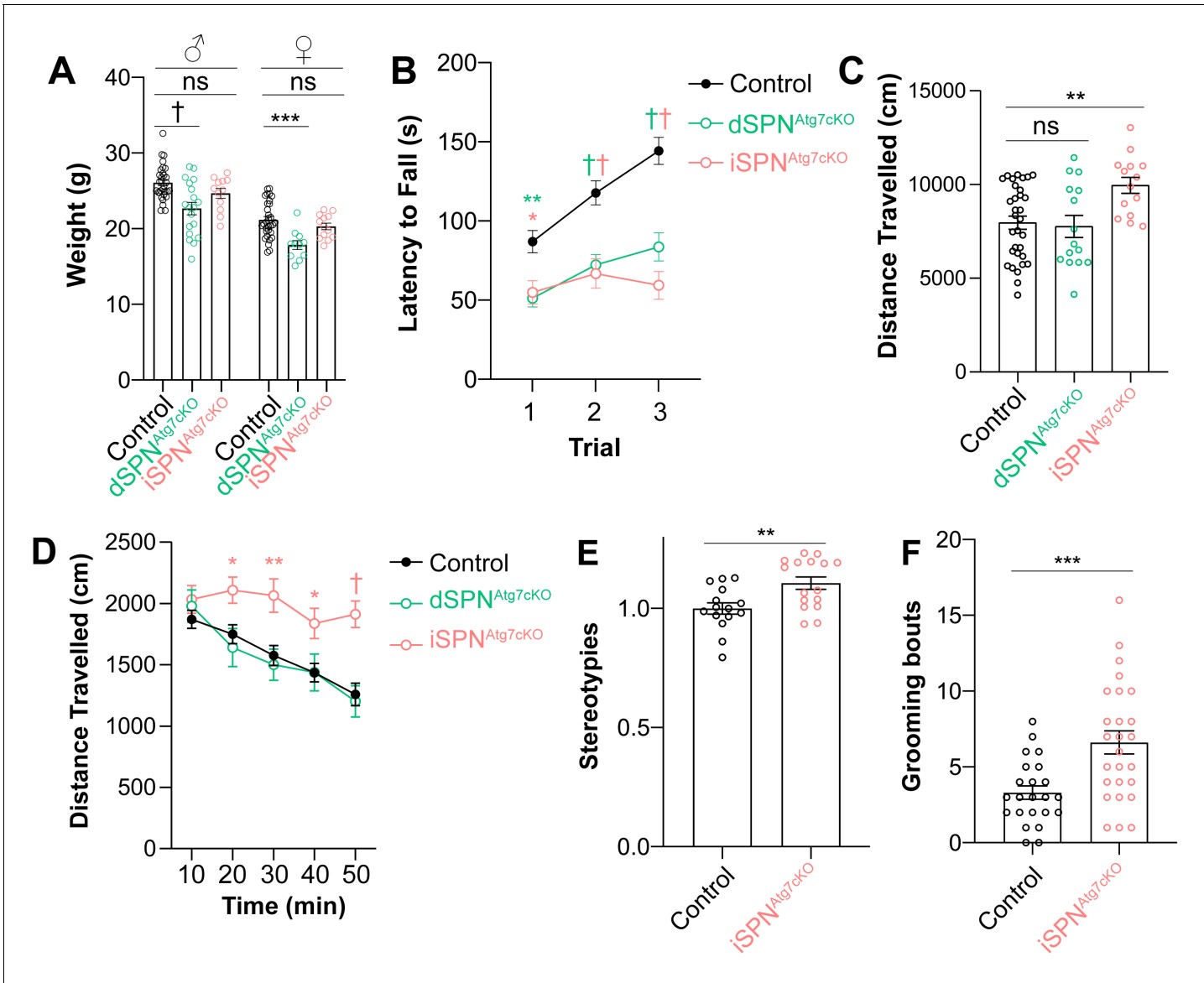

**Figure 2.** Atg7 in SPNs is required for motor performance and learning. (**A**) Nine-week-old male and female dSPN[Atg7cKO] mice, but not iSPN[Atg7cKO] mice, weigh less than controls. Data were analyzed by two-way ANOVA followed by Bonferroni post-hoc test. Sex x Genotype: $F_{(2,111)}$ = 0.1073, p = 0.8983; Genotype: $F_{(2,111)}$ = 17.58, p<0.0001; Sex: $F_{(1,111)}$ = 87.65, p<0.0001. (**B**) Both dSPN[Atg7cKO] and iSPN[Atg7cKO] mice have a lower latency to fall off the accelerating rotarod. Control n = 50, dSPN[Atg7cKO]n = 34, iSPN[Atg7cKO]n = 23. Data were analyzed by two-way ANOVA followed by Bonferroni post-hoc test. Trial x Genotype: $F_{(4,208)}$ = 6.198, p<0.0001; Trial: $F_{(2,208)}$ = 26.34, p<0.0001; Genotype: $F_{(2,104)}$ = 20.73, p<0.0001. (**C**) iSPN[Atg7cKO] mice, but not dSPN[Atg7cKO], demonstrate locomotor hyperactivity in the open field arena. One-way ANOVA followed by Bonferroni post-hoc test. (**D**) Time course of locomotor activity in the open field. Control n = 32, dSPN[Atg7cKO]n = 15, iSPN[Atg7cKO]n = 14. Data were analyzed by two-way ANOVA followed by Bonferroni post-hoc test. Time x Genotype: $F_{(8,240)}$ = 2.547, p = 0.0111; Time: $F_{(4,240)}$ = 21.99, p<0.0001; Genotype: $F_{(2,60)}$ = 6.270, p = 0.0034. (**E**) Automated scoring of stereotypies in the open field. Control: n = 15; iSPN[Atg7cKO]: n = 16. Data analyzed by two-tailed, unpaired t test. $t_{29}$ = 2.994, p = 0.0056. (**F**) Manual scoring of grooming bouts over thirty minutes following habituation during a separate session. Control: n = 23; iSPN[Atg7cKO]: n = 26. Data analyzed by two tailed, unpaired t test. $t_{47}$ = 3.623, p = 0.0007. See **Table 1** for detailed statistics split by sex. ns p>0.05, *p<0.05, **p<0.01, ***p<0.001, † p<0.0001.

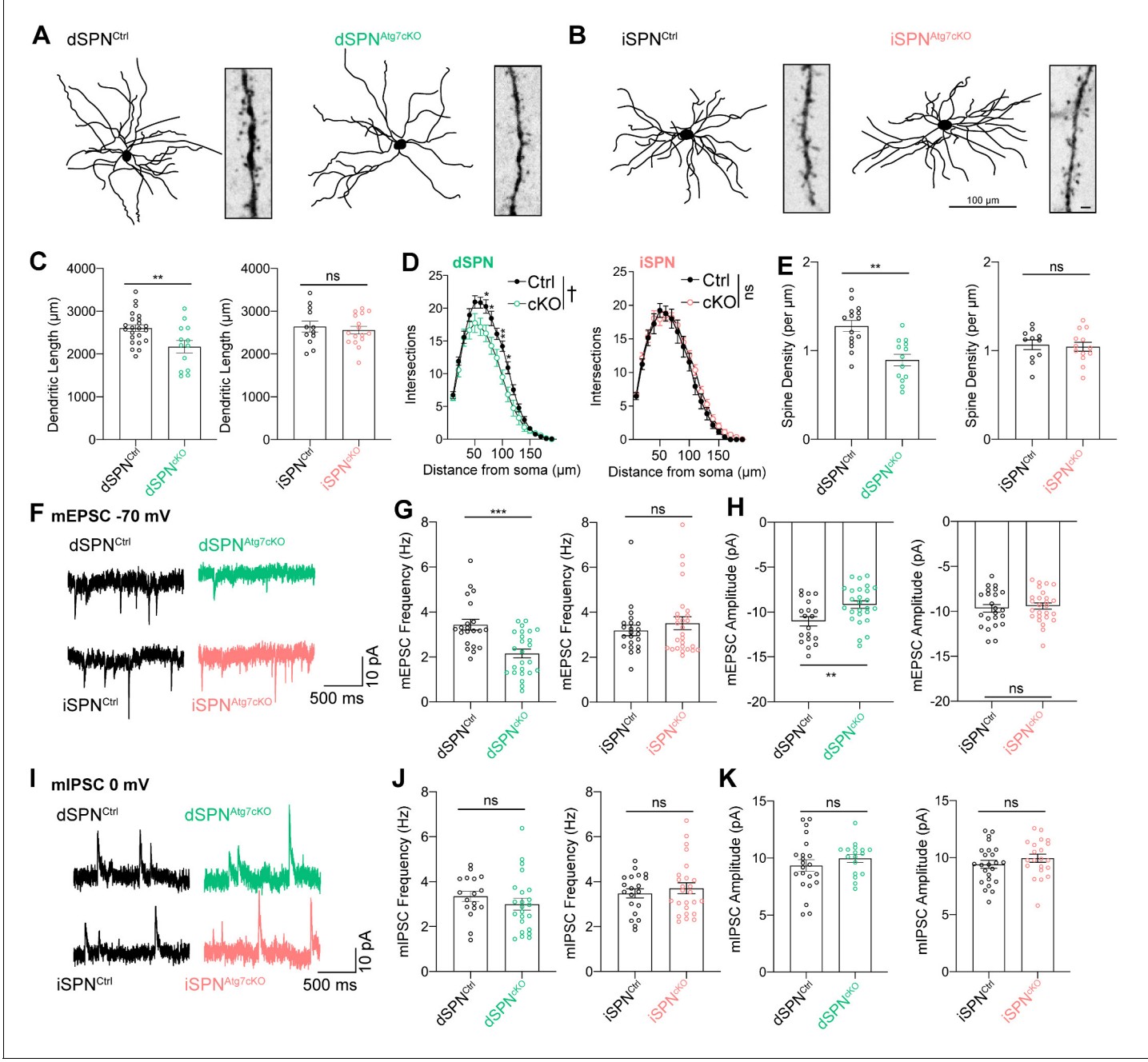

**Figure 3.** Atg7 contributes to synaptic function and dendritic complexity in dSPNs but not iSPNs. (**A**) Sample dendritic trees of dSPNs from control of dSPN^Atg7cKO mice. Reconstructions of neurobiotin filled neurons (left) and dendritic segment (right). (**B**) Sample dendritic trees from reconstructed iSPNs in control or iSPN^Atg7cKO mice. Left: reconstructed dendritic tree, scale bar 100 μm. Right: dendritic segment, scale bar 1 μm. (**C**) Cumulative dendritic length is significantly reduced in dSPNs from dSPN^Atg7cKO mice compared to control (left) but not in iSPNs from iSPN^Atg7cKO mice compared to control (right). dSPN^Ctrl: n = 22 cells, five mice, dSPN^cKO: n = 12,4. iSPN^Ctrl: n = 12,3; iSPN^cKO: n = 16,3. (**D**) Sholl analysis reveals a significant reduction in dendritic complexity in dSPNs from dSPN^Atg7cKO mice compared to control (left) but not in iSPNs from iSPN^Atg7cKO mice compared to control (right). dSPN^Ctrl: n = 22, 5, dSPN^cKO: n = 12,4. iSPN^Ctrl: n = 13,3; iSPN^cKO: n = 16,3. (**E**) Dendritic spine density on dendritic segments 50–100 μm from the soma. dSPN^Ctrl: n = 15, 5, dSPN^cKO: n = 13,4. iSPN^Ctrl: n = 8,3; iSPN^cKO: n = 8,3. (**F**) Representative traces of mEPSCs in dSPNs (top) and iSPNs (bottom). (**G-H**) A significant reduction in (**G**) mEPSC frequency and (**H**) mEPSC amplitude in dSPNs from dSPN^Atg7cKO mice compared to control but no difference in iSPN mEPSC frequency or amplitude between genotypes. Frequency: dSPN^Ctrl: n = 21, 5, dSPN^cKO: n = 25,4. iSPN^Ctrl: n = 22,3; iSPN^cKO: n = 26,4. Amplitude: dSPN^Ctrl: n = 19, 5, dSPN^cKO: n = 26,4. iSPN^Ctrl: n = 23,5; iSPN^cKO: n = 26,4. (**I**) Representative traces of mIPSCs in dSPNs (top) and iSPNs (bottom) (**J-K**) No difference in mIPSC frequency or amplitude after loss of autophagy in either dSPNs or iSPNs. Frequency: dSPN^Ctrl: n = 17, 5,

*Figure 3 continued on next page*

*Figure 3 continued*

dSPN$^{cKO}$: n = 23,4. iSPN$^{Ctrl}$: n = 20,5; iSPN$^{cKO}$: n = 25,4. Amplitude: dSPN$^{Ctrl}$: n = 17,5, dSPN$^{cKO}$: n = 22,4. iSPN$^{Ctrl}$: n = 21,5; iSPN$^{cKO}$: n = 24,4. See **Table 2** for detailed statistics. ns p>0.05, *p<0.05, **p<0.01, † p<0.0001.

(mEPSC; $V_{Hold}$ = −70 mV) and miniature inhibitory postsynaptic currents (mIPSC; $V_{Hold}$ = 0 mV) in whole-cell patch clamp recordings in the presence of tetrodotoxin (TTX) to block action potentials. The use of this internal solution permits measurement of both mEPSCs and mIPSCs in individual cells. In a subset of cells, we confirmed that inward currents at $V_{Hold}$ = −70 mV were glutamatergic in origin by bath applying the AMPAR antagonist, CNQX (5 µM) and outward currents at $V_{Hold}$ = 0 mV were GABAergic by bath application of picrotoxin (25 µM; data not shown). We found a reduction in both mEPSC frequency and amplitude in dSPNs lacking Atg7 without a change in mIPSC frequency or amplitude (*Figure 3F–K*). In iSPNs lacking Atg7, in contrast, neither mEPSC frequency and amplitude nor mIPSC frequency and amplitude were different than controls (*Figure 3F–K*), consistent with the absence of a change in dendritic length or spine density. These data demonstrate that, as in excitatory neurons of the cortex and hippocampus (*Tang et al., 2014*; *Yan et al., 2018*), autophagy contributes to synaptic function in dSPNs but is dispensable for proper synaptic transmission onto iSPNs.

**Table 2.** Detailed statistics for *Figure 3*.

| Figure | Groups (n;N)[*]: Mean (sem) | Test[$] | Results | p-value |
|---|---|---|---|---|
| 3C (left) | dSPN$^{Ctrl}$ (23;5): 26.05 (77.83)<br>dSPN$^{Atg7cKO}$ (13;4): 2172 (145.6) | Two-tailed, unpaired t test | $T_{34}$ = 2.888 | **0.0067** |
| 3C (right) | iSPN$^{Ctrl}$ (12;3): 2640 (128.4)<br>iSPN$^{Atg7cKO}$ (16;3): 2556 (90.11) | Two-tailed, unpaired t test | $T_{26}$ = 0.5380 | 0.5952 |
| 3D (left) | dSPN$^{Ctrl}$ (22;5) dSPN$^{Atg7cKO}$ (12;4) | Two-way repeated measures ANOVA | Distance: $F_{(18,576)}$ = 198.9<br>Genotype: $F_{(1,32)}$ = 7.124<br>Interaction: $F_{(18,576)}$ = 2.981 | **Distance:<0.0001**<br>**Genotype: 0.0118**<br>**Intx:<0.0001** |
| 3D (right) | iSPN$^{Ctrl}$ (13;3) iSPN$^{Atg7cKO}$ (16;3) | Two-way repeated measures ANOVA | Distance: $F_{(18,486)}$ = 204.9<br>Genotype: $F_{(1,27)}$ = 0.8747<br>Interaction: $F_{(18,486)}$ = 0.5927 | **Distance:<0.0001**<br>Genotype: 0.3580<br>Intx: 0.9057 |
| 3E (left) | dSPN$^{Ctrl}$ (15;5): 1.428 (0.062)<br>dSPN$^{Atg7cKO}$ (13;4): 0.8928 (0.066) | Two-tailed, unpaired t test | $T_{26}$ = 4.286 | **0.0002** |
| 3E (right) | iSPN$^{Ctrl}$ (8;3): 1.062 (0.07246)<br>iSPN$^{Atg7cKO}$ (8;3): 1.054 (0.06648) | Two-tailed, unpaired t test | $T_{16}$ = 0.07856 | 0.9384 |
| 3G (left) | dSPN$^{Ctrl}$ (21;5): 3.435 (0.2402)<br>dSPN$^{Atg7cKO}$ (25;4): 2.155 (0.1928) | Two-tailed, unpaired t test | $T_{44}$ = 4.206 | **0.0001** |
| 3G (right) | iSPN$^{Ctrl}$ (22;5): 3.192 (0.2325)<br>iSPN$^{Atg7cKO}$ (26;4): 3.344 (0.2382) | Two-tailed, unpaired t test | $T_{46}$ = 0.4523 | 0.6531 |
| 3H (left) | dSPN$^{Ctrl}$ (19;5): −11.03 (0.5341)<br>dSPN$^{Atg7cKO}$ (26;4): −9.174 (0.4162) | Two-tailed, unpaired t test | $T_{43}$ = 2.788 | **0.0079** |
| 3H (right) | iSPN$^{Ctrl}$ (23;5): −9.669 (0.4282)<br>iSPN$^{Atg7cKO}$ (26;4): −9.417 (0.3507) | Two-tailed, unpaired t test | $T_{47}$ = 0.4599 | 0.647 |
| 3J (left) | dSPN$^{Ctrl}$ (17;5): 3.351 (0.2287)<br>dSPN$^{Atg7cKO}$ (23;4): 3.000 (0.2634) | Two-tailed, unpaired t test | $T_{38}$ = 0.9654 | 0.3404 |
| 3J (right) | iSPN$^{Ctrl}$ (20;5): 3.480 (0.2026)<br>iSPN$^{Atg7cKO}$ (25;4): 3.708 (0.2449) | Two-tailed, unpaired t test | $T_{43}$ = 0.6942 | 0.4913 |
| 3K (left) | dSPN$^{Ctrl}$ (17;5): 9.972 (0.3499)<br>dSPN$^{Atg7cKO}$ (22;4): 9.346 (0.5072) | Two-tailed, unpaired t test | $T_{37}$ = 0.9558 | 0.3454 |
| 3K (right) | iSPN$^{Ctrl}$ (21;5): 9.950 (0.3571)<br>iSPN$^{Atg7cKO}$ (24;4): 9.409 (0.3485) | Two-tailed, unpaired t test | $T_{43}$ = 1.081 | 0.2859 |

[*]n is the number of cells, N is the number of mice.

[$] Post hoc analysis: for two way ANOVA, we used Bonferroni post-hoc test. For one way ANOVA, we used the Holm-Sidak posthoc test.

## Autophagy controls iSPN intrinsic excitability

Despite the absence of synaptic deficits in iSPNs lacking Atg7, iSPN[Atg7cKO] mice demonstrated loco-motor hyperactivity and deficits in motor learning (*Figure 2*). How is iSPN function affected by loss of Atg7? SPN activity is governed by the interplay of synaptic excitation and inhibition with a low intrinsic excitability (*Kreitzer, 2009*; *Wilson and Kawaguchi, 1996*). We, therefore, hypothesized that iSPNs lacking Atg7 may be dysfunctional due to a change in their intrinsic excitability. In whole cell recordings, iSPNs from iSPN[Atg7cKO] mice exhibited a depolarized RMP, increased input resistance, and decreased rheobase, suggesting that autophagy is required for the normally low intrinsic excitability of iSPNs (*Figure 4A–D*). These effects occurred in the absence of a change in membrane capacitance (*Figure 4E*). The combination of increased input resistance, decreased rheobase and depolarized RMP led to a left-shifted current-response curve, suggesting that iSPNs were hyperresponsive to depolarizing inputs (*Figure 4F*).

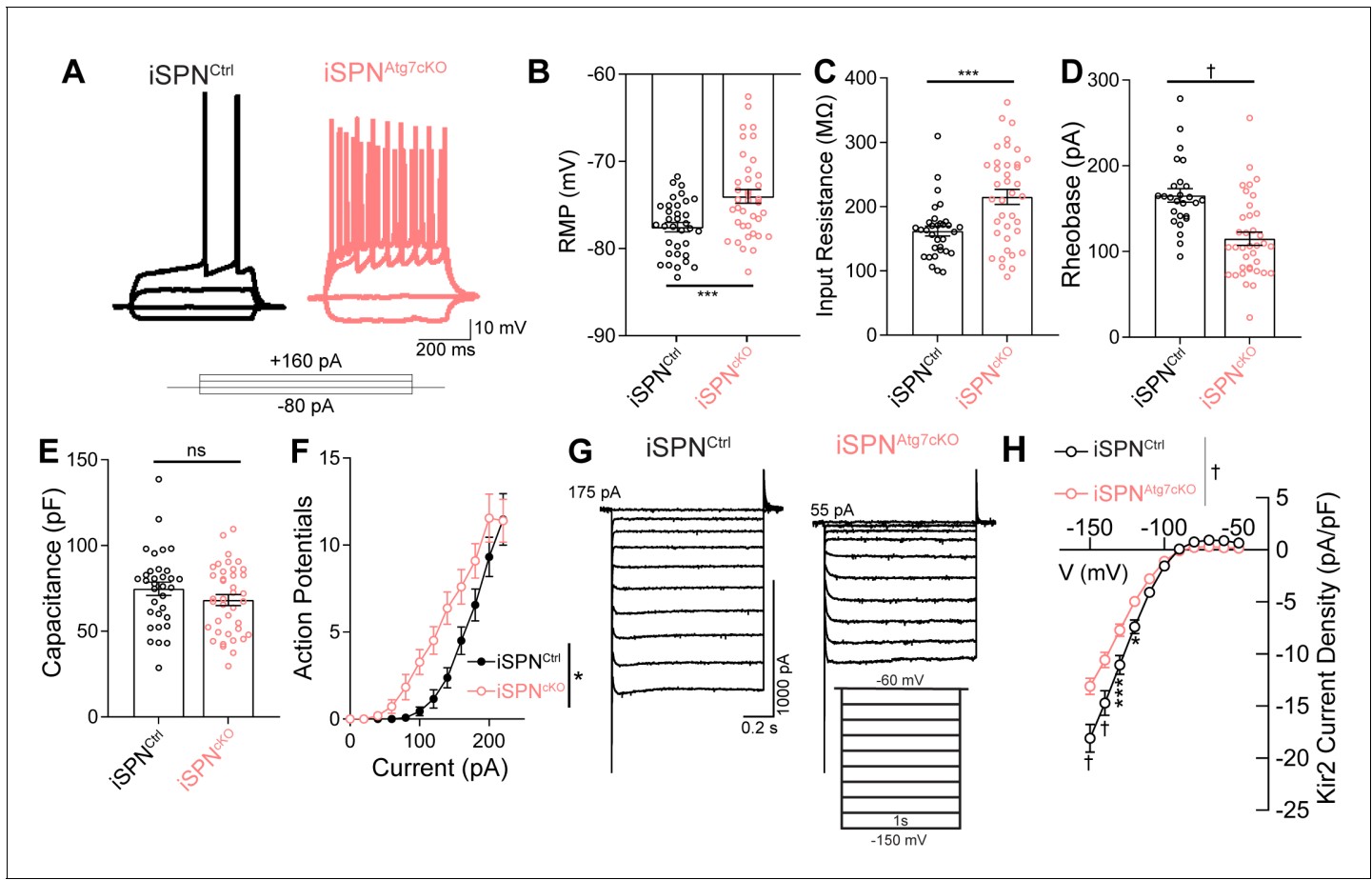

**Figure 4.** Loss of Atg7 leads to intrinsic hyperexcitability due to reduced Kir2 currents in iSPNs. (**A**) Representative current clamp traces in iSPNs from control or iSPN[Atg7cKO] mice. (**B-F**) iSPNs lacking Atg7 display (**B**) depolarized resting membrane potential ($t_{70}$=3.617, p = 0.0006; iSPN[Ctrl]: n=34 (8), iSPN[Atg7cKO]: n=38 (9)), (**C**) elevated input resistance ($t_{68}$=3.630, p = 0.0005; iSPN[Ctrl]: n=32 (8), iSPN[Atg7cKO]: n=38 (9)), (**D**) decreased rheobase ($t_{61}$=4.456, p<0.0001; iSPN[Ctrl]: n=26 (7), iSPN[Atg7cKO]: n=37(9)), (**E**) no change in capacitance ($t_{67}$ = 0.8096, p = 0.4210; iSPN[Ctrl]: n=33 (8), iSPN[Atg7cKO]: n=39 (9)) and (**F**) a left-shifted current-response curve [APs/500 msec (Current x Genotype: $F_{(13,772)}$ = 1.538, p = 0.0983; Current $F_{(13,772)}$=421.5, p<0.0001; Genotype: $F_{(1,772)}$ = 6.586, p = 0.0141); iSPN[Ctrl]: n=25(7), iSPN[Atg7cKO]: n=36(8)]. Data in (**B-F**) analyzed with a two-tailed, unpaired t test. Data in (**F**) analyzed with two-way repeated measures ANOVA. (**G**) Representative voltage clamp recordings of Kir2 currents. (**H**) iSPNs have lower Kir2 current density in iSPN[Atg7cKO] mice compared to iSPNs in iSPN[Ctrl] mice. iSPN[Ctrl]: n = 17(5), iSPN[Atg7cKO]: n=25(6). Data analyzed with a two-way repeated measures ANOVA, followed by Bonferooni post-hoc test. Voltage x Genotype: $F_{(10,400)}$ = 13.39, p<0.0001. ns p>0.05, * p<0.05, *** p<0.001, † p<0.0001. The online version of this article includes the following figure supplement(s) for figure 4:

**Figure supplement 1.** Atg7 is required for maximal Kir2 current but does not affect Kir2 voltage dependence.

We next asked whether Atg7 also regulated intrinsic excitability in dSPNs. While neither the RMP nor the rheobase of dSPNs in dSPN[Atg7cKO] were different from controls, dSPNs lacking Atg7 had an elevated input resistance and decreased membrane capacitance (*Table 3*). These results are consistent with change in the passive membrane properties of dSPNs caused by reduced dendritic length and complexity (*Figure 3*), in contrast to a prim ary effect on intrinsic excitability (see Discussion).

The change in intrinsic excitability in iSPNs from iSPN[Atg7cKO] may arise as a response to differences in the striatal network activity, in which case, dSPNs from iSPN[Atg7cKO] mice (which have autophagy) may also be affected, or they could be cell-intrinsic, in which case dSPNs in these mice would not be affected. We recorded from dSPNs in iSPN[Atg7cKO] mice and found no difference in the RMP, rheobase, input resistance or capacitance in dSPNs from iSPN[Atg7cKO] mice compared to control dSPNs, suggesting that Atg7 regulates intrinsic excitability in iSPNs through a cell-intrinsic mechanism (*Table 3*).

These results suggest that Atg7 may directly regulate intrinsic excitability in iSPNs but not dSPNs and provide a new mechanism through which autophagy can control neuronal function.

## Kir2 currents are decreased in the absence of autophagy

Although either reduced dendritic complexity (*Gertler et al., 2008*; *Mainen and Sejnowski, 1996*) or changes in specific ionic conductances can lead to intrinsic hyperexcitability following somatic current injection, we did not observe a change in dendritic arborization in iSPNs lacking autophagy (*Figure 3*), suggesting that a change in ion channel function underlies this phenotype. Inwardly rectifying potassium currents, mediated by Kir2.1 and Kir2.3 channels (referred to as Kir2 currents) (*Cazorla et al., 2012*; *Karschin et al., 1996*; *Shen et al., 2007*), are the predominant potassium conductance active around the resting membrane potential of SPNs and are the critical determinants of RMP, input and input resistance (*Nisenbaum et al., 1994*; *Wilson and Kawaguchi, 1996*). As iSPNs lacking autophagy exhibit a depolarized RMP and elevated input resistance, we hypothesized that autophagy may regulate Kir2 function.

Kir2 currents are measured as an inwardly rectifying, barium-sensitive current triggered by hyperpolarizing current steps (*Figure 4—figure supplement 1A–F*). We found a decrease in the whole cell Kir2 current density in iSPNs from iSPN[Atg7cKO] mice compared to control iSPNs (*Figure 4G–H*). This was not associated with a change in the voltage dependence of the Kir2 current (*Figure 4—figure supplement 1G–I*). Interestingly, there was no difference in Kir2 current density in dSPNs from dSPN[Atg7cKO] or iSPN[Atg7cKO] mice compared to controls (*Table 3*). These results suggest that autophagy is required for Kir2 currents in iSPNs but not dSPNs and further support the conclusion that autophagy controls neuronal function via cell-type-specific pathways.

## Autophagy controls Kir2 degradation

Because autophagy has not been described to regulate neuronal intrinsic excitability or potassium channel function, we further investigated the relationship between Atg7 and Kir2 channels.

To test if loss of Atg7 leads to reduced Kir2 current via downregulation of Kir2.1 or Kir2.3 mRNA levels, we coupled immunofluorescence and RNAScope analysis to measure the abundance of Kir2.1 and Kir2.3 mRNA on a single-cell level in DARPP32+ (to label all SPNs) and p62+ cells. Loss of Atg7 did not affect the number of Kir2.1 or Kir2.3 RNAScope puncta per cell in iSPN[Atg7cKO] mice compared to controls (*Figure 5A–C*). These results suggest that decreased Kir2 current in the absence of Atg7 does not result from a change in Kir2.1 or Kir2.3 mRNA expression.

Instead of regulating Kir2 channels at the transcriptional level, Atg7 could affect Kir2 protein stability, which would explain reduced Kir2 current. We generated total striatal lysates from control, dSPN[Atg7cKO], or iSPN[Atg7cKO] mice and confirmed that loss of Atg7 led to an increase in p62 levels in

**Table 3.** Intrinsic excitability in dSPNs from control, dSPNAtg7cKO, and iSPNAtg7cKO mice.

| Genotype | RMP (mV) | $R_{in}$ (MΩ) | Rheobase (pA) | Capacitance (pF) | Kir2 current density (pA/pF) |
|---|---|---|---|---|---|
| dSPN[Control] | −77.39 ± 0.8508 (24) | 145.7 8.110 (24) | 175 6.285 (24) | 80.28 ± 4.808 (22) | −16.59 ± 0.8121 (20) |
| dSPN[Atg7cKO] | −77.75 ± 0.9328 (30) | 189.3 12.52 (25) ** | 156.1 9.409 (26) | 62.22 ± 4.766 (27) * | −17.05 ± 0.8715 (22) |
| dSPNs from iSPN[Atg7cKO] | −78.40 ± 0.7481 (16) | 129.1 8.452 (14) | 177.8 8.685 (14) | 81.58 ± 4.728 (14) | −15.20 ± 1.675 (14) |

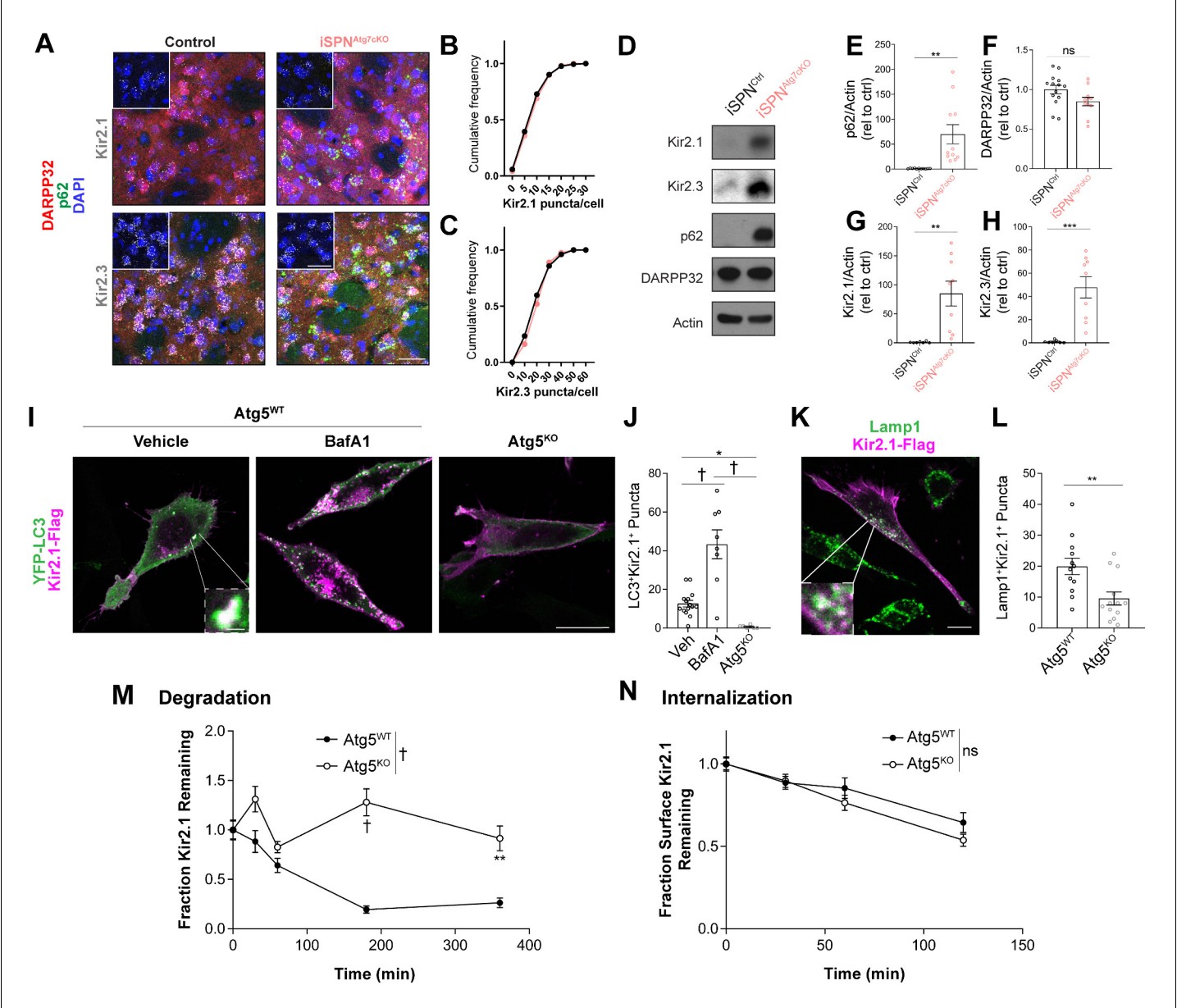

**Figure 5.** Autophagy is required for Kir2 degradation. (**A–C**) There was no difference in Kir2.1 or Kir2.3 mRNA expression in SPNs of iSPN$^{Atg7cKO}$ mice in an RNAscope assay (N = 77–173 cells from three mice per group). Inset shows just RNAscope and DAPI. Scale bar 30 µm. Data analyzed by the Kolmogorov-Smirnov test, Genotype effect: p>0.05 for Kir2.1 and Kir2.3. (**D**) Representative western blots of specified proteins from total striatal lysates from iSPN$^{Ctrl}$ or iSPN$^{Atg7cKO}$ mice. Quantifications of (**E**) p62 ($t_{20}$ = 3.551, p = 0.0020), (**F**) DARPP32 ($t_{23}$ = 1.984, p = 0.0593), (**G**) Kir2.1 ($t_{14}$ = 3.435, p = 0.0040), and (**H**) Kir2.3 ($t_{14}$ = 4.492, p = 0.0005) relative to actin. p62: iSPN$^{ctrl}$: n = 11, iSPN$^{Atg7cKO}$: n = 11. DARPP32: iSPN$^{ctrl}$: n = 14, iSPN$^{Atg7cKO}$: n = 11. Kir2.1: iSPN$^{ctrl}$: n = 7, iSPN$^{Atg7cKO}$: n = 9. Kir2.3: iSPN$^{ctrl}$: n = 7, iSPN$^{Atg7cKO}$: n = 9. Data analyzed by two-tailed, unpaired t test. (**I-J**) Kir2.1 is localized in LC3-GFP+ puncta in Atg5$^{WT}$ but not Atg5$^{KO}$ MEFs. BafilomycinA1 (BafA1; 100 nM 2 hr) treatment increases the number of LC3/Kir2.1-double labeled puncta in Atg5$^{WT}$ MEFs. Scale bar 20 µm. Inset scale bar 1 µm. Analyzed by one-way ANOVA followed by Bonferroni post-hoc test. $F_{(2,26)}$=25.64, p<0.0001. (**K-L**) A reduction of Lamp1$^+$Kir2.1$^+$ puncta in Atg5$^{KO}$ MEFs. Scale bar 20 µm, inset scale bar 1 µm. Data analyzed by two-tailed, unpaired t test. $t_{23}$ = 3.083, p = 0.0053. (**M**) Reduced degradation of SNAP-tag labeled Kir2.1 in Atg5$^{KO}$ MEFs. N: (WT,KO): T = 0 min (35,30), T = 30 min (30,27), T = 60 min (25,38), T = 180 min (33,67), T = 360 min (40,20). Data analyzed by two-way ANOVA. Genotype x time: $F_{4,335}$ = 7.880, p<0.0001. (**N**) No significant difference in the internalization of antibody-labeled surface Kir2.1 channels in Atg5$^{KO}$ MEFs. N: (WT,KO): T = 0 min (76,56), T = 30 min (52,37), T = 60 min (41,42), T = 120 min (28,38). Data analyzed by two-way ANOVA. Genotype x time: $F_{3,362}$ = 0.8038, ns; Time: $F_{(3,362)}$ = 25.88, p = 0.0001; Genotype: $F_{(1,362)}$ = 1.877, ns. ns p>0.05, *p<0.05, **p<0.01, ***p<0.001, † p<0.0001.

The online version of this article includes the following source data and figure supplement(s) for figure 5:

**Source data 1.** Source data for *Figure 5*.

*Figure 5 continued on next page*

Figure 5 continued

**Figure supplement 1.** Loss of autophagy does not affect Kir2.1 expression in dSPNAtg7cKO mice.
**Figure supplement 1—source data 1.** Source data for *Figure 5—figure supplement 1*.
**Figure supplement 1—source data 2.** Source Data for *Figure 5—figure supplement 1* continued.
**Figure supplement 2.** Higher steady-state levels of Kir2.1 and reduced degradation of Kir2.1 in Atg5[KO] MEFs.
**Figure supplement 2—source data 1.** Source Data for *Figure 5—figure supplement 2B*.
**Figure supplement 2—source data 2.** Source Data for *Figure 5—figure supplement 2D*.
**Figure supplement 3.** Kir2.1 degradation is disrupted in Atg7[KO] MEFs.
**Figure supplement 3—source data 1.** Source Data for *Figure 5—figure supplement 3A*.

both dSPN[Atg7cKO] and iSPN[Atg7cKO] mice compared to controls (*Figure 5D–E* and *Figure 5—figure supplement 1F*).

Furthermore, we validated our striatal dissections by blotting for DARPP32, an SPN marker, and found no difference across genotype (*Figure 5D,F* and *Figure 5—figure supplement 1*). Remarkably, however, total striatal levels of Kir2.1 and Kir2.3 were elevated in iSPN[Atg7cKO] mice compared to controls (*Figure 5D,G–H*), while levels of Kir2.1 and Kir2.3 in dSPN[Atg7cKO] mice were not different from controls (*Figure 5—figure supplement 1A–C*; note y-axis difference between *Figure 5D,G–H* and *Figure 5—figure supplement 1A–C*). These data suggest that Atg7 oppositely regulates Kir2 channel abundance and Kir2 current, specifically in iSPNs.

Although autophagy has previously been shown to regulate integral plasma membrane proteins by controlling membrane protein trafficking, interaction with partners, or degradation (*Gao et al., 2018*; *Rowland, 2006*; *Shehata et al., 2012*; *Sumitomo et al., 2018*; *Yan et al., 2018*), the unexpected inverse relationship between total Kir2 protein and Kir2 current in iSPNs lacking Atg7 warranted further validation. To address this, we virally overexpressed Kir2.1 in dSPNs or iSPNs from control, dSPN[Atg7cKO] or iSPN[Atg7cKO] mice. We hypothesized that if reduced Kir2 currents were driven by reduced expression of Kir2 channels, viral overexpression would normalize Kir2 currents in cells lacking Atg7. If, however, Atg7 controlled Kir2 protein function, viral overexpression of Kir2.1 would not normalize Kir2 currents in cells lacking Atg7.

We injected iSPN[Atg7cKO], dSPN[Atg7cKO] and control (A2Acre[+] Atg7[WT] or D1cre[+]Atg7[WT]) mice unilaterally with an adeno-associated virus (AAV) to drive Cre-dependent overexpression of either Kir2.1 and the fluorescent protein, ZsGreen (*Rothwell et al., 2014*) or an AAV expressing mCherry as a control on the contralateral side (*Figure 6A*). After 3–4 weeks, acute brain slices were prepared and Kir2 current density measured using whole-cell voltage clamp recordings. Kir2 current density was higher in dSPNs overexpressing Kir2.1 than mCherry in Cre[Drd1aey262] and dSPN[Atg7cKO] mice (*Figure 6B*). Similarly, Kir2.1 overexpression increased Kir2 current density in iSPNs from Cre[Adora2aK-G139]Atg7[wt] mice (*Figure 5—figure supplement 2C*). In contrast, Kir2.1 overexpression in SPNs from Cre[Adora2aKG139]Atg7[Fl/Fl] (iSPN[Atg7cKO]) mice did not display a higher Kir2 current density than mCherry expressing SPNs (*Figure 6C*), confirming that Atg7 is required for overexpressed Kir2.1 channels to be functional. These results further highlight the cell-type specificity of the requirement of Atg7 for Kir2.1 current.

We confirmed that the effect of loss of Atg7 on Kir2 current could be complemented in adulthood by constructing a virus to Cre-dependently express Atg7 in iSPNs (*Figure 6D*). Viral reexpression of Atg7, but not a fluorophore-only control, decreased p62 accumulation in transduced cells (*Figure 6D*). Furthermore, reexpression of Atg7 corrected SPN hyperexcitability and increased Kir2 currents in iSPN[Atg7cKO] cells but had no effect in control iSPNs from iSPN[ctrl] mice (*Figure 6D–J*). These results demonstrate that reduced Kir2 current in iSPN[Atg7cKO] mice could be complemented by Atg7 reexpression, but not Kir2.1 overexpression, in adulthood.

We next examined why Kir2 currents were decreased but Kir2 protein was increased, in the absence of Atg7, to define the mechanism through which autophagy may regulate neuronal intrinsic excitability. To do this, we expressed a Kir2.1 fusion with a C-terminal FLAG and SNAP-tag (Kir2.1-[FlagSNAP]) in a well-validated, and transfectable line of mouse embryonic fibroblasts (MEFs) lacking Atg5, a protein required for autophagosome biosynthesis (*Figure 5—figure supplement 2A*) (*Kuma et al., 2004*). We chose to use a well-validated transformed MEF cell line lacking Atg5 in contrast to primary striatal culture for two reasons: (1) the maturation of SPN intrinsic excitability occurs

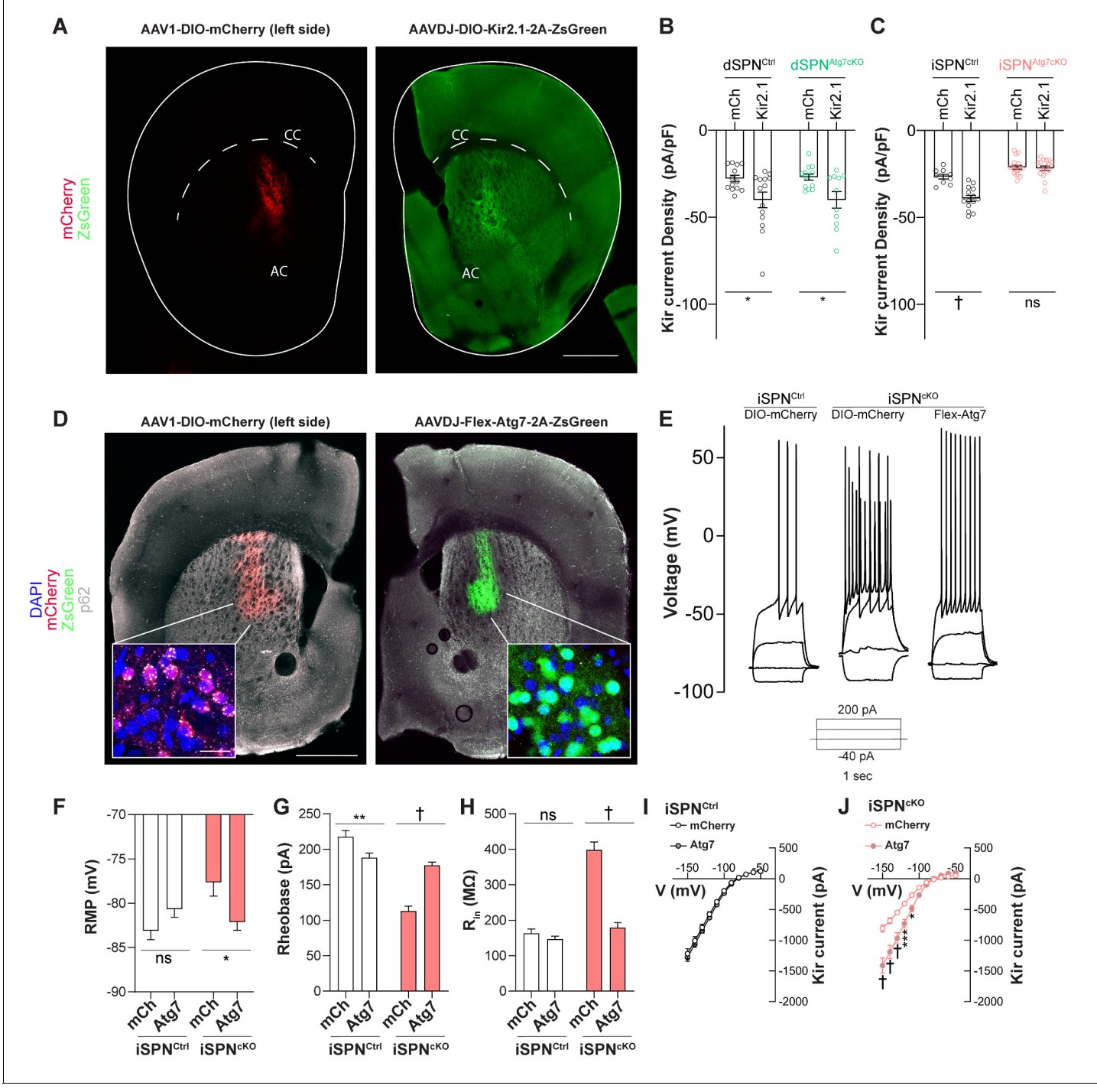

**Figure 6.** Atg7 reexpression but not Kir2.1 overexpression rescues changes in iSPN physiology. (**A**) Immunofluorescent images of striatal hemisections from mice injected with AAV1-DIO-mCherry or AAVDJ-DIO-Kir2.1-t2A-zsGreen stained against mCherry (left) or GFP (right). Scale bar 500 µm. (**B**) Significant effect of Kir2.1 overexpression on Kir2 current density in dSPNs from control and dSPNs from dSPN$^{Atg7cKO}$ mice. Data analyzed with two-way ANOVA. Genotype x virus: $F_{(1,47)}=0.01278$, p = 0.9105; Genotype: $F_{(1,47)}=0.01840$, p = 0.8937; Virus: $F_{(1,47)}$ = 13.23, p=0.0007. N = dSPN$^{Ctrl}$ (D1cre Atg7$^{wt/wt}$, mCh: n = 13(4), Kir2.1: n = 14(4)) or dSPN$^{Atg7cKO}$ (D1cre Atg7$^{Fl/Fl}$, mCh: n = 13(4), Kir2.1: n = 11 (3)). (**C**) Significant effect of Kir2.1 overexpression on Kir2 current density in iSPNs from control but not from iSPN$^{Atg7cKO}$ mice. Data analyzed with two-way ANOVA. Genotype x virus: $F_{(1,54)}$ = 17.62, p = 0.0001. N = iSPN$^{Ctrl}$ (A2Acre Atg7$^{wt/wt}$, mCh: n = 9 (3), Kir2.1: n = 14(3)) or iSPN$^{Atg7cKO}$ (A2Acre Atg7$^{Fl/Fl}$, mCh: n = 18(5), Kir2.1: n = 17(4)). (**D**) Representative hemisections from iSPN$^{Atg7cKO}$ mice injected with AAV1-DIO-mCherry (left) or AAVDJ-Flex-Atg7-2A-ZsGreen (right) showing reduction in p62 puncta in cells expressing Atg7 (right) compared to mCherry (left). Scale Bar 500 µm. Inset, scale bar 20 µm. (**E**) Representative current clamp traces from mCherry-expressing iSPN in a control (A2Acre) mouse, mCherry expressing iSPN in an iSPN$^{Atg7cKO}$ mouse or

*Figure 6 continued on next page*

Figure 6 continued

an Atg7-expressing iSPN from an iSPN$^{Atg7cKO}$ mouse. (F-J) Reexpression of Atg7 normalizes RMP [N = iSPN$^{Ctrl}$: mCh: n = 25(5), Kir2.1: n = 26(5)) or iSPN$^{Atg7cKO}$: mCh: n = 13(3), Kir2.1: n = 15(3). Genotype x virus: F$_{(1,76)}$=9.491, p = 0.0029], rheobase [N = iSPN$^{Ctrl}$: mCh: n = 25(5), Kir2.1: n = 26(5)) or iSPN$^{Atg7cKO}$: mCh: n = 14(3), Kir2.1: n = 15(3). Genotype x virus: F$_{(1,75)}$=36.54, p<0.0001], R$_{in}$ [N = iSPN$^{Ctrl}$: mCh: n = 25(5), Kir2.1: n = 26(5)) or iSPN$^{Atg7cKO}$: mCh: n = 13(3), Kir2.1: n = 16(3). Genotype x virus: F$_{(1,76)}$=55.55, p<0.0001, and Kir2 current in iSPNs from iSPN$^{Atg7cKO}$ mice compared to mCherry control but only reduces R$_{in}$ in controls. All data analyzed by two-way ANOVA. In (I), N = iSPN$^{Ctrl}$: mCh: n = 9 (5), Kir2.1: n = 9 (5). Voltage x virus: F$_{(10,160)}$=0.1170, p = 0.9996; Voltage: F$_{(10,160)}$ = 178.8, p<0.0001; Virus: F$_{(1,16)}$=0, p = 0.9929. In (J), N = iSPN$^{Atg7cKO}$: mCh: n = 10(3), Kir2.1: n = 12 (3). Genotype x virus: F$_{(10,200)}$ = 18.56, p<0.0001. ns p>0.05, *p<0.05, **p<0.01, ***p<0.001, † p<0.0001.

postnatally, suggesting that the intrinsic excitability of dissociated SPNs in culture may depend on processes distinct from those *in vivo* (*Kuo and Liu, 2019*; *Lieberman et al., 2018*); (2) almost all SPNs in primary culture express markers of dSPNs (*Falk et al., 2006*), and this population does not display Kir2 regulation by autophagy *in vivo*.

Transformed MEFs lacking Atg5 (Atg5$^{KO}$ MEFs) transiently transfected with Kir2.1$^{FlagSNAP}$ demonstrated a higher steady-state level of Kir2.1 expression but reduced LC3B-ii/LC3B-i, a measurement of autophagic activity (*Figure 5—figure supplement 3B-C*; *Kabeya et al., 2000*). Immunofluorescence analysis demonstrated colocalization of Kir2.1$^{FlagSNAP}$ with YFP-LC3 puncta, a marker of autophagosomes, in MEFs (*Figure 5I–J*). The colocalization of Kir2.1$^{FlagSNAP}$ with YFP-LC3 increased following incubation with bafilomycin to collapse the lysosomal pH gradient and block autophagosome fusion with lysosomes (*Yamamoto et al., 1998*) (*Figure 5I–J*). We observed that the Kir2.1-$^{FlagSNAP}$ colocalization with YFP-LC3 was dependent on Atg5 (*Figure 5I–J*). Furthermore, in the absence of Atg5, we found a reduced number of Kir2.1$^+$Lamp1$^+$ late endosomes/lysosomes (*Figure 5K–L*), supporting the conclusion that autophagy plays a role in the lysosomal degradation of Kir2.1.

To address whether Kir2.1 degradation itself was dependent on autophagy, we pulse-labeled transfected Atg5$^{WT}$ and Atg5$^{KO}$ MEFs with SNAPcell ligand and fixed cells following a chase interval (*Figure 5M*). We found that the half-life of labeled Kir2.1$^{FlagSNAP}$ was greatly extended in Atg5$^{KO}$ compared to Atg5$^{WT}$ MEFs (*Figure 5M*). We further confirmed that the Kir2.1$^{FlagSNAP}$ half-life was extended in Atg5$^{KO}$ MEFs using a cycloheximide pulse-chase assay (*Figure 5—figure supplement 2D–E*). This demonstrates that autophagy is required for Kir2.1 degradation.

Thus, multiple independent approaches indicate that autophagy is required for Kir2.1 degradation. Previous reports have suggested that Kir2.1 is endocytosed and trafficked to the lysosome for degradation in an ESCRT-dependent manner (*Jansen et al., 2008*; *Kolb et al., 2014*). Autophagy could contribute to lysosomal degradation of Kir2.1 by affecting internalization and endocytosis, or the transport of endocytosed Kir2.1 to the lysosome. To address this, we engineered a hemagglutinin (HA) tag (Kir2.1$^{extHA-FlagSNAP}$) into an extracellular loop of Kir2.1, at a site which has previously been used to tag the extracellular face of Kir2.1 without affecting channel function (*Chen et al., 2002*). We labeled live Atg5$^{WT}$ or Atg5$^{KO}$ MEFs transfected with Kir2.1$^{extHA-FlagSNAP}$ with anti-HA antibodies and measured the remaining surface resident-labeled channels after a chase period. We found no difference in the internalization of Kir2.1$^{extHA-FlagSNAP}$ in the absence of autophagy (*Figure 4N*), suggesting that autophagy is required for Kir2.1 degradation at a post-endocytic step.

We generated primary Atg7$^{KO}$ MEFs to confirm that Atg7, in addition to Atg5, was required for Kir2.1 degradation (*Figure 5—figure supplement 3A*). We pulse labeled primary Atg7$^{WT}$ and Atg7$^{KO}$ MEFs transfected with Kir2.1$^{FlagSNAP}$ and found an elevated level of SNAPcell-labeled Kir2.1-$^{FlagSNAP}$ after a 120 min chase in Atg7$^{KO}$ MEFs compared to controls (*Figure 5—figure supplement 3A–C*).

We conclude that autophagy is required for the lysosomal delivery and degradation of endocytosed Kir2.1 channels.

## Lack of autophagy leads to increased surface levels of Kir2 channels despite reduced channel function

If autophagy is required for Kir2.1 degradation, but Kir2 channel abundance is elevated in the absence of autophagy, why are Kir2 currents decreased in cells lacking autophagy? One possibility is that Kir2 channels could be mislocalized in the absence of autophagy and are absent from the plasma membrane.

To address whether Kir2.1 function is regulated by autophagy in MEFs, we measured Kir2 currents using whole-cell patch clamp recordings in Kir2.1$^{FlagSNAP}$ transfected MEFs. We detected a Ba$^{2+}$-sensitive, inwardly rectifying current that was absent from untransfected cells (*Figure 7A*). Despite the increase in total Kir2.1$^{FlagSNAP}$ levels in Atg5$^{KO}$ MEFs (*Figure 5—figure supplement 3B–C*), the Kir2 current in Atg5$^{KO}$ MEFs was decreased relative to Atg5$^{WT}$ MEFs (*Figure 7B*). This

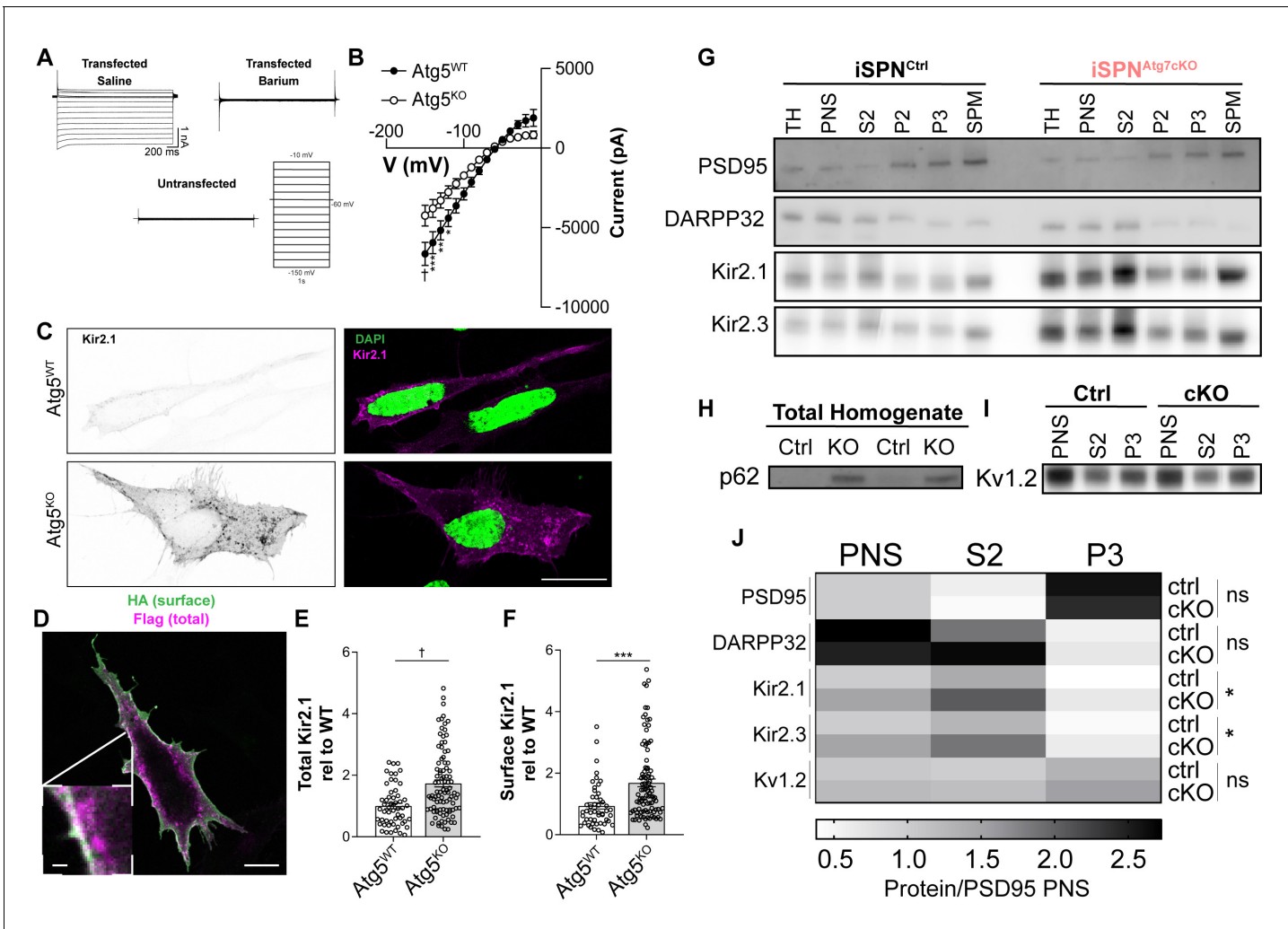

**Figure 7.** Autophagy is not required for Kir2 trafficking. (**A**) Sample traces from MEFs transfected with Kir2.1 demonstrate an inwardly rectifying, barium sensitive current that is absent in untransfected cells. (**B**) Atg5$^{KO}$ MEFs have reduced Kir2.1 current compared to Atg5$^{WT}$ MEFs. WT: N = 8, KO: N = 12. Voltage step x genotype: $F_{(14,\ 252)}=8.985$, p<0.0001. (**C**) Representative micrographs of Atg5$^{WT}$ and Atg5$^{KO}$ MEFs. On right, the Kir2.1 channel in the WT image has been contrasted to compare staining pattern. Scale bar 20 µm. (**D-F**) Elevated levels of both surface and total levels of Kir2.1 in Atg5$^{KO}$ MEFs. Scale bar 20 µm, inset scale bar 1 µm. (**E**) $t_{147}=4.511$, p<0.0001. (**F**) $t_{147}=4.511$, p<0.0001. (**G-J**) Subcellular fractionation reveals elevated levels of Kir2.1 and Kir2.3 in all fractions of iSPN$^{Atg7cKO}$ mice compared to controls but no change in the relative distribution of Kir2.1 or Kir2.3 between genotypes. (**H**) p62 is elevated in the total homogenate of iSPN$^{Atg7KO}$ mice. (**I**) No change in distribution or level of Kv1.2 between genotypes. iSPN$^{Ctrl}$: N = 5. iSPN$^{Atg7cKO}$: N = 3. TH, total homogenate; PNS, post-nuclear supernatant; S2, 20,000xg supernatant; P2, 20,000xg pellet; P3, resuspended P2 spun at 100,000xg; SPM, synaptic plasma membranes isolated from 1.0M and 1.2M sucrose interface. Data were analyzed by two-way ANOVA for each analyzed protein with fraction and genotype as factors. No significant interaction between fraction and genotype was found for any protein. Fraction was significant for each protein. Genotype was only significant for Kir2.1 and Kir2.3. Kir2.1: Genotype: $F_{(1,6)}=9.373$, p = 0.0222. Kir2.3: Genotype: $F_{(1,6)}=6.615$, p = 0.0422. See *Table 1* for detailed statistics. ns p>0.05, *p<0.05, **p<0.01, ***p<0.001, † p<0.0001. Experiments in B, E, and F were combined from at least three independent experiments.

The online version of this article includes the following source data and figure supplement(s) for figure 7:

**Source data 1.** Source Data for *Figure 7*.
**Figure supplement 1.** Plasma membrane floatation and MBCD treatment.

phenocopies the situation in the iSPN[Atg7cKO] striatum and suggests that the mechanism through which Kir2.1 is regulated by autophagy can be explored in this heterologous system.

Autophagy may be required for both the localization and the degradation of Kir2.1, which would explain why, in the absence of autophagy, Kir2.1 levels are elevated but Kir2.1 current is decreased. We found, however, that Kir2.1[FlagSNAP] was located on the surface and in intracellular vesicles in both control and Atg5[KO] MEFs (*Figure 7C*). In addition, similar patterns of Kir2.1 localization were observed in Atg7[WT] and Atg7[KO] primary MEFs (*Figure 5—figure supplement 3D*).

Next, we measured the steady-state amount of surface-resident Kir2.1 by fixing MEFs transfected with Kir2.1[extHA-FlagSNAP] and staining for the HA tag without permeabilization. Subsequent permeabilization and staining for the FLAG tag permitted quantification of the total Kir2.1 protein expressed per cell. We found that both surface and total levels of Kir2.1 were elevated in Atg5[KO] MEFs relative to controls (*Figure 7D–F*), further emphasizing that the reduced Kir2 currents in the absence of autophagy do not arise from fewer surface-resident channels.

We confirmed that Kir2.1 and Kir2.3 were also similarly localized *in vivo* in iSPN[Atg7cKO] striatum compared to control using subcellular fractionation. Kir2.1 and Kir2.3 are localized to the somato-dendritic region and to dendritic spines in SPNs (*Shen et al., 2007*). We isolated striatal synaptosomes from iSPN[Ctrl] or iSPN[Atg7cKO] mice using a sucrose gradient approach (*Bermejo et al., 2014*). To confirm that we had enriched for synaptic plasma membranes, we blotted for the cytosolic protein DARPP32 or the postsynaptic density scaffolding protein PSD95. We found that there was no difference in the levels of PSD95 in total striatal lysates in iSPN[Atg7cKO] mice compared to controls (*Figure 5—figure supplement 1H–J*) and that we were able to enrich PSD95 and deplete DARPP32 in the synaptic plasma membrane (SPM) fractions in both genotypes (*Figure 7G,J*). Kir2.1 and Kir2.3 were enriched, compared to the postnuclear supernatant (PNS), in both the S2 fraction, containing cytosol and light membranes, and SPM/S3 fraction in both iSPN[Ctrl] and iSPN[Atg7cKO] striatum (*Figure 7G*). Interestingly, although there was no relative difference in the distribution of Kir2.1 and Kir2.3, there were higher levels of both Kir2.1 and Kir2.3 in all fractions in iSPN[Atg7cKO] compared to iSPN[Ctrl] striatum (*Figure 7G,J*). These data support the conclusion that although total and synaptic Kir2.1 and Kir2.3 levels are elevated, their distribution is unaffected by the absence of Atg7 *in vivo*.

To address whether loss of Atg7 in iSPNs disrupts degradation or localization of other potassium channels in iSPNs, we examined the localization of another potassium channel expressed by dSPNs and iSPNs in the conditional knockout lines. We found that there was no change in the localization or total level of $K_v1.2$, a voltage-gated potassium channel expressed by SPNs (*Figure 7I–J* and *Figure 5—figure supplement 1H–I*) (*Shen et al., 2004*). This suggests that loss of autophagy in iSPNs does not lead to global disruption of ion channel degradation.

These results were further confirmed by floating striatal membrane pellets in a discontinuous iodixanol gradient, which separates plasma membranes (light fraction) from other organelles (heavy fractions). Kir2 channels classically exhibit a bimodal distribution between heavy and light membrane fractions as a population of these channels exist in cholesterol-rich membrane domains (*Tikku et al., 2007*). No difference in the distribution of endogenous Kir2.1 or $K_V1.2$ was observed between iSPN[Ctrl] and iSPN[Atg7cKO] striatum using this gradient approach (*Figure 7—figure supplement 1A–D*). We confirmed that Kir2 channels were not trapped in cholesterol-rich membranes, where they are inactive (*Romanenko et al., 2004*), by treating acute brain slices from iSPN[Atg7cKO] mice with methyl-β-cyclodextrin (MβCD), to remove cholesterol, and found that Kir2 currents were not increased iSPNs relative to iSPNs from vehicle-treated slices (*Figure 7—figure supplement 1E*).

These results demonstrate that Kir2 channels are localized to the plasma membrane but are less functional in the absence of Atg7.

## Kir2 channels are acetylated and inactivated in the absence of autophagy

As Kir2 channels are localized to the plasma membrane in higher quantities in the absence of autophagy both in vivo and in MEFs (*Figure 7D–J*), it remained unclear why Kir2 currents were decreased. We hypothesized that these channels may be inhibited by a post-translational modification in the absence of autophagy. Several proteins that are selectively degraded by autophagy undergo lysine modification by ubiquitin or acetylation (*Khaminets et al., 2016*). Immunoprecipitation of Kir2.1[extHA-FlagSNAP] from Atg5[WT] or Atg5[KO] cells indicated that in the absence of autophagy

the level of Kir2.1 acetylation was markedly increased while its ubiqutination status was unaffected (*Figure 8A*).

Both of these modifications can signal for proteins to enter the autophagy pathway for degradation and we therefore hypothesized that the acetylated lysine residue was also required for autophagic degradation of Kir2.1, as this would explain why a modified channel accumulates in the absence of autophagy: acetylated Kir2.1 may be targeted for degradation but, in the absence of autophagy, cannot be eliminated.

To identify the acetylated lysine that may be responsible for inhibition of Kir2.1 and targeting it for autophagic degradation, we searched for lysines in Kir2.1 previously implicated in its degradation. Kir2.1 contains a motif in its C-terminus that regulates Kir2.1 surface residence and degradation (*Figure 8C*) (*Ambrosini et al., 2014*), is evolutionarily conserved, and is present in Kir2.3 (*Figure 8B*). Three lysines are present within this sequence, two of which, K338 and K346, are known

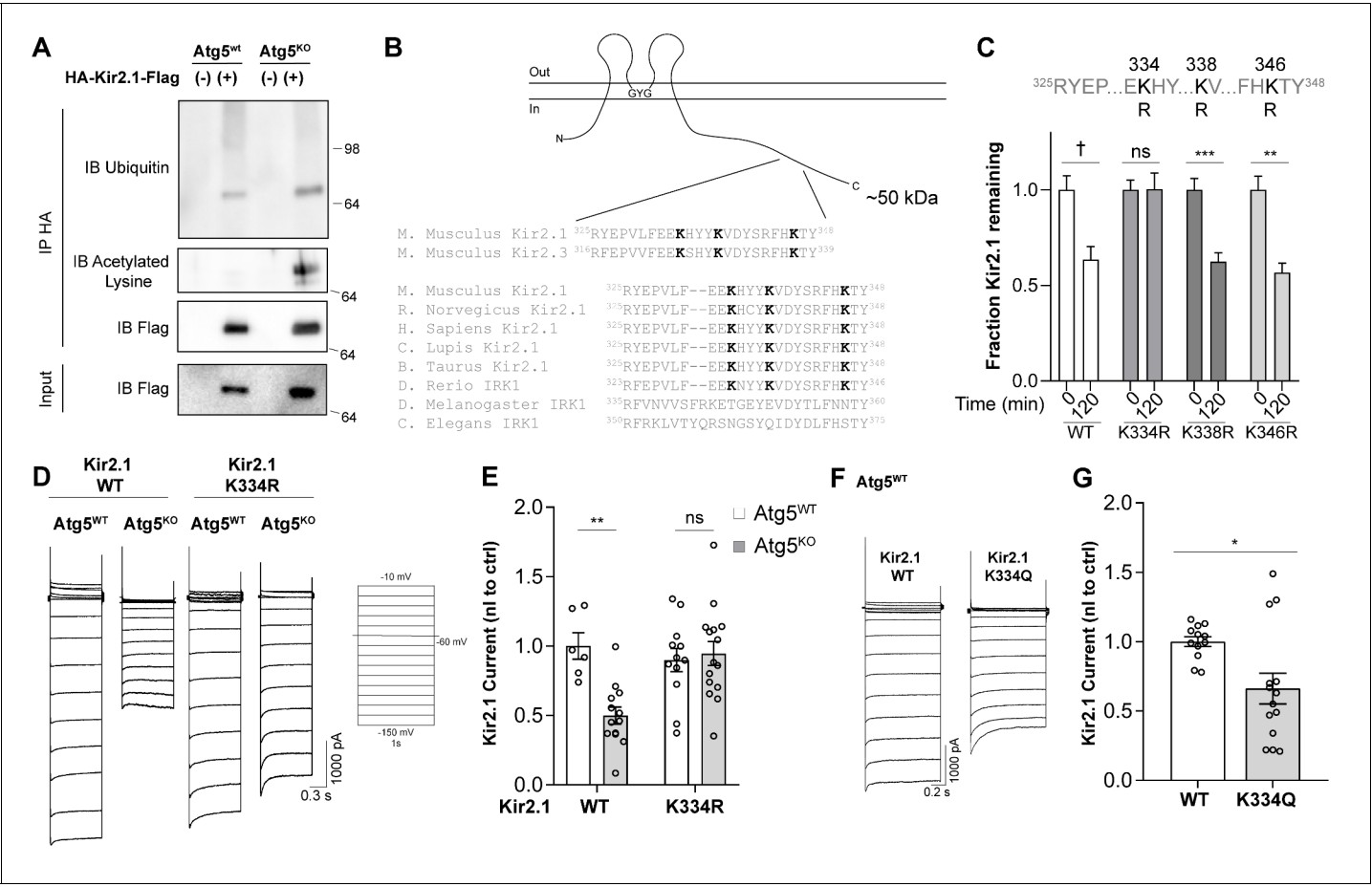

**Figure 8.** Hyperacetylation of Kir2.1 at K334 in the absence of autophagy inhibits channel activity. (**A**) Immunoprecipitation of Kir2.1 reveals elevated levels of acetylated lysines on Kir2.1 in Atg5KOMEFs without a change in ubiquitination. Representative blots shown from at least three independent replicates. (**B**) A conserved motif in the C-terminal tail of Kir2.1 and Kir2.3 contains three modifiable lysines and has previously been implicated in Kir2 channel degradation. (**C**) A degradation screen in which K334, K338 and K346 were mutated to the unmodifiable residue, arginine, reveals that K334 is required for Kir2.1 degradation in Atg5WT MEFs. N: (T=0min, T = 120 min) WT (52,79), K334R (27,51), K338R (40,41), K346R (23,11). Data analyzed by two-way ANOVA followed by Bonferroni post-hoc test. Kir genotype x time: $F_{(3,390)}$ = 3.211, p = 0.0230. (**D-E**) Kir2.1 K334R current is normalized in Atg5KO MEFs but does not affect Kir2.1 current in Atg5WT MEFs. Data analyzed by two-way ANOVA followed by Bonferroni post-hoc test. Cell genotype x Kir genotype: $F_{(1,42)}$ = 9.603, p = 0.0035. (**F-G**) Kir2.1 K334Q, with an acetylation-mimic at K334, has reduced current in Atg5WT MEFs. Voltage step protocol is the same as in (**D**). Data analyzed by two-tailed unpaired t test. $t_{24}$=2.707, p=0.0123. ns p>0.05, * p<0.05, ** p<0.01, *** p<0.001, † p<0.0001. Experiments in C, E and G were combined from at least three independent experiments.

The online version of this article includes the following source data and figure supplement(s) for figure 8:

**Source data 1.** Source Data for *Figure 8*.

**Figure supplement 1.** Reduced acetylation of Kir2.1 K334R in Atg5KO MEFs but acetylation status at K334 does not affect surface residence of Kir2.1.

to undergo ubiquitination (*Figure 8B*) (*Ambrosini et al., 2014*; *Wagner et al., 2012*). We mutated K334, K338 and K346 individually to arginines to prevent addition of ubiquitin or acetyl groups and screened these mutants in a degradation assay in Atg5$^{WT}$ cells. Kir2.1 K334R underwent significantly less degradation than Kir2.1 WT, K338R or K346R (*Figure 8C*), suggesting that K334 is required for degradation of Kir2.1.

We assessed whether the Kir2.1 K334R channel was less acetylated in Atg5$^{KO}$ MEFs that WT channel. We transfected Atg5$^{KO}$ MEFs with WT Kir2.1$^{extHA-FlagSNAP}$ or Kir2.1$^{extHA-FlagSNAP}$ K334R and immunoprecipitated lysates with an anti-HA antibody. Immunoprecipitates were then sequentially blotted for FLAG and acetyl-lysines and we found that lysine acetylation of the Kir2.1$^{extHA-FlagSNAP}$ K334R mutant was reduced by ~50% (*Figure 8—figure supplement 1A–B*), suggesting that other residues are acetylated in Kir2.1 but are not necessary for inhibition of the channel in Atg5$^{KO}$ cells.

We next examined whether the K334R mutant, which could not be modified, was also fully functional even in the absence of autophagy by transfecting Atg5$^{WT}$ and Atg5$^{KO}$ cells with Kir2.1 K334R. We found that the K334R mutation did not affect channel function in Atg5$^{WT}$ cells but significantly increased channel function in Atg5$^{KO}$ cells compared to Kir2.1 WT (*Figure 8D–E*), suggesting that modification of K334 was necessary for the inhibition of Kir2 function in the absence of autophagy.

We next engineered a mutant Kir2.1 channel that harbors an acetylation-mimic glutamine instead of K334. Kir2.1 K334Q had reduced function in Atg5$^{WT}$ cells compared to Kir2.1 WT (*Figure 8F–G*), demonstrating that acetylation of Kir2.1 at K334 is both necessary and sufficient to control channel function.

How may acetylation of Kir2.1 at K334 affect channel function? One possibility is that surface residence of the channel could be affected. To examine this possibility, we stained for surface-resident channels, as described above, in Atg5$^{WT}$ and Atg5$^{KO}$ cells expressing WT Kir2.1$^{extHA-FlagSNAP}$, Kir2.1-$^{extHA-FlagSNAP}$ K334R, and Kir2.1$^{extHA-FlagSNAP}$ K334Q. We found that WT, K334R and K334Q all had increased surface expression in Atg5$^{KO}$ MEFs compared to Atg5$^{WT}$ MEFs but there were no differences in surface residence between WT or mutant channels (*Figure 8—figure supplement 1C*). This suggests that acetylation of Kir2.1 affects channel activity on the surface but does not affect surface levels of the channel.

These data indicate that, in the absence of autophagy, Kir2 channels are present on the plasma membrane but are inhibited by acetylation at K334. We conclude that acetylation of K334 acts to promote the autophagy-dependent degradation of Kir2.1.

## Discussion

Although originally studied in the central nervous system in the context of neurodegenerative disease, autophagy has recently been identified as a key regulator of neurotransmission. Emphasizing the need to understand autophagic function at the synapse, autophagic dysfunction has been reported in neurodevelopmental disorders that are thought to arise from synaptic deficits. Here, we investigated the role of autophagy in the function of both classes of GABAergic projection neurons in the striatum, a region implicated in the pathophysiology of several neurodevelopmental disorders. We found that, in dSPNs, the required autophagy gene, Atg7, is required for dendritic structure and spine density and the function of excitatory inputs onto dSPNs. In iSPNs, we did not detect changes in dendritic arborization, spine density or excitatory and inhibitory inputs in the absence of Atg7. In contrast, however, loss of Atg7 in iSPNs led to intrinsic hyperexcitability, due to reduced Kir2 function. We then demonstrated that Kir2 channels are degraded by autophagy and have decreased activity in the absence of autophagy. Furthermore, loss of Atg7 in either dSPNs or iSPNs led to deficits in behavioral tasks considered to depend at least in part on the striatum (*Durieux et al., 2012*). These results demonstrate that autophagy contributes to striatal function via distinct molecular pathways in dSPNs and iSPNs.

### Autophagic control of synaptic function in dSPNs

Autophagy is required for normal synaptic structure and function in *C. elegans*, the *Drosophila* neuromuscular junction, and excitatory neurons in the mouse cortex and hippocampus (*Glatigny et al., 2019*; *Nikoletopoulou et al., 2017*; *Shehata et al., 2012*; *Shen and Ganetzky, 2009*; *Stavoe et al., 2016*; *Tang et al., 2014*; *Vanhauwaert et al., 2017*; *Yan et al., 2018*). We report that, in dSPNs, loss of autophagy led to reduced complexity of the dendritic tree,

lower dendritic spine density, and functional deficits in excitatory inputs. The mechanism through which autophagy may regulate dendritic structure and function in dSPNs remains unknown. Dendritic complexity and spine density can be controlled by neurotrophic factors (*Horch et al., 1999*; *Li et al., 2012*), mitochondrial function (*Li et al., 2004*), neuromodulatory signaling (*Villalba and Smith, 2013*), and synaptic inputs and neurotransmission (*Alvarez and Sabatini, 2007*; *Lambot et al., 2016*). Each of these cellular pathways is modulated by autophagy and therefore could lead to altered dSPN dendritic arborization and excitatory synaptic inputs (*Kononenko et al., 2017*; *Nikoletopoulou et al., 2017*; *Youle and Narendra, 2011*). A key unanswered question is whether these synaptic pathologies arise during early dSPN development or as a degenerative process and could be addressed via a longitudinal analysis of these parameters in dSPNAtg7cKO mice. We hypothesize that, given the dynamic changes in autophagic flux that occur in SPNs during early postnatal development (P10-P18; *Lieberman et al., 2019a*), autophagy may play a role in dendritic growth and excitatory synaptogenesis on dSPNs.

Although we did not observe a difference between the dendritic morphology of dSPNs and iSPNs, there are conflicting reports on morphological differences between dSPNs and iSPNs (*Cepeda et al., 2008*; *Gertler et al., 2008*; *Suárez et al., 2014*). Future studies must aim to reconcile these disparate findings.

Interestingly, in some neuronal populations, loss of autophagy leads to increased cell size and increased synaptic contacts (*Shen and Ganetzky, 2009*; *Tang et al., 2014*). Mechanistic studies aimed at defining how autophagy regulates synapse formation or maintenance would provide insight into how loss of the same cellular pathway could also yield reduced spine density in dSPNs.

Perhaps, most surprising is the fact that iSPNs that lack Atg7 do not show changes in dendritic complexity, spine density or excitatory inputs. One possible explanation would be a difference in the timing of Cre expression and Atg7 knockout between dSPN[Atg7cKO] and iSPN[Atg7cKO] mice. However, Cre-mediated recombination has been reported in both driver lines before postnatal day 7 (*Kozorovitskiy et al., 2015*; *Kozorovitskiy et al., 2012*), which is before excitatory synaptogenesis occurs in the striatum (*Hattori and McGeer, 1973*; *Tepper et al., 1998*). An additional confounding factor may be Cre-mediated recombination that occurs in the cortex of dSPN[Atg7cKO] mice, which could synergize with dSPN loss of Atg7 to affect synaptic inputs (data not shown). iSPN[Atg7cKO] mice do not demonstrate Cre-mediated recombination outside of the striatum (data not shown). Future efforts to define the underlying mechanisms that contribute to cell-type-specific roles for autophagy in the striatum could include Cre-driver lines in which both dSPNs and iSPNs are targeted to rule out a contribution of different expression time courses between the Cre lines used in this study.

Alternatively, excitatory synaptogenesis onto iSPNs may occur in an autophagy-independent manner. Once the mechanism through which autophagy regulates dendritic complexity and synaptogenesis in dSPNs is identified, future studies could compare this process with iSPNs and define why loss of autophagy in iSPNs does not affect iSPN synaptogenesis.

Atg7 is also involved in non-autophagic processes such as LC3-associated phagocytosis and secretion of lysosomal contents (*Subramani and Malhotra, 2013*). It is possible that Atg7 contributes to dSPN dendritic morphology and synaptic inputs via these processes as opposed to 'classical' autophagic degradation. Future studies comparing loss of other autophagy-associated genes that do not contribute to LC3-associated phagocytosis or lysosomal exocytosis may provide further insight.

## Autophagy regulates iSPN intrinsic excitability via Kir2 currents

In contrast to dSPNs, Atg7 was not required for excitatory or inhibitory transmission onto iSPNs. Rather, whole cell recordings revealed an intrinsic hyperexcitability in iSPNs lacking Atg7 that was characterized by a depolarized RMP, elevated input resistance, decreased rheobase and left-shifted current response curve.

SPN intrinsic excitability is tightly regulated by a complement of potassium channels (*Nisenbaum et al., 1994*; *Shen et al., 2005*; *Shen et al., 2004*; *Wilson and Kawaguchi, 1996*). At hyperpolarized potentials, the RMP and input resistance of the cell is determined by Kir2 currents, mediated by the Kir2.1 and Kir2.3 channels (*Cazorla et al., 2012*; *Karschin et al., 1996*; *Shen et al., 2007*). SPNs are thought to be depolarized from their resting potential of around −80 mV by the coordinated activity of excitatory inputs and once the cell reaches ∼ −60 mV,

the Kir2 current inactivates, and other potassium currents, such as $I_A$, $I_D$, and $I_M$ regulate the number of action potentials fired and the spike frequency (*Kreitzer, 2009*; *Wilson and Kawaguchi, 1996*). The depolarized RMP and increased input resistance around the RMP strongly support a primary role for reduced Kir2 currents in the hyperexcitable phenotype of iSPNs lacking Atg7.

Autophagy has not been previously implicated in the regulation of potassium channels. To address the mechanism through which Atg7 may control Kir2 channels and, in turn, neuronal activity, we conducted experiments in heterologous cells and *in vivo* in iSPN$^{Atg7cKO}$ mice with largely congruent results. We found that ectopically expressed Kir2.1 was localized in an LC3$^+$ compartment in MEFs in an Atg5-dependent manner. Treatment with Bafilomycin A1, an inhibitor of the lysosomal proton pump which prevents fusion of autophagosomes with lysosomes and degradation of autophagic cargo (*Yamamoto et al., 1998*), increased the number of Kir2.1$^+$ LC3$^+$ puncta. This is consistent with previous studies demonstrating that Kir2.1 undergoes lysosomal degradation in mammalian cells (*Jansen et al., 2008*; *Kolb et al., 2014*). Finally, using the SNAPtag system and a cycloheximide pulse chase assay, we found that the half-life of Kir2.1 was greatly extended in Atg5$^{KO}$ and Atg7$^{KO}$ MEFs. These data were supported by the finding that Kir2.1 and Kir2.3 protein levels were higher in the striatum of iSPN$^{Atg7cKO}$ mice without a change in mRNA expression. Together, these data indicate that Atg7-dependent autophagy is required for Kir2 function and degradation.

We further used a viral overexpression strategy to confirm that the decreased Kir2 activity in iSPNs lacking Atg7 did not arise from reduced Kir2 mRNA expression. Viral overexpression of Kir2.1 failed to increase Kir2 currents in iSPNs lacking Atg7, demonstrating that Atg7 is required for newly synthesized channels to become functional. This experiment also permitted us to acquire additional support for differences in Kir2 handling between dSPNs and iSPNs. Although Kir2.1 overexpression failed to increase Kir2 currents in iSPNs lacking Atg7, it did increase Kir2 currents in dSPNs lacking Atg7. These experiments support our conclusions that Atg7 is required for Kir2 channel function but not synthesis, and that dSPNs and iSPNs have different requirements for Atg7 in Kir2.1 activity. We hypothesize that newly synthesized Kir2.1 in iSPNs lacking Atg7 is acetylated and does not undergo autophagic degradation, preventing functional increases in Kir2.1 current.

Autophagy associated genes, such as Atg5 and Atg7, have been proposed to play a role in other cellular processes such as LC3-associated phagocytosis, lysosomal exocytosis and endocytosis (*Subramani and Malhotra, 2013*). Here, we argue that Kir2.1 is a direct substrate of autophagy for the following reasons. First, Kir2.1 degradation is deficient in *both* Atg5$^{KO}$ and Atg7$^{KO}$ MEFs. Additional studies in cells lacking other required autophagy proteins such as Fip200 would further support this hypothesis. Second, we have demonstrated direct colocalization with LC3$^+$ vesicles and that the amount of Kir2.1 in LC3$^+$ vesicles increases in response to blockade of autophagosome-lysosome fusion. Finally, Kir2.1 internalization, measured with antibody feeding assays, is not affected in Atg5$^{KO}$ MEFs, arguing against a role of Atg5 in endocytosis of Kir2.1. Collectively, these data indicate that Kir2.1 must pass through an autophagy-dependent organelle, presumably an amphisome, before undergoing lysosomal degradation.

While autophagic degradation is classically initiated when cytosolic proteins are sequestered in newly formed autophagosomes or autophagic membranes surround membrane-bound organelles (*Klionsky, 2007*), autophagy may be required for Kir2.1 degradation via a distinct mechanism. We found that autophagy is not required for Kir2.1 internalization, suggesting that autophagy acts at a post-endocytic step. Endosomes fuse with autophagosomes to form amphisomes, and this step is required for the degradation of some proteins and the overall function of the autophagosomal and endosomal system (*Filimonenko et al., 2010*; *Hollenbeck, 1993*; *Liang et al., 2008*; *Rabinowitz et al., 1992*; *Sanchez-Wandelmer and Reggiori, 2013*; *Wang et al., 2016*). We speculate that Kir2.1$^+$ endosomes require fusion with an autophagic intermediate for lysosomal delivery. Alternatively, autophagic membranes may form de novo on the surface of Kir2.1$^+$ endosomes, leading to the lysosomal degradation of endosomal contents. We cannot exclude the possibility that Kir2 channels are endocytosed in early autophagic structures that mature into autophagosomes in an Atg5/7-dependent manner for degradation (*Ravikumar et al., 2010*; *Wu et al., 2016*). These potential mechanisms are consistent with data demonstrating a role for autophagy in the degradation of other membrane proteins including

Notch1 (*Wu et al., 2016*), GluR1 (*Shehata et al., 2012*), and the Na+/H+ exchanger regulatory factor 2 (*Gao et al., 2018*).

## Selectivity and cell-type specificity of the autophagic degradation of Kir2 channels

Recent work has highlighted a role for selective, as opposed to bulk, autophagy in the maintenance of cellular homeostasis. Here, we find that loss of autophagy does not lead to a global disruption of membrane or synaptic protein levels, but instead has relatively specific effects on Kir2.1 and Kir2.3 channels. For example, the protein level of the voltage-gated potassium channel, Kv1.2, which is expressed by both dSPNs and iSPNs (*Shen et al., 2004*) is not affected by loss of autophagy. Furthermore, components of the postsynaptic density, such as PSD95, which are degraded by autophagy in excitatory neurons (*Nikoletopoulou et al., 2017*; *Tang et al., 2014*; *Yan et al., 2018*), are not affected by loss of autophagy in dSPNs or iSPNs. Finally, AMPA receptors, which can be degraded by autophagy in excitatory neurons (*Shehata et al., 2012*), do not seem to be affected in iSPNs as mEPSC amplitude is unchanged in iSPN$^{Atg7cKO}$ mice. An unbiased proteomics approach in iSPN$^{Atg7cKO}$ mice would provide additional evidence for the relatively selective degradation of Kir2 channels in iSPNs.

How might this specificity be achieved? Other forms of selective autophagy depend on cargo adapter proteins which interact with autophagosomal proteins, such as LC3, and specific cargo (*Svenning and Johansen, 2013*; *Zaffagnini and Martens, 2016*). In some cases, autophagy cargo adapters select substrates depending on substrate post-translational modifications, providing an additional mechanism that could regulate selectivity (*Khaminets et al., 2016*). Finally, substrates can directly interact with autophagosome proteins via intrinsic LC3-interacting motifs (*Birgisdottir et al., 2013*). Future studies should address possible cargo adapters that bind to Kir2 channels and are required for Kir2 channel degradation or whether Kir2 channels have intrinsic LC3-interacting motifs that could explain why Kir2 channels undergo autophagy-dependent degradation but other membrane proteins in iSPNs do not.

In contrast to iSPNs, Kir2.1 and Kir2.3 abundance and Kir2 currents were unaffected by loss of autophagy in dSPN$^{Atg7cKO}$ mice. As Kir2 channels are expressed in both populations of SPNs and play important roles in both dSPN and iSPN physiology (*Cazorla et al., 2012*; *Gertler et al., 2008*; *Lieberman et al., 2018*; *Shen et al., 2007*), these data suggest that the degradation of Kir2 channels by autophagy occurs via cell type-specific mechanisms. This cell-type specificity could arise from differential expression of cargo adapters, post-translational modification of Kir2 channels that target them for autophagic degradation, or increased activity of alternative endolysosomal protein degradation pathways that contribute to Kir2 degradation in cardiomyocytes (*Ambrosini et al., 2014*; *Jansen et al., 2008*; *Kolb et al., 2014*). Future work will focus on the mechanism that confers cell-type specificity to the autophagic degradation of Kir2 channels.

## Kir2 channels are acetylated and have reduced activity in the absence of autophagy

In previous cases where autophagy regulates membrane proteins, substrate protein levels correlate with substrate protein activity (*Gao et al., 2018*; *Shehata et al., 2012*; *Wu et al., 2016*). Furthermore, most mutations or manipulations that inhibit lysosomal degradation of Kir2.1 lead to increased Kir2 currents (*Ambrosini et al., 2014*; *Jansen et al., 2008*; *Kolb et al., 2014*; *Varkevisser et al., 2013*). In iSPNs, however, loss of autophagy led to increased Kir2 abundance and decreased Kir2 current.

One possible explanation for the inverse correlation between Kir2 protein abundance and Kir2 current in iSPNs and MEFs lacking autophagy could be via changes in its trafficking or surface residence. However, we found that surface levels of heterologously expressed Kir2.1 were elevated in the absence of autophagy, and loss of autophagy was not associated with intracellular Kir2.1 inclusions or increased ER localization in MEFs. This was consistent with our subcellular fractionation results in striatal tissue from control and iSPN$^{Atg7cKO}$ mice, which displayed no difference in the distribution of Kir2 channels but higher levels within each fraction. We thus conclude that autophagy is not required for Kir2 trafficking.

We, therefore, hypothesized that Kir2 channels were inhibited in the absence of autophagy via post-translational modification. We measured the relative levels of ubiquitination and acetylation, two PTMs that can target proteins for autophagic degradation (*Khaminets et al., 2016*), of Kir2.1 in autophagy-deficient cells. We found that acetylation but not ubiquitination of Kir2.1 increased in this condition. K334, a conserved lysine that is in the same intracellular loop as two other lysines that can be ubiquitinated in Kir2.1 (*Ambrosini et al., 2014*; *Ziv et al., 2011*), was required for Kir2.1 degradation in Atg5$^{WT}$ cells. Mutation of K334 to arginine, which cannot be acetylated, rescued channel function in the absence of autophagy. Finally, changing K334 to an acetylation-mimic glutamine was sufficient to reduce channel function in Atg5$^{WT}$ cells. These results demonstrate that acetylation of Kir2.1 at K334 is required for proper channel degradation and function in the absence of autophagy.

While it is likely not the only acetylated residue on Kir2.1, we found that K334 acetylation plays a critical role in channel function and half-life. Although we were unable to biochemically isolate acetylated Kir2.1 in wild-type cells, the genetic evidence suggests that Kir2.1 acetylation at this residue is required for channel degradation. We hypothesize that acetylated Kir2.1 may represent a short-lived intermediate and that the channel is deacetylated when it reaches an Atg5-dependent organelle. Thus, in the absence of autophagy, acetylated Kir2.1 may never colocalize with its cognate deacetylase, explaining the elevated level of acetylated Kir2.1 in Atg5$^{KO}$ MEFs. Defining the specific acetyltransferase and deacetylase that control Kir2.1 acetylation would provide significant insight into how this process is regulated and whether this represents a response the physiological stimuli known to regulate Kir2.1 activity in the striatum (*Cazorla et al., 2012*; *Lieberman et al., 2018*; *Shen et al., 2007*; *Zhao et al., 2016*).

Although acetylation was initially described as a PTM for histones and other nuclear proteins, numerous examples suggest that acetylation may affect the function of non-nuclear proteins, including some proteins involved in autophagy (*Narita et al., 2019*). K334 is located within the Kir2 C-terminus, near regions that interact with obligate cofactors, suggesting a possible mechanism for how acetylation at this residue could affect channel function. Finally, it is attractive to speculate that Kir2 acetylation at K334 could act as a dynamic regulator of Kir2 currents over both long and short timescales: acetylation at K334 would not only reduce channel activity at the surface but also target Kir2 channels for autophagic degradation and affect long-term Kir2 activity.

Several alternative models could explain our data including decreased levels of a required cofactor for Kir2 in autophagy-deficient cells and changes in membrane localization that are below the detection limit of our approaches. Future examination of the Kir2 interactome in wild-type and autophagy-deficient cells could elucidate these mechanisms.

## Implications for neuropsychiatric disorders with dysfunctional autophagy

We demonstrate here that the absence of autophagy in dSPNs or iSPNs results in deficits in striatal-based behaviors. Our data are consistent with reports of hyperactivity in the open-field and reduction of motor learning in multiple animal models with disrupted SPN function (*Durieux et al., 2012*; *Fuccillo, 2016*; *Peça et al., 2011*; *Rothwell et al., 2014*; *Wang et al., 2017*). A disruption of striatal-based learning, hyperactivity and increased stereotypies, as seen in dSPN$^{Atg7cKO}$ and iSPN$^{Atg7cKO}$ mice, is also observed in neurodevelopmental syndromes including autism spectrum disorder (ASD) (*Fuccillo, 2016*). Genetic variants in autophagy-associated genes are risk factors for the development of ASDs and reduced autophagic function is observed in human post-mortem tissue from ASD cases and in mouse models of the disease, suggesting that changes in autophagic function in the striatum in addition to other brain regions may contribute to the pathophysiology of ASDs (*Lieberman et al., 2019b*; *Poultney et al., 2013*; *Tang et al., 2014*; *Yan et al., 2018*). Identifying the cell-type-specific effects of loss of autophagy in distinct neuronal subtypes may permit targeted development of therapeutics that act downstream of the autophagy machinery to correct neural circuit deficits and avoid undesired effects of non-specific autophagic activation.

# Materials and methods

## Key resources table

| Reagent type (species) or resource | Designation | Source or reference | Identifiers | Additional information |
|---|---|---|---|---|
| Gene (Mouse) | Atg7 | | Ensembl: ENSMUSG00000030314 | |
| Gene (Mouse) | Atg5 | | Ensembl: ENSMUSG00000038160 | |
| Strain, strain background (*M. musculus*, Male and Female) | C57/Bl6J | | | |
| Strain, strain background (*E. coli*) | DH5α | ThermoFisher | Cat # 18265017 | |
| Genetic reagent (Mouse) | B6.Cg-Tg(Drd1a-td Tomato)6Calak/J | Jackson Laboratories | RRID: IMSR_JAX:016204 | |
| Genetic reagent (Mouse) | Tg(Adora2a-cre)KG139Gsat | MMMRC | RRID:MMRRC_031168-UCD | |
| Genetic reagent (Mouse) | Tg(Drd1-cre)EY262Gsat | MMMRC | RRID:MMRRC_030989-UCD | |
| Genetic reagent (Mouse) | $Atg7^{tm1.1Tchi}/Atg7^{tm1.1Tchi}$ | Gift of Masaaki Komatsu | RRID:MGI:3590136 | (*Komatsu et al., 2005*) |
| Genetic reagent (Mouse) | 129S1/Sv-$Hprt^{tm1(CAG-cre)Mnn}$/J | (*Tang et al., 2002*) | RRID:IMSR_JAX:004302 | |
| Cell line (Mouse) | $Atg5^{-/-}$ transformed MEF | Gift of Ana Maria Cuervo; (*Kuma et al., 2004*) | RRID:CVCL_0J75 | |
| Cell line (Mouse) | $Atg5^{+/+}$ transformed MEF | Gift of Ana Maria Cuervo; (*Kuma et al., 2004*) | | |
| Cell line (Mouse) | $Atg7^{-/-}$ primary MEF | This study | | See Materials and methods |
| Cell line (Mouse) | $Atg7^{+/+}$ primary MEF | This study | | See Materials and methods |
| Antibody | Rabbit anti-Red fluorescent protein polyclonal | Rockland | Cat # 600-401-379 | See Table S3 in *Supplementary file 1* |
| Antibody | Rabbit anti-DARPP32 monoclonal | Cell Signaling Technology | Cat # 2306S | See Table S3 in *Supplementary file 1* |
| Antibody | Mouse anti-beta actin monoclonal | Novus Biologicals | Cat # NB600-501 | See Table S3 in *Supplementary file 1* |
| Antibody | Mouse anti-Kir2.1 monoclonal | Antibodies Incorporated | Item # 73–210 RRID:AB_11000720 | See Table S3 in *Supplementary file 1* |
| Antibody | Mouse anti-Kir2.3 monoclonal | Antibodies Incorporated | Item # 75–069 RRID:AB_2130742 | See Table S3 in *Supplementary file 1* |
| Antibody | Mouse anti-$K_v$1.2 monoclonal | Antibodies Incorporated | Item # 75–008 RRID:AB_2296313 | See Table S3 in *Supplementary file 1* |
| Antibody | Rabbit anti-PSD95 polyclonal | Abcam | Cat # Ab18258 | See Table S3 in *Supplementary file 1* |
| Antibody | Guinea pig anti p62 polyclonal | American Research Products | Cat # 03-GP62-C | See Table S3 in *Supplementary file 1* |
| Antibody | Rabbit anti p62 polyclonal | MBL | Cat # PM045 | See Table S3 in *Supplementary file 1* |
| Antibody | Rabbit anti LC3B polyclonal | Novus Biologicals | Cat # NB600-1384 | See Table S3 in *Supplementary file 1* |
| Antibody | Chicken anti GFP polyclonal | Abcam | Cat # Ab13970 | See Table S3 in *Supplementary file 1* |
| Antibody | Rat anti Lamp1 monoclonal | Iowa Hybridoma Bank | Cat # 1D4B | See Table S3 in *Supplementary file 1* |
| Antibody | Rabbit anti Kir2.1 polyclonal | Alomone Labs | Cat # APC-026 | See Table S3 in *Supplementary file 1* |

*Continued on next page*

Continued

| Reagent type (species) or resource | Designation | Source or reference | Identifiers | Additional information |
|---|---|---|---|---|
| Antibody | Mouse anti ubiquitin antibody monoclonal VU-1 | LifeSensors | Cat # VU101 | See Table S3 in *Supplementary file 1* |
| Antibody | Mouse anti-acetyl-lysine monoclonal, clone 4G12 | Millipore | Cat # 05–515 | See Table S3 in *Supplementary file 1* |
| Antibody | Mouse Anti-Flag M2 monoclonal | Sigma | Cat # F1804 | See Table S3 in *Supplementary file 1* |
| Antibody | Rabbit anti-HA polyclonal | Abcam | Cat # ab9110 | See Table S3 in *Supplementary file 1* |
| Antibody | Mouse anti-NeuN, monoclonal clone A60 | Millipore | Cat # MAB377 | See Table S3 in *Supplementary file 1* |
| Antibody | Rabbit Anti-Atg7 (D12B11) monoclonal | Cell Signaling Technology | Cat # 8558 | See Table S3 in *Supplementary file 1* |
| Antibody | Mouse anti-Tubulin (TU-01) | Invitrogen | Cat # 13–8000 | See Table S3 in *Supplementary file 1* |
| Antibody | Pierce Protein A/G Magnetic Beads | Thermo | Cat # 88802 | See Table S3 in *Supplementary file 1* |
| Antibody | Goat anti Guinea Pig IgG (H+L) Secondary antibody, Alexa 488 | Invitrogen | Cat # A-11073 | See Table S3 in *Supplementary file 1* |
| Antibody | Goat anti Guinea Pig IgG (H+L) Secondary antibody, Alexa 647 | Invitrogen | Cat # A-24150 | See Table S3 in *Supplementary file 1* |
| Antibody | Donkey anti-Rabbit IgG (H+L) Secondary Antibody, Alexa 488 | Invitrogen | Cat # A-21206 | See Table S3 in *Supplementary file 1* |
| Antibody | Donkey anti-Rabbit IgG (H+L) Secondary Antibody, Alexa 594 | Invitrogen | Cat # A-21207 | See Table S3 in *Supplementary file 1* |
| Antibody | Goat anti-Mouse IgG1 (H+L) Secondary Antibody, Alexa 488 | Invitrogen | Cat # A-21121 | See Table S3 in *Supplementary file 1* |
| Antibody | Donkey anti-Mouse IgG1 (H+L) Secondary Antibody, Alexa 594 | Invitrogen | Cat # A-21125 | See Table S3 in *Supplementary file 1* |
| Antibody | Goat anti-Chicken IgY (H+L) Secondary Antibody, Alexa 488 | Invitrogen | Cat # A-11039 | See Table S3 in *Supplementary file 1* |
| Antibody | Goat anti-Rat IgG1 (H+L) Secondary Antibody, Alexa 647 | Invitrogen | Cat # A-21248 | See Table S3 in *Supplementary file 1* |
| Antibody | Donkey anti-Rabbit IgG (H+L) Secondary IRDye 680LT | LI-COR | P/N 925–68023 | See Table S3 in *Supplementary file 1* |
| Antibody | Streptavidin, Alexa 488 conjugate | Invitrogen | S11223 | See Table S3 in *Supplementary file 1* |
| Antibody | Donkey anti-Mouse IgG (H+L) conjugated to HRP | Jackson Immunoresearch | Code: 715-035-151 | See Table S3 in *Supplementary file 1* |
| Antibody | Donkey anti-Rabbit IgG (H+L) conjugated to HRP | Jackson Immunoresearch | Code: 715-035-152 | See Table S3 in *Supplementary file 1* |
| Recombinant DNA reagent | AAVDJ-CMV-DIO-Kir2.1-t2A-ZsGreen | Stanford Viral Vector Core (*Rothwell et al., 2014*) | AAV61 | |
| Recombinant DNA reagent | AAV2-EF1a-DIO-mCherry ($3.2 \times 10^{12}$ viral genomes/mL) | UNC Viral Vector Core | | |
| Recombinant DNA reagent | AAVDJ-hSyn-FLEX-mmAtg7-t2A-ZsGreen ($1.7 \times 10^{13}$ genome copies/mL) | Vector BioLabs | This study | |
| Sequence-based reagent | RNAScope Multiplex Fluorescent Reagent | Advanced Cell Diagnostics | Cat # 320850 | |

*Continued on next page*

Continued

| Reagent type (species) or resource | Designation | Source or reference | Identifiers | Additional information |
|---|---|---|---|---|
| Sequence-based reagent | RNAScope probe: Mm-Kcnj2 (Kir2.1) | Advanced Cell Diagnostics | Cat # 476261 | |
| Sequence-based reagent | RNAScope probe: Mm-Kcnj4 (Kir2.3) | Advanced Cell Diagnostics | Cat # 525181-C3 | |
| Commercial assay or kit | BCA protein assay kit | ThermoFisher | Cat # 23225 | |
| Commercial assay or kit | Immobilon Western Chemiluminescent HRP Substrate | Millipore | Can # WBKLS0500 | |
| Commercial assay or kit | XhoI | New England Biolabs | Cat # R0146S | |
| Commercial assay or kit | BamHI-HF | New England Biolabs | Cat # R3136S | |
| Commercial assay or kit | HindIII-HF | New England Biolabs | Cat # R3104S | |
| Commercial assay or kit | DpnI | New England Biolabs | Cat # R0176S | |
| Commercial assay or kit | Herculase II fusion DNA polymerase | Agilent Technologies | Cat # 600675 | |
| Commercial assay or kit | dNTP mix | ThermoFisher | Cat # R0191 | |
| Commercial assay or kit | Halt Protease Inhibitor Cocktail (100X) | ThermoFisher | Cat # 78430 | |
| Chemical compound, drug | Bafilomycin A1 | Tocris Biosciences | Cat # 1334 | |
| Chemical compound, drug | Tetrodotoxin citrate | Tocris Biosciences | Cat # 1069/1 | |
| Chemical compound, drug | Picrotoxin | Tocris Biosciences | Cat # 1128/1G | |
| Chemical compound, drug | SNAP-Cell 505-Star | New England Biolabs | Cat # S9103S | |
| Chemical compound, drug | SNAP-Cell TMR-Star | New England Biolabs | Cat # S9105S | |
| Chemical compound, drug | Neurobiotin tracer | Vector Laboratories | Cat # SP-1120 | |
| Chemical compound, drug | TrypLE | ThermoFisher | Cat # 12604054 | |
| Chemical compound, drug | Trichostatin A | Tocris Biosciences | Cat # 1406 | |
| Chemical compound, drug | Nicotinamide | Tocris Biosciences | Cat # 4106 | |
| Chemical compound, drug | N-ethylmaleimide | Sigma | Cat # E3876 | |
| Chemical compound, drug | PR-619 | Sigma | Cat # SML0430 | |
| Software, algorithm | Igor | Wavemetrics | RRID:SCR_000325 | |
| Software, algorithm | pClamp | Molecular Devices | RRID:SCR_011323 | |
| Software, algorithm | Image Studio Lite | LI-COR | RRID:SCR_014211 | |
| Software, algorithm | Benchling | | RRID:SCR_013955 | |
| Software, algorithm | GraphPad Prism 7 | | RRID:SCR_002798 | |
| Software, algorithm | ImageJ | NIH | RRID:SCR_003070 | |

## Contact for reagent and resource sharing

Further information and requests for resources and reagents should be directed to and will be fulfilled by the Lead Contact, David Sulzer (ds43@columbia.edu).

## Experimental model and subject details

### Animals

A2A-cre (KG139) and D1-cre (ey262) were obtained from the Mutant Mouse Resource and Research Center (MMRRC). Mice were backcrossed onto C57Bl6J mice from Jackson Laboratories (Bar Harbor, ME). Atg7$^{Fl/Fl}$ mice were a gift of Masaaki Komatsu (Komatsu et al., 2005) and backcrossed onto the C57Bl6 background. D1-tomato mice (B6.Cg-Tg(Drd1a-tdTomato)6Calak/J) were obtained from Jackson Laboratories. iSPN$^{Ctrl}$ (Atg7$^{Fl/Fl}$) and iSPN$^{Atg7cKO}$ experimental mice were obtained as follows. A2Acre hemizygous mice were crossed with Atg7$^{Fl/Fl}$ mice. A2Acre Atg7$^{Fl/wt}$ offspring were crossed with Atg7$^{Fl/Fl}$ mice, yielding experimental offspring. For experiments with mice expressing the D1-tomato allele, either parent (A2Acre Atg7$^{Fl/wt}$ or Atg7$^{Fl/Fl}$) harboring a single D1-tomato allele was used. The D1-tomato and the A2Acre alleles were always maintained in hemizygosity. The same strategy was used to generate dSPN$^{Ctrl}$ (Atg7$^{Fl/Fl}$) and dSPN$^{Atg7cKO}$ (D1cre Atg7$^{Fl/Fl}$) mice. For viral injections, some crosses included A2Acre Atg7$^{Fl/wt}$, D1cre Atg7$^{Fl/wt}$ with Atg7$^{Fl/wt}$ to generate A2Acre Atg7$^{wt/wt}$ (or D1cre Atg7$^{wt/wt}$) and A2Acre Atg7$^{Fl/Fl}$ (or D1cre Atg7$^{Fl/Fl}$) littermates. Mice were housed on a 12:12 hr light:dark cycle and provided with food and water ad libitum. Mice were genotyped using Transnetyx genotyping services (Memphis, TN). Pups were weaned on postnatal day 18–21 into same sex groups of 3–5 mice. All subjects were randomly assigned to groups. All experiments were conducted according to NIH guidelines and approved by the Institutional Animal Care and Use Committees of Columbia University and the New York State Psychiatric Institute.

### Generation of primary Atg7WT and Atg7KO MEFs

To generate Atg7 wildtype and knockout MEFs, male C57BL/6 mice homozygous for the conditional Atg7$^{Fl}$ allele (Komatsu et al., 2005), were crossed to female mice expressing Cre recombinase in the X-linked *Hprt* gene (Tang et al., 2002), permitting excision of the loxP-flanked exon 14 of Atg7 at the zygote or early cleavage stage in progeny. Mice in the F1 generation (Atg7$^{+/-}$) were intercrossed to generate pregnant dams with F2 embryos for experimental isolation.

At E14.5, embryos were collected from deeply anesthetized pregnant dams. Uterine horns were dissected and rinsed in ice-cold Hank's buffered saline solution (HBSS), and embryos were individually separated into HBSS. The liver, heart, and head were removed from the embryos and collected for genomic DNA extraction. The remaining tissue was incubated in 0.25% trypsin first overnight at 4°C, then 10 min at 37°C. Samples were triturated with fire-polished glass pipettes and diluted in MEF media: DMEM (Thermo Fisher) with 10% FBS (Thermo Fisher), 0.1 mM beta-mercaptoethanol (Sigma), and 1x Antibiotic-Antimycotic (Thermo Fisher). Cells were filtered through a 100-micron sieve and plated in MEF media. After plating, genotypes were confirmed and cells were collected for western blot analysis.

## Method details

### Behavior

Cohorts of iSPN$^{Ctrl}$ and iSPN$^{Atg7cKO}$ or dSPN$^{Ctrl}$ and dSPN$^{Atg7cKO}$ mice were assembled from 4 to 6 litters of the cross described above born within 3 weeks of each other. Behavioral experiments were conducted on these cohorts between 3 and 5 months of age. A similar number of male and female mice was used in behavioral experiments. Data were combined for *Figure 2* as no interaction between sex and genotype was observed. See *Table 1* for behavioral data split by sex. All behavioral results were combined from at least three independent cohorts of mice. Mice in these cohorts were handled weekly by the experimenter and weighed at these times. Two dSPN$^{Atg7cKO}$ mice were excluded from the behavioral analysis due to observed handling-induced seizures.

### Open field

Each mouse is gently placed in the center of a clear Plexiglas arena (27.31 × 27.31×20.32 cm, Med Associates ENV-510) lit with dim light (~10 lux), and is allowed to ambulate freely for 50 min. Infrared (IR) beams embedded along the X, Y, Z axes of the arena automatically track distance moved,

horizontal movement, vertical movement, stereotypies, and time spent in center zone. At the end of the test, the mouse is returned to the home cage and the arena is cleaned with 70% ethanol and wiped dry.

### Rotarod

The accelerating rotarod test is widely to evaluate the motor coordination and motor learning in rodents. Two to four experimental mice are placed on the still rotarod (Ugo Basile) in one lane of the rotating drum (3 cm in diameter, textured to avoid slips) for two minutes. The mice are returned to the homecage for thirty minutes. Mice are then returned to the rotarod, rotating at a constant speed for 5 rpm, for 2 min, followed by return to the homecage for thirty minutes. For the first experimental session (Trial 1), experimental mice are placed on the rotating drum. The rotating speed of the Rotarod (Ugo Basile) is set at 5 rpm. Once all mice are steady, the rotation speed is gradually increase from 5 to 40 rpm over a 5-min period. Latency to fall is recorded automatically when the animal lands on the sensing platform below each lane or when the mouse grabbed on to the drum and executed a full revolution without running. To avoid injury, a cut-off latency of 300 s is applied to each trial. Animals were returned to the home cage for 30 min followed by two additional trials. The drum and the platform are wiped clean with paper towel moistened with 70% ethanol and let dry.

### Home cage repetitive behaviors

Repetitive behaviors and stereotypies such as excessive self-grooming, circling, jumping, and back flipping were evaluated in a simple test using clean mouse cages as described in a recent study (*Yang et al., 2015*). Each subject mouse was placed in a clean home cage, allowed to habituate for 30 min and monitored for an additional 30 min. The cage bottom was covered with a thin layer of clean bedding. The cage is equipped with a metal wire bar lid as some repetitive back flipping involve the use of the metal lid. Occurrences of self-grooming were quantified for one minute every ten minutes from the final thirty minutes and summed together.

### Viral injections

D1cre Atg7$^{wt/wt}$, D1cre Atg7$^{Fl/Fl}$, A2Acre Atg7$^{wt/wt}$ or A2Acre Atg7$^{Fl/Fl}$ mice (2–3 months old litter-mates of both sexes) were anesthetized with 5% isoflurane. Animals were transferred onto a Kopf Stereotax and maintained under isoflurane anesthesia (1–2%). Hair was removed, the scalp was sterilized using chlorhexidine solution and an incision was made. The coordinates of Bregma and Lambda were determined. Virus was injected at AP +0.8, ML, −1.6, and DV −2.8 from the dura. A small hole was drilled into the skull and 230 nL of virus (see above table for viral titers) was injected through a glass pipet using a Nanoject 2000 (Drummond Scientific; 50 pulses of 4.6 nL). The glass pipet was slowly withdrawn after 5 min. The skull was closed using vicryl sutures. Animals were housed for 3–4 weeks before being sacrificed for electrophysiology.

### Cloning/molecular biology

The Kir2.1 coding sequence was PCR amplified from pAAV-mmKir2.1 (Gift from C. Kellendonk) into FUGW-VAMP2-Flag-SNAP (*Sheehan et al., 2016*) using the XhoI-BamHI sites (Tables S1 and S2 in *Supplementary file 1*). Kir2.1-ExtHA was synthesized by Genewiz with the HA tag inserted at amino acid 114 (*Chen et al., 2002*) and subcloned into FUGW-mmKir2.1-FLAG-SNAP. For cell culture electrophysiology experiments, mmKir2.1 CDS was PCR amplified from FUGW-mmKir2.1-FLAG-SNAP into the HindIII-BamHI sites of pcDNA2.1-eGFP (Addgene 13031).

Site-directed mutagenesis of FUGW-mmKir2.1-FLAG-SNAP or FUGW-mmKir2.1-ExtHA-FLAG-SNAP was conducted using Herculase II polymerase and primer pairs containing the desired mutation according to the manufacturer's instructions (See Table S2 in *Supplementary file 1*). The PCR reaction was then treated with DpnI for 4–6 hr at 37°C, column purified (Qiagen) and transformed into DH5α competent cells. Mutated sequences were confirmed by Sanger sequencing (Genewiz or Eton Biosciences).

Plasmid preps for transfection were generated using Qiagen midi or maxiprep kits following manufacturer instructions.

## Cell culture

Atg5[WT] and Atg5[KO] MEFs were obtained from Ana Maria Cuervo (Albert Einstein College of Medicine, New York). Cells were cultured using standard techniques in Dulbecco's modified Eagle medium (DMEM; Gibco) supplemented with 10% fetal bovine serum (Gibco) and antibiotic-antimycotic mix (Gibco). Cells were not passaged more than ten times. Cells were incubated at 37°C in 5% $CO_2$. Cell lines were Mycoplasma negative. Genotype of the cell line was validated biochemically with assays for autophagy function as described in the manuscript.

## Transfection

For experiments, cells were plated and allowed to settle for 1–2 days. Cells were then transfected with Calfectin and plasmid DNA prepared with Qiagen Maxiprep or Midiprep kits according to manufacturer's instructions. Experiments were conducted 36–48 hr after transfection.

## Electrophysiology

### Slice

Animals of either sex in *Figures 3–4* were 4–6 weeks old at the time of experiment. Animals in *Figure 6* were 3–4 months old. No interaction between sex and genotype was observed in electrophysiological or morphological analysis; however, we were underpowered for these analyses. Acute slices were generated as described (*Lieberman et al., 2018*). Briefly, mice underwent rapid cervical dislocation without anesthesia. Brains were removed and submerged in ice-cold cutting solution (in mM): 10 NaCl, 2.5 KCl, 25 $NaHCO_3$, 0.5 $CaCl_2$, 7 $MgCl_2$, 1.25 $NaH_2PO_4$, 180 sucrose, 10 glucose bubbled with 95% $O_2$/5% $CO_2$ to pH 7.4. The cerebellum was removed and the brain was mounted on its caudal surface on a vibratome. 250 μm coronal sections including the striatum were collected.

Slices were allowed to recover at 34°C for 30 min in artificial cerebrospinal fluid (ACSF; in mM): 125 NaCl, 2.5 KCl, 25 $NaHCO_3$, 2 $CaCl_2$, 1 $MgCl_2$, 1.25 $NaH_2PO_4$ and 10 glucose bubbled with 95% $O_2$/5% $CO_2$ to pH 7.4. Slices were then maintained at room temperature throughout the remainder of the experiment.

Slices were transferred to a recording chamber under constant perfusion with carbogenated ACSF at 1.5–2 mL/min. Cells in the dorsal striatum were identified using a 40X water immersion objective and IR/DIC optics. D1-tomato, mCherry or GFP fluorescence was observed as indicated. Liquid junction potential was not corrected. Data were digitized at 10 kHz and filtered at 5 kHz. Signals were acquired on an Axon Instruments Axopatch 200B, digitized on a Digidata 1440A (Axon Instruments). All recordings were made within 6 hr of slice preparation.

For synaptic recordings (in voltage clamp), whole cell recordings were established with glass pipettes (3–6 MΩ) filled with internal solution of (in mM): 120 $CsMeSO_3$, 5 NaCl, 10 HEPES, 1.1 EGTA, 4 MgATP, 0.3 NaGTP, 10 TEA, 1 mg/mL Qx314 Bromide, pH 7.25. Once the whole-cell configuration was established, the internal solution was allowed to diffuse into the cell for at least 10 min. Pipette capacitance was compensated. Series resistance was continuously monitored and cells were excluded if series resistance was > 25 MΩ, changed by more than 20% or pipette capacitance transients changed by more than 20%. TTX (1 μM) was perfused in the ACSF. mEPSCs were recorded at −70 mV for a 2–4 min epoch. Cells were then held at 0 mV for a 2–4 min epoch to record mIPSCs. Traces were analyzed offline using a custom Igor script which has previously been reported by our lab for amperometric quantal analysis (*Mosharov and Sulzer, 2005*). Source code is included with the manuscript. Traces were imported to Igor and converted from. abf files. Traces were 'smoothed' using the plugin. mEPSC traces were inverted (as the plugin detects positive-going peaks) and peaks were detected using a threshold of 4 standard deviations above the noise. Median interevent interval was used to calculate the event frequency and median peak amplitude is presented. For mIPSC traces, a threshold of 3 standard deviations was set. Settings were determined following manual inspection of a subset of records (from both control and cKO neurons) and then kept the same for final analysis of all traces.

For current clamp recordings, glass pipets were filled with (in mM): 115 potassium gluconate, 20 KCl, 20 HEPES, 1 $MgCl_2$, 2 MgATP, 0.2 NaGTP adjusted to pH 7.25 with KOH, osmolarity 285 mOsm. ACSF contained 1.5 mM $CaCl_2$ instead of 2 mM $CaCl_2$ to reduce contribution of calcium currents to Kir2 current measurements (*Lieberman et al., 2018*). Excitability parameters and Kir2 currents were measured as described in *Lieberman et al. (2018)*. For current density measurements,

the barium-sensitive current (0.1 mM $BaCl_2$) was normalized to that cell's capacitance measured in current clamp recordings as described (*Lieberman et al., 2018*).

## MEFs

MEFs were cultured as described above. For electrophysiology experiments cells were cultured on 15 mm diameter glass coverslips placed in 35 mm diameter tissue culture dishes. Cells were transfected 36–48 hr before the experiment as described above. At the time of the experiment, coverslips were washed three times in recording saline (in mM): 135 NaCl, 4.8 KCl, 1.8 $CaCl_2$, 1 $MgCl_2$, 10 Glucose, 5 HEPES, pH 7.4 (*Ambrosini et al., 2014*). Cells were identified using IR/DIC optics and fluorescence was confirmed. In recordings in *Figure 5*, cells were transfected with Kir2.1-Flag-SNAP and labeled 1 hr prior to recording with SNAPcell TMR Star as described below. Because of the low intensity of the TMR star fluorescence, we subcloned Kir2.1 into pcDNA-eGFP as described above to have native GFP fluorescence for recordings in *Figure 6*. No difference was found in maximal Kir2 amplitude between cells transfected with Kir2.1-Flag-SNAP compared to Kir2.1-eGFP. Glass pipets (2–4 MΩ) were filled with (in mM): 115 potassium gluconate, 20 KCl, 20 HEPES, 1 $MgCl_2$, 2 MgATP, 0.2 NaGTP adjusted to pH 7.25 with KOH, osmolarity 285 mOsm. After the whole cell configuration was established, cells were voltage clamped at −60 mV and currents were recorded using the same protocol as used for SPNs above (*Lieberman et al., 2018*).

## Neuronal fills/spine counting

Dendritic reconstructions were obtained and analyzed essentially as described in *Lieberman et al. (2018)*. For dendritic reconstructions, neurobiotin (1 mg/mL, Vector Laboratories) was added to the internal solution. Once the whole cell configuration was established, the neurobiotin was allowed to diffuse into the recorded cell for 15 min. The patch pipette was then carefully withdrawn. The slice was removed from the recording chamber and fixed overnight in 4% PFA in 0.1M phosphate buffer (PB), pH7.4. Only one cell was filled per slice. Slices were then washed in TBS. Slices were stained overnight with Alexafluor488-conjugated streptavidin (1:200, ThemoFisher) in 0.6% TritonX-100 in TBS. Slices were then washed in TBS and mounted on slides. Slices were allowed to dry overnight in the dark before being coverslipped with Fluoromount-G containing DAPI. Slides were imaged on a Leica SP5 confocal microscope in system optimized Z-stacks using a 20X objective. Segments of the dendritic tree approximately 50 microns from the soma were subsequently imaged with a 63x/1.4x oil objective with system optimized z stacks to permit analysis of spine density.

Dendritic trees were traced using the simple neurite tracer plugin in ImageJ. Traced neurites were collapsed into a max projection and analyzed using the Sholl analysis plugin. Dendritic spine density was manually determined from max projections of z-stacks obtained using the 63x objective. Dendritic protrusions defined by being 0.5–2 µm long with a shaft and head were counted from at least three dendritic segments of 20–50 µm per cell. The number of spines per micron were then averaged for each cell. At least three cells were analyzed per animal. Data were combined from 3 to 5 animals per group.

## In vivo biochemistry

Whole striatal lysates were generated by acute cervical dislocation and decapitation. The brain was rapidly removed and the striatum was removed as described in *Lieberman et al. (2018)*. Striata were sonicated in RIPA buffer containing 1X protease inhibitor cocktail (Pierce). Protein concentration was determined using the BCA kit and lysates were diluted in sample buffer. Lysates were stored at −80°C until analysis.

The synaptosome preparation technique was adapted from a recent study (*Bermejo et al., 2014*). Two striata from a single mouse were removed as above and dounce homogenized in 1 mL of 320 mM sucrose, 4 mM HEPES pH 7.4 on ice. All subsequent steps were conducted at 4°C. Homogenized tissue (TH, total homogenate) was spun at 1000xg for 10 min. The supernatant was collected (PNS; post-nuclear supernatant) and spun at 10000xg for 15 min. The supernatant was collected and labeled S2. The pellet (Crude synaptosomal pellet; P2) then underwent hypotonic lysis by resuspending the pellet in 1 mL ddH$_2$O followed by 4 strokes of dounce homogenization. Four µL of 1 M HEPES pH7.4 was added and the sample was rotated for 30 min. Until this point, all centrifugation steps were completed on a tabletop ultracentrifuge. The sample was then spun at 25000 x g for

20 min in a fixed angle S140-AT rotor in a Sorvall ultracentrifuge. The supernatant (S3) was collected. The pellet was resuspended in 320 mM Sucrose, 4 mM HEPES pH7.4 and layered on top of a discontinuous sucrose gradient (0.8M, 1.0M, 1.2M). The sample was then spun 150,000xg for 2 hr. The 1.0M, 1.2M interface was collected and diluted to 320 mM sucrose, 4 mM HEPES pH7.4. The samples was then spun at 200,000 x g for 30 min to collect the synaptic plasma membranes (SPM). Aliquots were collected from each fraction described during the procedure. Equivalent protein concentrations of each fraction was used for Western blot analysis.

For the optiprep (iodixanol) gradient, samples were prepared using a procedure modified from Optiprep Application Sheet S62). Striata were dissected as for synaptosomal preparation and dounce homogenized in 250 mM Sucrose, 1 mM EDTA, 20 mM HEPES pH7.4. This homogenate was spun for 10 min at 1000xg. The supernatant was spun on an S80AT3 fixed-angle rotor at 100,000xg for 1 hr. The pellet was resuspended in 30% iodixanol, 250 mM Sucrose, 1 mM EDTA, 20 mM HEPES pH7.4. On top of the pellet, the following layers were loaded in equal volumes: 30%, 25%, 17%, 10%, 2.5% and spun at 165,000xg for 3.5 hr. 100 µL fractions were collected and equal volumes were run on a gel for western blot analysis.

Preparations were completed for the specified number of adult mice (3–4 months old, littermate and sex-matched iSPN$^{Ctrl}$ and iSPN$^{Atg7cKO}$) in parallel. For the total lysate experiment, samples were collected from four separate cohorts and combined for final analysis.

## Perfusion fixation

Perfusion fixation was conducted as described (*Lieberman et al., 2018*). Animals were deeply anesthesized with an intraperitoneal injection of euthasol. Following loss of the righting and toe-pinch reflex, animals were transcardially perfused with 10–20 mL 0.9% NaCl followed by 50 mL of freshly prepared 4% paraformaldehyde in 0.1M PB, pH 7.4. Brains were removed and post-fixed in 4%PFA overnight at 4°C. Brains were subsequently washed with PBS and sectioned in 30–50 µm sections on a Leica vibratome. Sections were stored at −20°C in cryoprotective solution (30% ethylene glycol, 30% glycerol, 0.1M PB, pH7.4) until staining.

## Immunohistochemistry

Immunohistochemical analysis was performed as described (*Lieberman et al., 2018*). Briefly, sections were removed from cryoprotectant solution and washed three times in TBS. Sections were then blocked in 10% normal donkey serum (NDS), 0.1% Triton-X 100, TBS for 1 hr at room temperature. Sections were then transferred to primary antibody solution in 2% NDS, 0.1% Triton-X 100, TBS overnight at 4°C. The next day, sections were washed three times in TBS and subsequently incubated in secondary antibodies in 2% NDS, 0.1% Triton-X 100, TBS at room temperature for 1 hr. Alexafluor conjugated secondary antibodies were used at a concentration of 1:500. Sections were then washed in TBS and mounted on slides, coversliped with Fluoromount-G containing DAPI and stored for imaging.

## RNAScope

Brain sections were removed from cryoprotectant and washed in TBS. Sections were then mounted on charged Superfrost slides and allowed to dry overnight. A border was made with a PAP pen around the sections. Sterile water was dropped onto the slide and allowed to sit twice for two minutes. Pretreat IV was dropped onto the slide and slides were incubated in a humidified oven at 40°C for 30 min. Sections were rinsed twice for 2 min each in sterile water. Premixed probes were added to the slide and allowed to incubate for 2 hr at 40°C. Each section received probes against either Kir2.1 or Kir2.3. A custom probe in the C3 channel was developed by Advanced Cell Diagnostics (Newark, CA) for Kir2.3. A commercially available probe in the C1 channel was used for Kir2.1. Signals were amplified using the RNAScope Fluorescence Multiplex Kit. Sections were then immunohistochemically labeled for DARPP32 and p62 as described above.

## Cell culture lysates

Cells were grown in 10 cm diameter dishes. Dishes were rinsed twice with PBS. RIPA buffer containing 1X protease inhibitors, deubiquitinase inhibitors (5 mM N-ethylmaleimide, 50 µM PR619) and deacetylase inhibitors (200 nM Trichostatin A and 5 mM Nicotinamide) was added to the plate.

Dishes were rotated at 4°C for 5 min. Cells were scraped off of the dish, sonicated and allowed to sit on ice for 1 hr. Lysates were then spun at 10,000xg for 10 min. The supernatant was saved. Protein concertation was determined.

## Cycloheximide pulse chase

Cells were plated and transfected in a 6-well plate. 36–48 hr after transfection with FUGW-Kir2.1-FLAG-SNAP, cells were treated with cycloheximide (0.1 mg/mL) for the specified amount of time. Cells were then lysed as described above.

## Immunoprecipitation

Cells were transfected with FUGW-Kir2.1-extHA-FLAG-SNAP or FUGW-Kir2.1-K334R-FLAG-SNAP 36–48 hr before lysis. Cell lysates were generated as above. All steps were performed at 4°C. For immunoprecipitation (IP), equal amounts of protein (0.5–1.0 mg) of lysate was precleared with Protein A/G magnetic beads (Thermo) for 1 hr. Beads were removed with a strong magnet. Four µg of rabbit anti-HA (Abcam) were added to the cleared supernatant and rotated overnight. The next day, 25 µL of Protein A/G magnetic beads were added and samples were rotated for 1 hr. Beads were collected, washed once in one volume of RIPA buffer, three times in one volume of RIPA buffer containing 500 mM NaCl (instead of 150 mM NaCl) and one final time in RIPA Buffer. Beads were then lysed in 1X sample buffer. The entire eluted volume of the IP was used for a gel.

## Western blot analysis

10% or 12% polyacrylamide homemade gels were prepared as described (*Santini et al., 2007*). For some experiments, 4–12% gradient NuPage gels were used. Protein was transferred on PVDF membrane (pore size 0.2 µm; Immobilon-Fl). Membranes were blocked in 5% non-fat milk (Biorad) in TBS-T (TBS + 0.1% Tween-20) at room temperature for 1 hr. Membranes were incubated overnight at 4° in primary antibody in 5% bovine serum albumin in TBS-T. Antibody concentrations are specified in Table S3 in *Supplementary file 1*. Blots were then washed three times in TBS-T, followed by incubation in secondary antibody in 5% milk TBS-T for one hour at room temperature. Blots were washed three more times in TBS-T.

Blots were developed as follows: Blots for PSD-95 and $K_V1.2$ were imaged using a LICOR Odyssey. All other blots were developed using HRP-conjugated secondaries and Immobilon enhanced chemiluminescence solutions (Millipore) and imaged on an Azure Biosystems C600 system.

For some experiments, blots were stripped using 2% SDS, 62 mM Tris pH6.8, 0.8% β-mercaptoethanol at 50–55°C for 30 min. Blots were then rinsed in ddH$_2$O and washed with TBS-T as described in *Kanner et al. (2017)*.

Blots developed using the ECL technique were analyzed in ImageJ using standard routines. Blots developed using the LICOR Odyssey were analyzed in ImageStudio Lite.

## Snap labeling/immune/quantification

For YFP-LC3 colocalization with Kir2.1, FUGW-Kir2.1-FLAG-SNAP and an expression vector expressing YFP-LC3 were co-transfected. Some cells were treated with Bafilomycin (100 nM) or DMSO vehicle control (DMSO concentration 0.1%) for 2 hr.

SNAPcell ligands were purchased from New England Biolabs, dissolved to 1 mM in DMSO and frozen in aliquots. Transfected cells (with FUGW-Kir2.1-FLAG-SNAP or specified mutants) on glass coverslips were placed in a 24-well plate in complete media. The media was replaced with complete media containing 1 µM SNAPcell ligand and incubated at 37°C in 5% CO$_2$ for 30 min. Coverslips were then washed three times in PBS and returned to the incubator in fresh complete media (without SNAPcell ligand) for 30 min. Complete media was replaced again. This was considered time = 0 min. At the specified time, cells were removed from the incubator and fixed as described below.

## Immunofluorescence

Cells were fixed as described (*Lieberman et al., 2017*). Cells on glass coverslips were removed from the incubator and washed three times in PBS. Cells were then fixed in 4% paraformaldehyde (Electron Microscopy Sciences) diluted in 5% Sucrose and 1X PBS for 5 min at room temperature.

Coverslips were then washed in PBS and incubated in pre-chilled methanol and stored at −20°C until staining.

Coverslips were removed and washed in PBS three times. Coverslips were then blocked in 10% Normal Donkey serum in PBS with 0.1% triton x-100 for 1 hr at room temperature. Coverslips were then incubated for 1 hr at room temperature in primary antibody (See Table S4) in 2% NDS, PBS, 0.1% Triton X-100. Coverslips were then washed three times in PBS and incubated in secondary antibody diluted in 2% NDS, PBS, 0.1% Triton X-100 for 1 hr at room temperature. Coverslips were then washed in PBS and mounted on glass slides with Fluoromount-G containing DAPI.

For experiments that quantified colocalization of Kir2.1 signal with another organelle marker, images were acquired using a confocal microscope (Leica SP5 system) in a single mid-nuclear plane using a 63/1.4x oil objective.

## Surface labeling/internalization

For determining the level of Kir2.1 on the cell surface in cell culture, cells were grown and transfected as described above with FUGW-Kir2.1-extHA-FLAG-SNAP. Cells were then fixed in 4% PFA, 5% Sucrose, 1X PBS for 10 min at room temperature. Coverslips were washed in PBS and then blocked in 10% NDS, PBS (without triton) for one hour at room temperature. Coverslips were then incubated in Rabbit anti HA (Abcam) in 2% NDS, PBS (no triton) for one hour at room temperature. Coverslips were washed three times in PBS and then blocked and permeabilized with 10% NDS, PBS and 0.1% Triton X-100 for 1 hr at room temperature. They were then transferred to primary antibody solution containing Ms anti FLAG M2 (Sigma) 2% NDS, PBS, 0.1% Triton X-100 for 1 hr at room temperature. Coverslips were then washed in PBS and transferred to secondary antibody in 2% NDS, PBS, 0.1% Triton X-100 for 1 hr followed by washes and mounting.

For quantifying Kir2.1 internalization, live, transfected cells on glass coverslips were washed in ice cold Earl's balanced salt solution (EBSS). Coverslips were then labeled with Rabbit anti HA antibody (10 µg/mL) in 5% BSA in EBSS for 15 min at 4°C. Cells were then washed 2x in ice-cold 5% BSA/ EBSS, 1x in PBS and into complete media. This was considered time = 0 min. Coverslips were fixed at the specified time as described for surface labeling. Coverslips were blocked for 30 min in 10% NDS, PBS followed by incubation in AlexaFluor conjugated anti Rabbit secondary antibody. Cells were then blocked and permeabilized for staining against the FLAG tag as described above.

## Epifluorescence imaging

For quantification of SNAPcell fluorescence, surface labeling and surface internalization, cells were imaged following staining on an Olympus IX81 inverted fluorescence microscope using a 63x/1.35 oil objective (Olympus) and the corresponding fluorescence fulter set. Images were captured using an Orca Flash 4 V3 camera (Hamamatsu) and MetaMorph software (Molecular Devices). For all experiments, specified fluorescence (i.e. SNAPcell ligand, surface signal) was normalized to FLAG fluorescence intensity (total Kir2.1 protein) for each cell to control for variations in the level of Kir2.1 expression following transient transfection. Exposures were set to ensure that no pixels were saturated. The exposure times were constant for each fluorescence channel across condition. For each transfection (independent replicate) at least 20 cells from three to four fields were imaged. In all experiments, data was combined from at least three independent replicates, unless otherwise specified.

## Image analysis

### Brain slice

NeuN, p62, or D1-tomato -positive cells were manually counted in single Z-planes of 500 µm$^2$ containing the dorsal striatum. Cells were counted from 4 to 8 images per animal. RNAScope puncta were manually counted in single Z-planes. The number of cells counted are listed in the Figure legend and arise from sections from at least four mice. All counting was conducted by an observer blind to genotype.

### Cell culture

For analysis of colocalization of Kir2.1 puncta with organelle markers (YFP-LC3, Lamp1, or EEA1), a mask was created using a multiple threshold analysis plugin on ImageJ (*Borgkvist et al., 2015*) over

YFP-LC3, Lamp1, or EEA1 puncta in single, mid-nuclear Z-planes. Results were similar when data were combined from multiple Z-sections per cell. Puncta with Kir2.1 intensity greater than two standard deviations above the background were considered Kir2.1+.

Kir2.1-FLAG-SNAP degradation was analyzed using epifluorescent images (as described above) and ImageJ. ROIs were set over cells in the FLAG channel (total Kir2.1) and the mean intensity of this ROI was collected from the FLAG channel and the SNAPcell labeled channel (pulse-labeled Kir2.1). The ratio of SNAPcell intensity of FLAG intensity was determined for each ROI to normalize SNAPcell intensity to total Kir2.1 levels (*Sheehan et al., 2016*). This ratio was collected form the specified number of cells and combined from three individual coverslips (separate transfections and labeling) per time point.

A similar approach was taken to analyze surface staining of Kir2.1-extHA-FLAG-SNAP or Kir2.1 internalization. An ROI was drawn over cells in the FLAG channel (total Kir2.1) and the mean intensity of the FLAG channel and the HA channel was collected. The intensity of the HA channel was normalized against the FLAG intensity for each cell/ROI. This ratio was collected form the specified number of cells and combined from three individual coverslips (separate transfections and labeling) per time point.

All image analysis was conducted blind to condition.

## Data analysis

All data analysis were conducted blinded to genotype and treatment. Electrophysiology data was analyzed offline using Clampfit Software (Molecular Devices, Sunnyvale, California). Statistical analysis was conducted in GraphPad Prism 7 (La Jolla, CA). No explicit power analysis was used but group sizes were determined based on past work from our group (*Lieberman et al., 2018*; *Tang et al., 2014*). Comparisons between two groups were made with a two-tailed, unpaired t-test. Comparisons between two or more groups were made with a one-way ANOVA followed by a Bonferroni post-hoc test. Comparisons with two variables were made with a two-way ANOVA followed by Bonferroni post-hoc tests unless otherwise specified. For electrophysiology N is the number of cells recorded from and the number of mice is listed in parentheses. Data were not formally tested for being parametric. Some datasets were formally tests for outliers with the Grubbs' test with a cutoff of $p < 0.05$. RNAScope data (puncta per cell) were analyzed with the Kolmogorov-Smirnov test.

## Acknowledgements

We thank Drs. C Kellendonk, M Quick, D Agalliu, E Gallo, M Sonders, M Salling and T Cheung for invaluable advice and discussions throughout this project; Dr. AM Cuervo for invaluable advice and cell lines; Drs. J Crittenden and E Mosharov for critical review of an earlier draft of the manuscript; and E Kanter and I Pigulevskiy for generous support.

## Additional information

### Funding

| Funder | Grant reference number | Author |
| --- | --- | --- |
| National Institute of Mental Health | 5F30MH114390-02 | Ori J Lieberman |
| National Institute of General Medical Sciences | T32GM007367 | Ori J Lieberman Christopher J Griffey |
| National Institute on Drug Abuse | R01DA007418 | David Sulzer |
| Simons Foundation | 514813 | David Sulzer |
| JPB Foundation | | David Sulzer |
| National Institute of Neurological Disorders and Stroke | R00NS087112 | Emanuela Santini |
| National Institute of Neurological Disorders and Stroke | R01NS063973 | Ai Yamamoto |

The funders had no role in study design, data collection and interpretation, or the decision to submit the work for publication.

## Author contributions
Ori J Lieberman, Conceptualization, Data curation, Formal analysis, Funding acquisition, Validation, Investigation, Visualization, Methodology; Micah D Frier, Christopher J Griffey, Elizabeth Rafikian, Investigation; Avery F McGuirt, Validation, Investigation; Mu Yang, Supervision, Investigation, Methodology; Ai Yamamoto, Anders Borgkvist, Supervision, Methodology; Emanuela Santini, Supervision, Funding acquisition; David Sulzer, Conceptualization, Supervision, Funding acquisition

## Author ORCIDs
Ori J Lieberman (iD) https://orcid.org/0000-0002-0467-0875
Ai Yamamoto (iD) http://orcid.org/0000-0002-7059-2449
David Sulzer (iD) https://orcid.org/0000-0001-7632-0439

## Ethics
Animal experimentation: This study was performed in strict accordance with the recommendations in the Guide for the Care and Use of Laboratory Animals of the National Institutes of Health. All of the animals were handled according to approved institutional animal care and use committee (IACUC) protocols (AAAV9459 and AAAR4414) of Columbia University.

## Decision letter and Author response
Decision letter https://doi.org/10.7554/eLife.50843.sa1
Author response https://doi.org/10.7554/eLife.50843.sa2

# Additional files

## Supplementary files
- Source data 1. Source data for all Figures.
- Source code 1. Source code for peak identification.
- Supplementary file 1. Supplemental Tables.
- Transparent reporting form

## Data availability
All relevant data are present within the manuscript and supporting files. Source data files have been provided for all Figures.

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
