## [Decision Letter]

**Acceptance summary:**

This study succinctly demonstrates, through multiple orthogonal approaches and powerful conditional models, that macroautophagy can control neurotransmission and animal behavior with cell-type dependent mechanisms. The demonstration that hyperexcitability arises in iSPNs in the striatum due to loss of macroautophagy and Kir2-channel acetylation accumulation provides a mechanism to understand these observations. In other populations of striatal projection neurons, dSPNs, loss of macroautophagy led to more dramatic dendritic morphological changes. These are some of the first demonstrations of autophagy control of neuronal intrinsic excitability and highlights how strikingly different the effects of the same macroautophagy depletion can be on biochemically/morphologically similar cells. Many readers will be left with the important impression that one cannot a priori assume how different neuronal populations will respond to inhibition of autophagy.

**Decision letter after peer review:**

Thank you for submitting your article "Cell-type specific regulation of neuronal intrinsic excitability by macroautophagy" for consideration by *eLife*. Your article has been reviewed by three peer reviewers, including Andrew B West as the Reviewing Editor and Reviewer #1, and the evaluation has been overseen by Gary Westbrook as the Senior Editor. The following individuals involved in review of your submission have agreed to reveal their identity: Austen J Milnerwood (Reviewer #3).

The reviewers and Editors have all discussed the reviews with one another and the Reviewing Editor has drafted this decision to help you prepare a revised submission.

Summary:

The authors present a dense body of work that is overall compelling and comprehensive, focusing on mice that combine Drd1-Cre and Adora2A-Cre with Atg7-fl/fl. The report has many strengths – autophagy in both dPSN and iPSN with electrophysiology analysis; describing a novel autophagy substrate Kir2 and potential molecular basis for its degradation and inhibition; and cell type-specific degradation of Kir2 in the brain, etc. This is solid work from a lab that has published many interesting reports on the role of autophagy in neuronal function.

Essential revisions:

1) All reviewers thought the manuscript was still in a draft format, with critical errors in cross-referencing figures, as well as sub-optimal organization for *eLife*. As described in the compendium of points made below, the manuscript is not ready for publication and was unnecessarily difficult to read through. A careful rewriting and re-editing is necessary.

2) Overall, reviewers thought that some experiments were underpowered (based on the proposed effect size, or say, a rodent experiment with few animals), and these should be distinguished better from central points (perhaps these data moved to supplement). In turn, some of the supplemental that presented critical data should be moved into the main figure set. Further, sex should be an included as a co-variable and mentioned in all legends, as well as additional clarity as to the total number of animals involved in each experiment. In general, all reviewers felt the strength of the conclusions did not always match the experimental rigor, so the text may need to be more conservatively modified in this regard.

3) The discourse is lacking elements of critical interpretation and alternative explanations, in general, which we think will lessen the impact of the work for some readers. Some relevant (specific) points to consider are listed below:

Reformatting: Figure 6 (confusingly referred to as Supplemental Figure 5 in the Results section?) should probably be a main figure with a broader discussion. Similarly, Figure 1 is very important and demonstrates functionally the success of the model in achieving autophagy deficiency in the majority of SPNs (with the D1-tomato being a very nice control here). There are no space requirements that should necessitate the constant flipping between supplement and main figure files for important experiment results. Along these lines, the abstract is overly tight and awkwardly crafted in flipping back and forth between the iSPN and dSPN data.

Interpretation: Does the timing of autophagy deficiency and p62 accumulation with respect to the developmental stage of outgrowth and synapse formation account for some of the phenotypic differences between iSPNs and dSPNs observed? (Does Drd1 and Adora2 expression activate in the exact same phase of SPN maturation, and even with that, could differential Cre strength (i.e., expression of Cre) exact a delay in recombination in iSPNs compared to dSPNs?)

Interpretation: the authors relatively ignore why the iSPNs are relatively resistant to the morphological changes caused by autophagy deficiency seen in other neurons, but more discussion here could help interpretation.

Interpretation: It is intriguing that accumulation of Kir2 occurs only in Atg7 specific deletion iPSN but not dPSN. If Kir2 is indeed a substrate of autophagy as the study indicated, it is puzzling that cKO of Atg7 in dPSN does not cause Kir2 accumulation. The authors could consider in the text the possibility that the phenotypes observed in Atg7 cKO mice are non-autophagy related, unless the authors also examined additional cKO mice in dPSN specific manner with another Atg deletion. Otherwise I suggest modifying the use of "autophagy" only throughout the manuscript in interpreting the data; at least for the conclusion "Atg7-mediated autophagy" could be better.

Interpretation: Reviewers were not satisfied by the stated explanation for the inhibition of Kir2 activity due to acetylation caused by lack of Atg7 or autophagy in iPSN. For example, the increased level of Kir2 and lack of Kir2-over-expression rescue seemingly necessitates a major fraction of Kir2 protein acetylated. But in theory those Kir2 that are destined for degradation only will be modified and inhibited. Therefore, additional mechanisms could be involved in the inhibition. How common is it for the Kir2 channel to be degraded by autophagy in CNS neurons in general? Would dPSN be an unusual cell type for Kir2 not being degraded by autophagy? Or autophagy is not essential for Kir2 degradation in dPSN? Speculate the mechanism?

Interpretation: In the second paragraph of the Introduction, the author should cite the previous work on dopamine neuron specific autophagy deficiency that causes reduced striatal dopamine transmission (Friedman LG, et al., 2012). In general, the Discussion section should be better focused.

Interpretation: No difference between D1 and D2 control morphology, unlike other reports e.g., Gertler 2006. Should be discussed? More of a concern, representative (best traces?) very choppy / noisy, especially for slice recording where baselines are usually very clean. This is a concern because the average mEPSC and mIPSC amplitudes are coming out at 5-7pA, I worry about what has been included in these data (what are the Ra and IRMS for these recordings?). Evoked currents in Figure S2 C, grouped data should be shown, not overlaid curves; moreover for this an n of 7 is not sufficient, such numbers can be achieved in single day / mouse preparation.

Interpretation: Figure 5 For the fractionation experiment I recommend they use the same terminology in the manuscript, the figure and the supplements. In the blot for Kir2.1 and 2.3, a close examination suggests either image handling issue or some weird stitching artefact. There are three white diagonal lines passing in kir2.1 KO P3, and for kir2.3 blot in ctrl S2 and ko SPNs. The reviewers must see (and the document must contain) high resolution original (uncropped) images of blots.

Interpretation: Figure 6 seems clear but where is the proof that the K334R is less acetylated (as in 6.A)? Related, what about the acetylation status of the Kir2.1 or 2.3 in the dSPN Atg7^KO^?

Interpretation: Figure 1) Effect size exaggerated by y axis starting above zero in two cases. Male and female mouse weights are different, good reason to split the data, but this should be maintained for all analyses (clear why it wasn't see below). Further, body weight should be taken into account, for rotarod especially, where weight and performance are tightly (negatively) correlated. Since this is a negative correlation, this may reveal differences between the two KO lines, however, see below. It is also of interest that the D1 mice, despite being bad, improve their performance (learn) over the three trials, as well as controls, whereas d2 do not.

Interpretation: how has the specificity of the Kir2.1 and 2.3 antibody been validated? There is very little info on the suppliers website and their antibody detected Kir2.1 in overexpressing cos cells, but not rat brain? is this a concern? Recommend some in house validation (with flag variants shown later in paper?),

The following areas suffer from "low N" problem. These experiments either need adequate sample size or should not be included in the manuscript:

Figure 1) Only 3 mice assessed for D1 dSPN and p62 ranging from only 50% and up. Justify?

Two of the clearest behavioral phenotypes (latency to fall and grooming bouts) are not even close to confidence at p=0.2 in males, and significance is only achieved by grouping the sexes. To support definitive claims about the effects of these deletions on behavior, the behavioral assessment must be more rigorous.

For slice data the number of cells and the number of animals from which these came should be given and at least 4 or 5 animals minimum; even in other more complete data sets in the main document sometimes the animal n=3, which is why I expect the information is buried in a table and not presented with the data. Recommend this would be at least 12-15 cells, animal n of 4 or 5 and presented throughout. Further, the evoked currents in S2.D appear to be emerging from the stimulus artefact, and more so in the dSPNcko, this will impact upon amplitude measures. The stimulating electrode is likely too close to the cell if there is no latency to the response. The current clamp data is much more clearly produced, the authors might consider removing the voltage clamp results, or increasing their observations a lot.

The three datum in Figure 5—figure supplement 1 are important but far too few against which to conclude there is no change (a strong trend exists) in Kir2.1 for dSPN atg7ko; especially given the huge range shown in figure 4G for iSPN with 9 data points.

---

## [Author Response]

Summary:The authors present a dense body of work that is overall compelling and comprehensive, focusing on mice that combine Drd1-Cre and Adora2A-Cre with Atg7-fl/fl. The report has many strengths – autophagy in both dPSN and iPSN with electrophysiology analysis; describing a novel autophagy substrate Kir2 and potential molecular basis for its degradation and inhibition; and cell type-specific degradation of Kir2 in the brain, etc. This is solid work from a lab that has published many interesting reports on the role of autophagy in neuronal function.

We thank the reviewers for their enthusiasm regarding this manuscript. As detailed below, we have edited and reorganized the manuscript to enhance clarity. In addition, to address reviewer concerns and suggestions, we have added multiple new data sets to the manuscript including: (1) electrophysiological recordings from 60 cells arising from 13 additional mice (including several following stereotaxic surgery), (2) western blot data from 28 additional mice (14 controls, 7 dSPN^Atg7cKO^, and 7 iSPN^Atg7cKO^ mice), (3) behavioral data from a new cohort of 36 animals including rotarod and grooming, as suggested by the reviewers, and (4) cell culture experiments aimed at explaining the mechanism through which acetylation affects Kir2 channel activity. We feel that these additional data substantially strengthen the conclusions in this manuscript and hope that the reviewers agree that it is now suitable for publication in *eLife*.

Essential revisions:1) All reviewers thought the manuscript was still in a draft format, with critical errors in cross-referencing figures, as well as sub-optimal organization for eLife. As described in the compendium of points made below, the manuscript is not ready for publication and was unnecessarily difficult to read through. A careful rewriting and re-editing is necessary.

We thank the reviewers for pointing out the general need to enhance the readability of the study. We have carefully edited the manuscript to improve its flow and organization. This has involved renumbering figures and tables, and to ease the comparison of versions, this table lists the differences in figures and tables between the revised and original submissions.

Revised submissionOriginal submissionAdditional Data in revisionFigure 1Figure S1Figure 2Figure 1Figure 3Figure 2Figure 4Figure 3Figure 4—figure supplement 1Figure S3Figure 5Figure 4Western blot analysis from 14 new mice (p62)Figure 5—figure supplement 1Figure S5Western blot analysis from 28 additional miceFigure 5—figure supplement 2Figure S6Figure 5—figure supplement 3Figure S6Figure 6Figure S4Recordings from 10 additional mice following stereotaxic viral injectionsFigure 7Figure 5Figure 7—figure supplement 1Figure S8Figure 8Figure 6Figure 8—figure supplement 1Not in original submissionAdditional cell culture data demonstrating that Kir2.1 K334R is less acetylated than WT channel and examining surface levels of Kir2.1 K334R and K334Q mutantsRemoved from revised manuscriptFigure S2As suggested by the reviewersTable 1Table S1Additional data have been included for mice of each sex as detailed below.Table 2Table S2Statistical information for all figures have been incorporated in the text of the figure legends, except for Figure 3 which remains in Table 2.Table 3Table S3Table S1Table S4Table S2Table S5Table S3Table S6

2) Overall, reviewers thought that some experiments were underpowered (based on the proposed effect size, or say, a rodent experiment with few animals), and these should be distinguished better from central points (perhaps these data moved to supplement). In turn, some of the supplemental that presented critical data should be moved into the main figure set. Further, sex should be an included as a co-variable and mentioned in all legends, as well as additional clarity as to the total number of animals involved in each experiment. In general, all reviewers felt the strength of the conclusions did not always match the experimental rigor, so the text may need to be more conservatively modified in this regard.

We thank the reviewers for these comments and believe that the additional data in the revised manuscript strengthens its conclusions. We now include 7 additional mice in the analysis of the cell-type specificity of Cre-mediated recombination in Figure 1; 36 mice of both sexes in the behavioral analysis in Figure 2; 13 mice in electrophysiological experiments in Figure 3 and Figure 6; and 28 mice in biochemical experiments in Figure 5 and Figure 5—figure supplement 1.

Please note that for many of the conclusions in this manuscript, we present data from multiple orthogonal approaches (for example, combining in vivo biochemistry with heterologous systems) that provide congruent findings. We hope that the independent approaches, in addition to the new data, will enhance the reviewers’ evaluation of rigor.

We have further largely rewritten the Discussion section to provide a more balanced perspective on our findings.

With regard to including sex as a variable in the manuscript, we now include additional male and female mice of all genotypes in the behavioral analysis of Figure 2. We performed two-way ANOVAs for each behavioral task, with sex and genotype as factors, and did not find a significant interaction between sex and genotype in any behavioral task (see Table 1). We thus conclude that sex does not significantly modify the genotype effect in these experiments and use pooled data from both sexes in subsequent figures. Furthermore, as we did not initially design these experiments to detect sex-related differences, we are underpowered to make claims about any sex differences in the electrophysiology and biochemistry experiments. We have added a statement in the Results section clarifying this:

“Note that we have combined data from both sexes in this analysis. We did not design these experiments to detect sex differences, as there was no interaction between sex and genotype in the behavioral analysis.”

3) The discourse is lacking elements of critical interpretation and alternative explanations, in general, which we think will lessen the impact of the work for some readers. Some relevant (specific) points to consider are listed below:

We acknowledge the reviewers’ deep engagement with our manuscript and appreciate the detailed suggestions. which we feel have strongly improved the manuscript. Our responses and changes to the manuscript are detailed below.

1) Reformatting: Figure 6 (confusingly referred to as Supplemental Figure 5 in the Results section?) should probably be a main figure with a broader discussion.

We apologize for the incorrect reference to this figure in the original submission. We now include this figure as a main figure (Figure 6 in the revised manuscript) as suggested by the reviewers.

In addition, we have extended our description of this experiment in the Results section and added a section about this figure in the Discussion section:

From the Results section:

“Although autophagy has previously been shown to regulate integral plasma membrane proteins by controlling membrane protein trafficking, interaction with partners, or degradation (Gao et al., 2018; Rowland et al., 2006; Shehata et al., 2012; Sumitomo et al., 2018; Yan et al., 2018), the unexpected inverse relationship between total Kir2 protein and Kir2 current in iSPNs lacking Atg7 warranted further validation. […] These results demonstrate that reduced Kir2 current in iSPN^Atg7cKO^ mice could be complemented by Atg7 reexpression, but not Kir2.1 overexpression, in adulthood.”

From the Discussion section:

“We further used a viral overexpression strategy to confirm that the decreased Kir2 activity in iSPNs lacking Atg7 did not arise from reduced Kir2 mRNA expression. Viral overexpression of Kir2.1 failed to increase Kir2 currents in iSPNs lacking Atg7, demonstrating that Atg7 is required for newly synthesized channels to become functional. […] We hypothesize that newly synthesized Kir2.1 in iSPNs lacking Atg7 these experiments is acetylated and does not undergo autophagic degradation, preventing functional increases in Kir2.1 current.”

Similarly, Figure 1 is very important and demonstrates functionally the success of the model in achieving autophagy deficiency in the majority of SPNs (with the D1-tomato being a very nice control here).

We agree with this point and have moved this to main Figure 1 in the revised manuscript.

We also now mention that the ‘floxed’ Atg7 mouse, D1cre, and A2Acre transgenic mice have been previous validated and used widely in the literature (for example (Durieux et al., 2012; Hernandez et al., 2012; Komatsu et al., 2006, 2005, Kozorovitskiy et al., 2015, 2012; Tang et al., 2014)).

There are no space requirements that should necessitate the constant flipping between supplement and main figure files for important experiment results. Along these lines, the abstract is overly tight and awkwardly crafted in flipping back and forth between the iSPN and dSPN data.

We have now incorporated figure supplements in line with the text in the revised manuscript as it would appear in the published *eLife* format to reduce the need to flip between the main text and the supplement. We have incorporated Figure S1 and Figure S4 from the original submission as Figure 1 and Figure 6, respectively, of the revised submission (see response to point 5 below).

We have rewritten the Abstract as follows:

“The basal ganglia are a group of subcortical nuclei that contribute to action selection and reinforcement learning by evaluating sensorimotor and reward inputs and controlling thalamocortical activity and motor output. The principal neurons of the striatum, GABAergic spiny projection neurons of the direct (dSPN) and indirect (iSPN) pathways, maintain low intrinsic excitability and require convergent excitatory inputs to fire and thereby filter information before transmitting it to basal ganglia output nuclei. What cellular mechanisms ensure proper synaptic and intrinsic properties in SPNs? Recent findings indicate that macroautophagy regulates synaptic activity in excitatory neurons, the *Drosophila* neuromuscular junction, and nematode neurons. Here, we examined the role of autophagy in SPN physiology and animal behavior by generating conditional knockouts of Atg7 in either dSPNs or iSPNs. Loss of autophagy in either SPN population led to changes in motor learning but distinct effects on cellular physiology. dSPNs, but not iSPNs, required autophagy for normal dendritic structure and synaptic input. In contrast, iSPNs, but not dSPNs, were intrinsically hyperexcitable. iSPN intrinsic hyperexcitability arose from reduced function of the inwardly rectifying potassium channel, Kir2. In the absence of autophagy, total and surface levels of Kir2 protein were elevated but Kir2 activity was inhibited by acetylation. These findings define a novel mechanism by which autophagy regulates neuronal activity: control of intrinsic excitability via the regulation of potassium channel function.”

2) Interpretation: Does the timing of autophagy deficiency and p62 accumulation with respect to the developmental stage of outgrowth and synapse formation account for some of the phenotypic differences between iSPNs and dSPNs observed? (Does Drd1 and Adora2 expression activate in the exact same phase of SPN maturation, and even with that, could differential Cre strength (i.e., expression of Cre) exact a delay in recombination in iSPNs compared to dSPNs?)

Perhaps the most striking and surprising findings in our study are the divergent consequences that loss of autophagy has on dSPNs and iSPNs. While a possible explanation could be differences in the age at which Cre expression occurs between the D1cre and A2Acre driver lines, the deletions both occur well before the changes in excitability.

In detail, both Cre driver lines used in this study (D1cre ey262 and A2Acre KG139) demonstrate widespread expression in the appropriate cell type by postnatal day 7 (Kozorovitskiy et al., 2015, 2012), which is before the maturation of SPN intrinsic excitability and the main period of striatal synaptogenesis (Lieberman et al., 2018; Peixoto et al., 2016; Tepper et al., 1998). In addition, we confirmed that similar numbers of p62+ cells are present in dSPN^Atg7cKO^ and iSPN^Atg7cKO^ mice at P28, the age at which the electrophysiological recordings in Figure 3 and Figure 4 were conducted (Author response image 1). Furthermore, similar fractions of dSPNs and iSPNs display p62 inclusions in the respective conditional knockouts during adulthood (Figure 1).

**Author response image 1. respfig1:** Similar numbers of p62+ cells in dSPN^Atg7cKO^ and iSPN^Atg7cKO^ striatum at P28. Data were analyzed with a one-way ANOVA followed by a Bonferroni post-hoc test.

Please also note that the approach we have taken to compare the effect of loss of a particular protein in dSPNs or iSPNs (use Credriver lines) has been reported previously in the literature with these same specific Cre driver lines (Daigle et al., 2014; Hutton et al., 2017; Koranda et al., 2018; Kozorovitskiy et al., 2012; Lambot et al., 2016; Rothwell et al., 2014; Tan et al., 2013; Urs et al., 2016, 2012).

This evidence suggests to us that the difference in the timing of Cre expression does not affect the distinct phenotypes observed between conditional knockout lines in this study and emphasizes the validity of this approach in analyzing cell-type specific effects of loss of autophagy on dSPNs and iSPNs. We think that the data provide support for an important implication of this study: that autophagy may be required in distinct cell types at different developmental windows.

In further comparison between cre lines, in the ey262 D1cre driver line, cortical recombination occurs as previously described (Kozorovitskiy et al., 2012), but this is absent in the A2Acre line (data not shown). The loss of autophagy in some populations of cortical neurons may contribute to the synaptic changes in dSPNs and, at this time, we are unable to exclude this possibility; however, we have attempted to address this in two ways. First, if the synaptic deficits arise solely due to presynaptic loss of Atg7, we would expect that mEPSC frequency would be different in iSPNs from dSPN^Atg7cKO^ mice because dSPNs and iSPNs share most cortical inputs (Guo et al., 2015). mEPSC frequency and amplitude was not different in iSPNs from dSPN^Atg7cKO^ compared to dSPN^Atg7ctrl^ mice (Author response image 2). We have chosen not to include this data in the revised submission because we do not have corresponding data in dSPNs from iSPN^Atg7cKO^ mice, but are happy to discuss this if the reviewers feel strongly.

**Author response image 2. respfig2:** No change in mEPSC frequency or amplitude in iSPNs from dSPN^Atg7cKO^ mice. mEPSCs were recorded from iSPNs in dSPN^Atg7cKO^ mice and compared to control iSPNs. We show here that iSPNs from dSPN^Atg7cKO^ mice have equivalent (**A**) mEPSC frequency and (**B**) mEPSC amplitude as iSPNs from control mice. Data from dSPNs arising from control or dSPN^Atg7cKO^ mice are reproduced here for reference from Figure 3. Control iSPN data, also reproduced from Figure 3. Data are analyzed by two-way ANOVA followed by Bonferroni posthoc test. (**A**) Subtype x Genotype: F_(1,75)_=11.57, p=0.0011. (**B**) Subtype x Genotype: F_(1,77)_=4.444, p=0.0383. iSPNs from dSPN^Atg7cKO^ mice: n=11,3. N for other group are the same as in Figure 3.

Second, we note that the paired pulse ratio of evoked excitatory post-synaptic currents (eEPSCs), the classical test of presynaptic release probability, in dSPNs from dSPN^Atg7cKO^ was not different from that of dSPNs from dSPN^Atg7Ctrl^ mice (Author response image 3). This suggests that at the time of our recording, there is no difference in the paired pulse ratio of excitatory inputs onto dSPNs (and so no apparent change in release probability) and, therefore, the change in mEPSC frequency in dSPNS from dSPN^Atg7cKO^ mice does not arise from ongoing dysfunction of presynaptic inputs. As per the reviewer suggestions below (see response to that point), we have not included this data in the revised submission.

We have added a sentence to the Results section mentioning that equal numbers of p62+ cells are present in the striatum at P28. We have also included an additional paragraph in the Discussion section of the revised submission that addresses the possibility that differences between the Cre driver lines may underlie the cell-type specific effects of loss of autophagy in the striatum.

Results section: “Finally, we confirmed that similar numbers of p62+ cells are present in the striatum of dSPNAtg7cKO and iSPNAtg7cKO mice at P28, when electrophysiological recordings in Figure 3 and Figure 4 were collected (data not shown).”

Discussion section: “Perhaps most surprising is the fact that iSPNs that lack Atg7 do not show changes in dendritic complexity, spine density or excitatory inputs. […] Future efforts to define the underlying mechanisms that contribute to cell-type specific roles for autophagy in the striatum could include Credriver lines in which both dSPNs and iSPNs are targeted to rule out a contribution of different expression time courses between the Cre lines used in this study.”

**Author response image 3. respfig3:** No change in paired-pulse ratio (PPR) of excitatory inputs to dSPNs from dSPN^Atg7cKO^ mice. (**A**) Representative traces of evoked EPSCs in dSPNs from control or dSPN^Atg7cKO^ mice. EPSCs were evoked in the presence of 25 μM picrotoxin following electrical stimulation within the striatum >100μm from the cell body. (**B**) No effect of loss of dSPN autophagy on the paired pulse ratio of evoked EPSCs. Control: n=7,2; dSPN^Atg7cKO^: n=11,3. Latency of EPSC is between 5-10 ms in all cells examined.

3) Interpretation: the authors relatively ignore why the iSPNs are relatively resistant to the morphological changes caused by autophagy deficiency seen in other neurons, but more discussion here could help interpretation.

We thank the reviewers for prodding us to expand our discussion about why iSPNs do not demonstrate the morphological changes that we and others have seen in other neuronal populations after loss of autophagy. This is a particularly interesting question given that dSPNs do have these changes.

Unfortunately, little is known about how autophagy may control neuronal morphology (O. J. Lieberman et al., 2019; Nikoletopoulou and Tavernarakis, 2018). Possibilities include changes in mitochondrial function, response to neurotrophic factors, and differential response by the cytoskeleton to neuromodulatory signaling. Future efforts will focus on comparing the effect of loss of autophagy on these molecular pathways in dSPNs and iSPNs. These possibilities are addressed in additional paragraphs in the new Discussion section:

“Excitatory synaptogenesis onto iSPNs may occur in an autophagy-independent manner. Once the mechanism through which autophagy regulates dendritic complexity and synaptogenesis in dSPNs is identified, future studies could compare this process with iSPNs and define why loss of autophagy in iSPNs does not affect iSPN synaptogenesis.”

and

“Autophagy is required for normal synaptic structure and function in *C. elegans*, the *Drosophila* neuromuscular junction, and excitatory neurons in the mouse cortex and hippocampus (Glatigny et al., 2019; Nikoletopoulou et al., 2017; Shehata et al., 2012; Shen and Ganetzky, 2009; Stavoe et al., 2016; Tang et al., 2014; Vanhauwaert et al., 2017; Yan et al., 2018). […] We hypothesize that, given the dynamic changes in autophagic flux that occur in SPNs during early postnatal development (P10-P18; (O. Lieberman et al., 2019)), autophagy may play a role in dendritic growth and excitatory synaptogenesis on dSPNs.”

4) Interpretation: It is intriguing that accumulation of Kir2 occurs only in Atg7 specific deletion iPSN but not dPSN. If Kir2 is indeed a substrate of autophagy as the study indicated, it is puzzling that cKO of Atg7 in dPSN does not cause Kir2 accumulation. The authors could consider in the text the possibility that the phenotypes observed in Atg7 cKO mice are non-autophagy related, unless the authors also examined additional cKO mice in dPSN specific manner with another Atg deletion. Otherwise I suggest modifying the use of "autophagy" only throughout the manuscript in interpreting the data; at least for the conclusion "Atg7-mediated autophagy" could be better.

As suggested by the reviewers, Atg7, although initially identified as required gene for macroautophagy, plays roles in other cellular processes such as LC3-associated phagocytosis and secretion of lysosomal contents (Subramani and Malhotra, 2013). Please note that while we only examined Atg7 conditional knockout mice, we found that Kir2.1 degradation was disrupted in both Atg7^KO^ and Atg5^KO^ MEFs, suggesting additional autophagy-associated genes are required for the phenotype we observe. Nevertheless, the morphological and synaptic deficits observed in dSPN^Atg7cKO^ may arise from an autophagy-independent role of Atg7, and moreover, Kir2 degradation may occur through autophagy-independent but Atg5 and Atg7-dependent processes. As the reviewer suggests, we have modified the language in our manuscript to emphasize that we are addressing the role of Atg7 in these neuronal population, and that this may affect additional cellular processes besides autophagy. We have added three sections to the Discussion section that address this point as follows:

We have added a section to the Discussion raising the possibility that the observed phenotypes are not mediated by autophagy per se:

“Atg7 is also involved in non-autophagic processes such as LC3-associated phagocytosis and secretion of lysosomal contents (Subramani and Malhotra, 2013). It is possible that Atg7 contributes to dSPN dendritic morphology and synaptic inputs via these processes as opposed to “classical” autophagic degradation. Future studies comparing loss of other autophagy-associated genes that do not contribute to LC3-associated phagocytosis or lysosomal exocytosis may provide further insight.”

We have also included an expanded section in the Discussion section about the cell-type specific differences in Kir2 degradation observed in the striatum:

“In contrast to iSPNs, Kir2.1 and Kir2.3 abundance and Kir2 currents were unaffected by loss of autophagy in dSPN^Atg7cKO^ mice. As Kir2 channels are expressed in both populations of SPNs and play important roles in both dSPN and iSPN physiology (Cazorla et al., 2012; Gertler et al., 2008; Lieberman et al., 2018; Shen et al., 2007), these data suggest that the degradation of Kir2 channels by autophagy occurs via cell type-specific mechanisms. This cell-type specificity could arise from differential expression of cargo adapters, post-translational modification of Kir2 channels that target them for autophagic degradation, or increased activity of alternative endolysosomal protein degradation pathways that contribute to Kir2 degradation in cardiomyocytes (Ambrosini et al., 2014; Jansen et al., 2008; Kolb et al., 2014). Future work will focus on the mechanism that confers cell-type specificity to the autophagic degradation of Kir2 channels.”

Finally, we have added a paragraph to the Discussion section outlining the evidence for our conclusion that Atg7 is required for autophagic degradation of Kir2 channels:

“Autophagy associated genes, such as Atg5 and Atg7, have been proposed to play a role in other cellular processes such as LC3-associated phagocytosis, lysosomal exocytosis and endocytosis (Subramani and Malhotra, 2013). Here, we argue that Kir2.1 is a direct substrate of autophagy for the following reasons. First, Kir2.1 degradation is deficient in both Atg5^KO^ and Atg7^KO^ MEFs. Additional studies in cells lacking other required autophagy proteins such as Fip200 would further support this hypothesis. Second, we have demonstrated direct colocalization with LC3^+^ vesicles and that the amount of Kir2.1 in LC3^+^ vesicles increases in response to blockade of autophagosome-lysosome fusion. Finally, Kir2.1 internalization, measured with antibody feeding assays, is not affected in Atg5^KO^ MEFs, arguing against a role of Atg5 in endocytosis of Kir2.1. Collectively, these data indicate that Kir2.1 must pass through an autophagy-dependent organelle, presumably an amphisome, before undergoing lysosomal degradation.”

5) Interpretation: Reviewers were not satisfied by the stated explanation for the inhibition of Kir2 activity due to acetylation caused by lack of Atg7 or autophagy in iPSN. For example, the increased level of Kir2 and lack of Kir2-over-expression rescue seemingly necessitates a major fraction of Kir2 protein acetylated. But in theory those Kir2 that are destined for degradation only will be modified and inhibited. Therefore, additional mechanisms could be involved in the inhibition.

We thank the reviewers for pointing this out. We agree that our data would suggest a model in which a large fraction of Kir2 channels would be acetylated in iSPNs or MEFs lacking autophagy. We propose that Kir2 channels must be acetylated to undergo autophagic degradation, and that this process proceeds unchecked in iSPNs or MEFs lacking autophagy as the acetylated channel is no longer degraded by autophagy or deacetylated in an autophagy-dependent organelle.

Although this model explains the data we present in this manuscript, alternative possibilities exist that we have not ruled out. For example, Kir2.1 may be unable to interact with a required cofactor in Atg5^KO^ MEFs or iSPNs lacking Atg7. Alternatively, we cannot exclude changes in Kir2 localization that our techniques were not sensitive enough to detect.

To address these alternative mechanisms for Kir2.1 inhibition, we have conducted additional experiments and added a paragraph to the Discussion section.

First, we measured the effect of increasing PIP2 levels in iSPNs lacking autophagy. PIP2 is a required lipid cofactor for Kir2 activity. To address whether this interaction is reduced in iSPNs lacking autophagy, we included diC8-PIP2, a water soluble PIP2 analog, which is sufficient to rescue reduced Kir2 activity in other models of SPN hyperexcitability (Lieberman et al., 2018), to our internal solution but found no effect on Kir2 currents (Author response image 4). In WT cells, diC8-PIP2 does not have an effect on Kir2 currents as the PIP2 binding site on Kir2 channels are

saturated under baseline conditions (Logothetis et al., 2007; Xie et al., 2008). In contrast, if decreases in PIP2 binding to Kir2 were responsible for the Kir2 inhibition in autophagy-deficient cells, diC8-PIP2 should increase Kir2 currents. The absence of an effect of diC8-PIP2 in autophagy-deficient iSPNs suggests that the loss of autophagy does not affect Kir2 interaction with PIP2. We have chosen not to include this in the revised manuscript as this was not part of a comprehensive characterization of Kir2 interactors, but can include it if the reviewers feel strongly that it should be a part of

this manuscript.

**Author response image 4. respfig4:** Increasing cellular PIP_2_ levels does not rescue Kir2 currents in Atg7-deficient iSPNs. (**A**) diC8-PIP2 (50 μM) was included in the intracellular solution. Percent change in Kir2 currents after 12 minutes of dialysis with the internal solution ((Lieberman et al., 2018). No effect of diC8-PIP2 was found in either genotype. Data analyzed with two-way ANOVA followed by Bonferroni test. Genotype x PIP2: F_(1,29)_=1.124, p=0.2978; Genotype: F_(1,29)_=0.8626, p=0.3607; PIP2: F_(1,29)_=0.03962, p=0.8436. Each condition arose from >3 mice each.

Second, we conducted experiments aimed at understanding whether acetylation affects Kir2 surface residence or instead affects channel function on the surface (Figure 8 – Figure Supplement 1). We quantified the surface levels of WT, K334R and K334Q Kir2.1

in both Atg5WT and AtgKO MEFs. In this experiment, we reproduced the increased surface levels of Kir2.1 WT in Atg5KO compared to Atg5WT MEFs. However, the surface levels of the K334R and K334Q mutants were not different that WT Kir2.1 in either Atg5WT or Atg5KO MEFs. These data provide additional mechanistic insight into how acetylation status of Kir2.1 at K334 could affect Kir2 channel activity: it must do so by regulating function at the surface as opposed to controlling surface residence. Future studies aimed at defining the biophysical mechanism for Kir2 channel inhibition in this system are warranted.

We have added an additional section to the Discussion where we enumerate alternatives to the model we propose:

“Several alternative models could explain our data including decreased levels of a required cofactor for Kir2 in autophagy-deficient cells and changes in membrane localization that are below the detection limit of our approaches. Future examination of the Kir2 interactome in wild-type and autophagy-deficient cells could elucidate these mechanisms.”

How common is it for the Kir2 channel to be degraded by autophagy in CNS neurons in general? Would dPSN be an unusual cell type for Kir2 not being degraded by autophagy? Or autophagy is not essential for Kir2 degradation in dPSN? Speculate the mechanism?

Kir2 channel degradation has not been examined in any detail in other CNS neurons. However, in cardiomyocytes, strong evidence supports a role for lysosomal degradation, but it is unclear if this occurs downstream of autophagy per se and/or other endosomal degradation pathways. Additional experiments comparing Kir2 channel handling in other neuronal populations would provide insight into these processes.

With regard to the lack of Kir2 accumulation in dSPN^Atg7cKO^ mice, we do not believe that this affects our conclusion that in iSPNs (and in cultured MEFs) Kir2 is degraded via an Atg5/7-dependent process (referred to here as autophagy). The evidence for autophagic degradation of Kir2 is: (1) Kir2 degradation is dependent on Atg7 and Atg5 in MEFs, and Atg7 in iSPNs; (2) Kir2.1 localizes in an LC3+ compartment in an Atg5-dependent manner in MEFs; (3) raising the lysosomal pH by treating cells with bafilomycin increases localization of Kir2.1 to an LC3+ compartment.

In dSPNs, alternative mechanisms must be in place for Kir2 degradation. For example, in cardiomyocytes, Kir2 channels may be degraded via alternative endolysosomal mechanisms, although the interaction between Kir2 and autophagy has not been formally examined in the heart (Ambrosini et al., 2014; Jansen et al., 2008; Kolb et al., 2014). At a molecular level, differences in the localization, expression, or activity of the (yet unidentified) acetyltransferase required for Kir2 degradation in some cell types may not guide Kir2 towards autophagic degradation in dSPNs. Alternatively, the (yet unidentified) adapter protein that recruits Kir2 to the Atg7-dependent autophagic intermediate may also be differentially expressed in dSPNs. Finally, we note that the absence of Kir2 accumulation in dSPN^Atg7cKO^ may not mean that Kir2 is not degraded by autophagy under basal conditions, but may instead suggest that dSPNs, but not iSPNs, possess a compensatory mechanism that is activated in the absence of autophagy and prevent Kir2 accumulation. Ongoing efforts in our lab are aimed at defining the molecular mechanisms that promote autophagic degradation of Kir2 and their identification will lead to new hypotheses about the difference between Kir2 trafficking in dSPNs and iSPNs.

We have included a new paragraph in the Discussion section speculating on possible mechanisms for the cell-type specific degradation of Kir2.1, as shown above in response to Intepretation point 4.

6) Interpretation: In the second paragraph of the Introduction, the author should cite the previous work on dopamine neuron specific autophagy deficiency that causes reduced striatal dopamine transmission (Friedman LG, et al., 2012). In general, the Discussion section should be better focused.

We thank the reviewer for pointing out this paper and apologize for the omission. We now cite this paper in the Introduction when describing the role of autophagy in neurotransmission. Note that this study did not show reduced striatal dopamine transmission, as they dissected the striatum and measured dopamine content with HPLC, rather than the release of transmitter.

7) Interpretation: No difference between D1 and D2 control morphology, unlike other reports e.g., Gertler 2006. Should be discussed? More of a concern, representative (best traces?) very choppy / noisy, especially for slice recording where baselines are usually very clean. This is a concern because the average mEPSC and mIPSC amplitudes are coming out at 5-7pA, I worry about what has been included in these data (what are the Ra and IRMS for these recordings?). Evoked currents in Figure S2 C, grouped data should be shown, not overlaid curves; moreover for this an n of 7 is not sufficient, such numbers can be achieved in single day / mouse preparation.

The literature regarding differences between dSPN and iSPN morphology is controversial. Although Gertler and Surmeier (Gertler et al., 2008) report differences in dendritic length between SPN subtypes, others do not find differences (Cepeda et al., 2008; Suárez et al., 2014). In our data, we do not see a clear difference in total dendritic length or dendritic complexity between control dSPNs and iSPNs; however, we note that this was not a preplanned comparison. We have added a sentence to the Discussion section where we note this difference that exists in the literature:

“Although we did not observe a difference between the dendritic morphology of dSPNs and iSPNs, there are conflicting reports on morphological differences between dSPNs and iSPNs (Cepeda et al., 2008; Gertler et al., 2008; Suárez et al., 2014). Future studies must aim to reconcile these disparate findings.”

More of a concern, representative (best traces?) very choppy / noisy, especially for slice recording where baselines are usually very clean. This is a concern because the average mEPSC and mIPSC amplitudes are coming out at 5-7pA, I worry about what has been included in these data (what are the Ra and IRMS for these recordings?). Evoked currents in Figure S2 C, grouped data should be shown, not overlaid curves; moreover, for this an n of 7 is not sufficient, such numbers can be achieved in single day / mouse preparation.

Regarding the mEPSC and mIPSCs, we apologize for the way the sample traces were rendered in Figure 2 from the original submission. We have replaced the corresponding sample traces in Figure 3 of the revised submission.

The root mean square noise on the rig in which this data was collected is consistently between 1.5-2.5 pA. The median mEPSC amplitudes in our recordings are ~ 10 pA, >3x the rms noise. Series resistance was monitored online both before and after the recorded epochs in which mEPSCs or mIPSCs were measured. If series resistance was >25 MOhms or changed by >20%, the cells were excluded. These criteria are clearly stated in the Materials and methods section of the original and revised manuscripts.

In addition, we analyzed the data using a custom Igor script that has been published and used repeatedly by our lab and others (Mosharov and Sulzer, 2005). Peaks are detected using the first derivative of the trace and the amplitude of the called peaks must then exceed a user-defined number of standard deviations (SD) of the root mean square noise. The SD cutoffs were set as 4x for mEPSCs and 3x for mIPSCs as described in the methods section. We are thus confident that our mEPSC and mIPSC data contain appropriate information that endorses the conclusions from Figure 3 of the revised manuscript.

Regarding the evoked EPSCs, as per the reviewers’ suggestion below, we have removed the input/output curves originally in Figure S2 of the original submission. Nevertheless, we emphasize that as stated in the Materials and methods section: all electrophysiology data arose from at least 3 mice (usually 4-5 mice) and all cell culture experiments were independently replicated in at least three separate experiments.

Interpretation: Figure 5 For the fractionation experiment I recommend they use the same terminology in the manuscript, the figure and the supplements. In the blot for Kir2.1 and 2.3, a close examination suggests either image handling issue or some weird stitching artefact. There are three white diagonal lines passing in kir2.1 KO P3, and for kir2.3 blot in ctrl S2 and ko SPNs. The reviewers must see (and the document must contain) high resolution original (uncropped) images of blots.

We thank the reviewer for pointing out inconsistencies in our references to different fractions from the fractionation experiment. We note that we conducted two orthogonal fractionation approaches and, thus, the terminology is distinct between the fractionation experiment in Figure 6 and in Figure 6—figure supplement 1.

We acquired the images of western blots in Figure 5—figure supplement 2, Figure 7—figure supplement 1, Figure 7, Figure 8, and Figure 8—figure supplement 1 on an Azure Biosystems C600 with the default settings. Beyond cropping the images to highlight the relevant band within the blot, no image editing or stitching was done. The three white diagonal lines mentioned by the reviewer are present on the original image from the Azure machine. Western blots in other figures were acquired using film and digitized with a scanner or with the LICOR Odyssey system and do not have this artifact.

We include full length blots for all western blots in the Supplementary files.

Interpretation: Figure 6 seems clear but where is the proof that the K334R is less acetylated (as in 6.A)? Related, what about the acetylation status of the Kir2.1 or 2.3 in the dSPN Atg7^KO^?

To address whether Kir2.1 K334R is less acetylated than WT channel, we immunoprecipitated WT or K334R Kir2.1 from Atg5^KO^ MEFs and immunoblotted for acetyllysine residues and Flag as in Figure 7A. We found an approximately 50% decrease in acetylation on Kir2.1 K334R relative to WT channel in Atg5^KO^ MEFs and have included this data in Figure 7 – figure supplement 1. We conclude that Kir2.1 is acetylated on multiple residues, but that the K334R mutant significantly reduces total acetylation of the channel.

We have attempted to isolate acetylated Kir2.1 or Kir2.3 from mouse striatum but have been unable to successfully immunoprecipitate the channel with the available antibodies against the endogenous protein. There are further no reports in the literature of immunoprecipitation of endogenous Kir2 channel from mouse brain.

Interpretation: Figure 1) Effect size exaggerated by y axis starting above zero in two cases. Male and female mouse weights are different, good reason to split the data, but this should be maintained for all analyses (clear why it wasn't see below). Further, body weight should be taken into account, for rotarod especially, where weight and performance are tightly (negatively) correlated. Since this is a negative correlation, this may reveal differences between the two KO lines, however, see below. It is also of interest that the D1 mice, despite being bad, improve their performance (learn) over the three trials, as well as controls, whereas d2 do not.

We thank the reviewers for these comments. We address them individually.

1) While the y-axis begin above zero more clearly displays the relative differences between groups in Figure 1A and 1E (Figure 1 of the original submission, Figure 2 in the revised submission), especially considering no mice weighed less than 15 g, we have changed the y-axis for both panels to begin at zero.

2) Rotarod performance indeed has a significant inverse correlation with body weight (McFadyen et al., 2003). We agree that this may suggest that the dSPN^Atg7cKO^ mice may be performing worse on the rotarod than iSPN^Atg7cKO^ mice because their latency to fall was similar, but they weighed significantly less. We have added a sentence to the Results addressing this possibility.

“We note that differences in motor learning may exist between dSPN^Atg7cKO^ and iSPN^Atg7cKO^ mice as the rotarod test is highly sensitive to body weight (McFadyen et al., 2003) and dSPN^Atg7cKO^ mice weight significantly less than iSPN^Atg7cKO^ mice (Figure 2A).”

3) We agree that one possible interesting feature of this experiment is that dSPN^Atg7cKO^ mice, in contrast to iSPN^Atg7cKO^ mice, seem to improve their performance over three trials, suggesting that they learn the task better. To examine this, we compared the difference in latency to fall between trial 1 and trial 3 in control, dSPN^Atg7cKO^ and iSPN^Atg7cKO^ mice (Author response image 5). We found that the learning was not significantly different between control and dSPN^Atg7cKO^ mice but was significantly worse in iSPN^Atg7cKO^ mice. There was not a significant difference between the learning of dSPN^Atg7cKO^ mice and iSPN^Atg7cKO^ mice. We therefore chose not to include a statement in the manuscript addressing differences in learning of the two conditional knockout mice.

**Author response image 5. respfig5:** Motor learning differences between control, dSPN^Atg7cKO^ and iSPN^Atg7cKO^ mice. The difference in latency to fall between trial 3 and trial 1 were plotted for each mouseincluded in Figure 2B. Data was analysed with a one-way ANOVA followed by a Bonferroni post-hoc test.

Interpretation: how has the specificity of the Kir2.1 and 2.3 antibody been validated? There is very little info on the suppliers website and their antibody detected Kir2.1 in overexpressing cos cells, but not rat brain? is this a concern? Recommend some in house validation (with flag variants shown later in paper?).

The Kir2.1 and Kir2.3 antibodies used in this study have been used in the literature in mouse brain, in particular in striatal lysates (Cazorla et al., 2012; Lieberman et al., 2018). As suggested, we have validated the Kir2.1 antibody using our Flag tagged variants in MEFs (Author response image 6). We find that the Kir2.1 antibody detects a band of the correct size (~75 kDa; Kir2.1 + Flag tag + SNAP tag) in Atg5^WT^ and Atg5^KO^ MEFs transfected with this construct but does not detect a band in untransfected sister cultures. This band was also detected by the Anti-Flag antibody. Note also the increase in p62 in Atg5KO MEFs and that the anti-Kir2.3 antibody does not detect a band in any of the lysates. This experiment has been repeated 4 times.

**Author response image 6. respfig6:** Specificity of Kir2.1 antibody. Lysates were generated from Atg5^WT^ and Atg5^KO^ MEFs, with or without transfection with Kir2.1^FlagSNAP^ and blotted for Flag, Kir2.1, Kir2.3, p62 and actin.

The following areas suffer from "low N" problem. These experiments either need adequate sample size or should not be included in the manuscript:Figure 1) Only 3 mice assessed for D1 dSPN and p62 ranging from only 50% and up. Justify?

Both the D1cre (eye262) and A2Acre (KG139) driver lines have been previously characterized and are extensively utilized in the study of different striatal cell types, and we chose to characterize the efficiency and specificity of the Cre-mediated recombination of the “floxed” Atg7 allele in these animal models. To increase the confidence in our analysis, we performed immunohistochemistry on 3 additional control mice (Atg7^Fl/Fl^D1T), 3 additional dSPNAtg7^cKO^ mice (D1cre Atg7^Fl/Fl^ D1T) and 1 additional iSPNAtg7^cKO^ mouse(A2Acre Atg7^Fl/Fl^ D1T). We have combined this data with data from the original submission in Figure 1—figure supplement 1 (see discussion above). We find that ~75-80% of dSPNs have p62 “aggregates,” which is a proxy for loss of autophagy (Klionsky et al., 2016; Komatsu et al., 2007; Tang et al., 2014). In addition, we find that approximately 15% of the p62+ cells are D1-tomato negative (putative iSPNs). Although this suggests that Cre-mediated recombination in this line is not 100% efficient, and has some non-specific recombination, these numbers are similar to existing reports in the literature and give us confidence that these animal models allow us to examine the role of autophagy in dSPN physiology.

Two of the clearest behavioral phenotypes (latency to fall and grooming bouts) are not even close to confidence at p=0.2 in males, and significance is only achieved by grouping the sexes. To support definitive claims about the effects of these deletions on behavior, the behavioral assessment must be more rigorous.

We agree that increasing the number of animals in our analysis would increase our confidence in the conclusions. To address this, we generated additional cohorts of dSPN^Atg7cKO^ mice and iSPN^Atg7cKO^ mice and respective littermate controls. These new cohorts included 15 controls (9 male and 6 female), 11 dSPN^Atg7cKO^ mice (7 male and 3 female) and 10 iSPN^Atg7cKO^ mice (5 male and 5 female). After adding the data from this cohort to our analysis (including the data from the original submission), we again found no interaction between sex and genotype following two-way ANOVA, suggesting that it is appropriate to combine males and females in our final analysis. For completeness, as in the original manuscript, we have included all data from these behavioral experiments in Table 1, separating them both by sex and by genotype. We hope that this additional cohort of mice satisfies the reviewers’ concerns about the rigor of the behavioral experiments.

For slice data the number of cells and the number of animals from which these came should be given and at least 4 or 5 animals minimum; even in other more complete data sets in the main document sometimes the animal n=3, which is why I expect the information is buried in a table and not presented with the data. Recommend this would be at least 12-15 cells, animal n of 4 or 5 and presented throughout.

We thank the reviewers for these comments. In the original submission, we reported the N for our electrophysiological experiments as follows: we conveyed the number of cells in our electrophysiological analyses in the bar graphs (with individual points representing single cells). For clarity of the figure legends (as suggested in the guide to authors on the *eLife* website), we included the number of animals for each group in a supplemental table along with detailed information about the statistical tests run.

To address the reviewers’ point, we have taken a two-pronged approach. First, we have included recordings from 13 additional mice (>60 additional cells), increasing the number of cells recorded and reconstructed in Figure 3, and the number of cells recorded from mice with viral injections in Figure 6. In all experiments, the recorded cells come from at least 4 mice. Second, we explicitly included the number of cells and the number of animals, the t or F statistic and statistical test used as well as exact p values in the figure legends. We have kept Table 2 with detailed information about the mean and error for each observation, the number of cells and mice, the statistical test used, t or F value (as appropriate), and p value in Figure 3 as this figure legend is unmanageable when this information is included and have added a sentence to the Results section of the manuscript highlighting the presence of this resource. We hope that these changes strengthen the reviewers’ and editor’s confidence in these conclusions.

Further, the evoked currents in S2.D appear to be emerging from the stimulus artefact, and more so in the dSPNcko, this will impact upon amplitude measures. The stimulating electrode is likely too close to the cell if there is no latency to the response. The current clamp data is much more clearly produced, the authors might consider removing the voltage clamp results, or increasing their observations a lot.

We thank the reviewer for this comment and have removed the data about evoked EPSCs (eEPSCs) from the supplemental data. However, we would like to emphasize that we see a latency of 510 ms for our recorded eEPSCs and they are blocked by NBQX, an AMPAR antagonist, supporting the conclusion that they are synaptically evoked responses.

The three datum in Figure 5—figure supplement 1 are important but far too few against which to conclude there is no change (a strong trend exists) in Kir2.1 for dSPN atg7ko; especially given the huge range shown in Figure 4G for iSPN with 9 data points.

To address this concern, we have generated striatal lysates from 28 additional mice (14 controls, 7 dSPN^Atg7cKO^, and 7 iSPN^Atg7cKO^) and blotted them for Kir2.1, Kir2.3, p62, Kv1.2, PSD95 and actin. The new data is included in Figure 5 and Figure 5—figure supplement 1. No conclusion was changed.